# Diffusion Bridge AutoEncoders for Unsupervised Representation Learning

**Yeongmin Kim**[1], **Kwanghyeon Lee**[1], **Minsang Park**[1], **Byeonghu Na**[1], **Il-Chul Moon**[1,2]
[1]Korea Advanced Institute of Science and Technology (KAIST), [2]summary.ai
{alsdudrla10,rhkdgus0414,pagemu,byeonghu.na,icmoon}@kaist.ac.kr

## Abstract

Diffusion-based representation learning has achieved substantial attention due to its promising capabilities in latent representation and sample generation. Recent studies have employed an auxiliary encoder to extract a corresponding representation from data and adjust the dimensionality of a latent variable $\mathbf{z}$. Meanwhile, this auxiliary structure invokes an *information split problem*; the information of each data instance $\mathbf{x}_0$ is divided into diffusion endpoint $\mathbf{x}_T$ and encoded $\mathbf{z}$ because there exist two inference paths starting from the data. The latent variable modeled by the diffusion endpoint $\mathbf{x}_T$ has several disadvantages. The diffusion endpoint $\mathbf{x}_T$ is computationally expensive to obtain and inflexible in terms of dimensionality. To address this problem, we introduce Diffusion Bridge AutoEncoders (DBAE), which enables $\mathbf{z}$-dependent endpoint $\mathbf{x}_T$ inference through a feed-forward architecture. This structure creates an information bottleneck at $\mathbf{z}$, ensuring that $\mathbf{x}_T$ depends on $\mathbf{z}$ during its generation. This results in $\mathbf{z}$ holding the full information of the data. We propose an objective function for DBAE to enable both reconstruction and generative modeling, with theoretical justification. Empirical evidence demonstrates the effectiveness of the intended design in DBAE, which notably enhances downstream inference quality, reconstruction, and disentanglement. Additionally, DBAE generates high-fidelity samples in an unconditional generation. Our code is available at https://github.com/aailab-kaist/DBAE.

## 1 Introduction

Unsupervised representation learning is a fundamental topic within the latent variable generative models (Hinton et al., 2006; Kingma & Welling, 2014; Higgins et al., 2017; Chen et al., 2016; Jeff; Alemi et al., 2018). Effective representation supports better downstream inference as well as realistic data synthesis. Variational autoencoders (VAEs) (Kingma & Welling, 2014) are frequently used because they inherently include latent representations with flexible dimensionality. Generative adversarial networks (GANs) (Goodfellow et al., 2014) with inversion (Abdal et al., 2019; 2020) are another method to find latent representations. Additionally, diffusion probabilistic models (DPMs) (Ho et al., 2020; Song et al., 2021c) have achieved state-of-the-art performance in terms of generation quality (Dhariwal & Nichol, 2021), naturally prompting efforts to explore unsupervised representation learning within the DPM framework (Preechakul et al., 2022; Zhang et al., 2022; Yue et al., 2024), which have recently dominated generative representation learning studies.

DPMs are a type of latent variable generative model, but inference on latent variables is not straightforward. DPMs progressively map from data $\mathbf{x}_0$ to a latent endpoint $\mathbf{x}_T$ via a predefined noise injection schedule, which does not facilitate learnable encoding. DDIM (Song et al., 2021a) introduces an ODE-based deterministic encoding from the data $\mathbf{x}_0$ to the endpoint $\mathbf{x}_T$. However, this encoding is determined by the choice of the forward process (Song et al., 2021c). Since the forward process with fixed noise injection is difficult to interpret as having semantic meaning, the ODE-based encoding remains challenging to consider as an effective semantic representation. Moreover, the encoding $\mathbf{x}_0$ into $\mathbf{x}_T$ is expensive because it requires solving the ODE, and its inflexible dimensionality poses disadvantages for downstream applications (Sinha et al., 2021).

To tackle this issue, recent DPM-based representation learning studies (Preechakul et al., 2022; Zhang et al., 2022; Kim et al., 2022b; Wang et al., 2023; Yang et al., 2023; Yue et al., 2024; Wu &

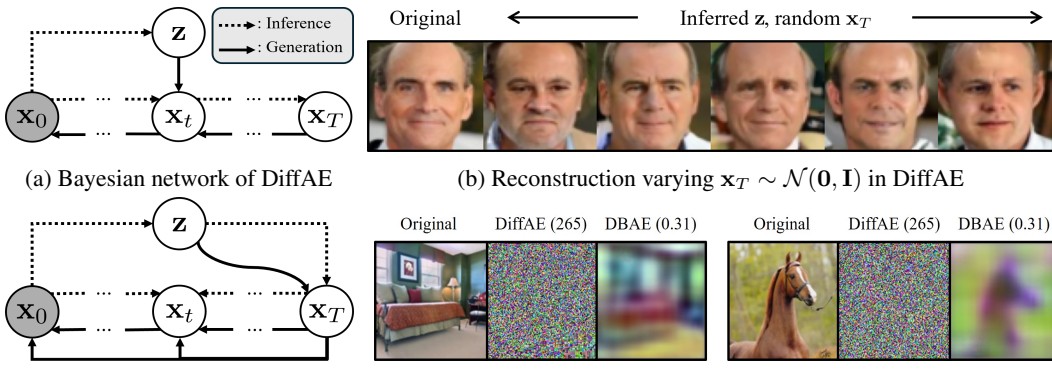

(a) Bayesian network of DiffAE

(b) Reconstruction varying $\mathbf{x}_T \sim \mathcal{N}(\mathbf{0}, \mathbf{I})$ in DiffAE

(c) Bayesian network of DBAE

(d) Inferred $\mathbf{x}_T$ from DiffAE and DBAE with inference time (ms).

Figure 1: Comparison between DiffAE (Preechakul et al., 2022) and DBAE. (a) depicts the simplified Bayesian network of DiffAE, illustrating two inference paths for the distinct latent variables $\mathbf{x}_T$ and $\mathbf{z}$. (b) shows the reconstruction using the inferred $\mathbf{z}$ in DiffAE on CelebA, where the reconstruction results perceptually vary depending on the selection of $\mathbf{x}_T$. (c) shows the simplified Bayesian network of DBAE with $\mathbf{z}$-dependent $\mathbf{x}_T$ inference. (d) shows the inferred $\mathbf{x}_T$ from DiffAE and DBAE.

Zheng, 2024) suggest an auxiliary latent variable $\mathbf{z}$ with an encoder used in VAEs, to combine the generation performance of diffusion models and the representation learning capabilities of VAEs. The encoder-generated latent variable $\mathbf{z}$ is obtained without solving the ODE, and the encoder also facilitates the learning of semantic representations with dimensionality reduction. The reconstruction capability from the extracted latent representation $\mathbf{z}$ is the primary focus of these studies, facilitating downstream inference, attribute manipulation, and interpolation.

This paper points out the remaining problem in auxiliary encoder models, which we refer to as the *information split problem*, hindering reconstruction capability. The information is not solely retained in the latent variable $\mathbf{z}$; rather, a portion is also distributed into the latent variable $\mathbf{x}_T$ as evidenced by Figure 1b. If the auxiliary encoder models only infer $\mathbf{z}$ and reconstruct using a random $\mathbf{x}_T$, the facial details of the original image are not properly reconstructed, indicating that the missing information is contained within $\mathbf{x}_T$. Furthermore, the inference of $\mathbf{x}_T$ is computationally expensive and inflexible in dimensionality. To address this issue, we introduce Diffusion Bridge AutoEncoders (DBAE), which incorporate $\mathbf{z}$-dependent endpoint $\mathbf{x}_T$ inference using a feed-forward architecture.

The proposed model DBAE systematically resolves the *information split problem*. Unlike the two split inference paths in the previous approach in Figure 1a, DBAE encourages $\mathbf{z}$ to become an information bottleneck during inference (dotted line in Figure 1c), making $\mathbf{z}$ more informative. DBAE establishes this bottleneck structure by defining a learnable forward process that starts from the data $\mathbf{x}_0$ and ends at the encoded endpoint $\mathbf{x}_T$ by utilizing Doob's $h$-transform. Moreover, DBAE does not require solving an ODE to infer endpoint $\mathbf{x}_T$, thereby making endpoint inference more efficient, as shown in Figure 1d. This efficient inference of $\mathbf{x}_T$ benefits interpolation and attribute manipulation tasks. In experiments, DBAE outperforms the previous works in downstream inference quality, reconstruction, disentanglement, and unconditional generation. DBAE also demonstrates satisfactory results in interpolation and attribute manipulation with its qualitative advantages.

## 2 PRELIMINARIES

### 2.1 DIFFUSION MODELS

Diffusion probabilistic models (DPMs) (Sohl-Dickstein et al., 2015; Ho et al., 2020) with a continuous time formulation (Song et al., 2021c) define a forward stochastic differential equation (SDE)

$$\mathrm{d}\mathbf{x}_t = \mathbf{f}(\mathbf{x}_t, t)\mathrm{d}t + g(t)\mathrm{d}\mathbf{w}_t, \quad \mathbf{x}_0 \sim q_{\text{data}}(\mathbf{x}_0), \tag{1}$$

where $\mathbf{w}_t$ denotes a standard Wiener process, $\mathbf{f} : \mathbb{R}^d \times [0, T] \to \mathbb{R}^d$ is a drift term, and $g : [0, T] \to \mathbb{R}$ is a volatility term. Eq. (1) starts from data distribution $q_{\text{data}}(\mathbf{x}_0)$ and gradually perturbs it into noise $\mathbf{x}_T$. Let the marginal distribution of Eq. (1) at time $t$ be denoted as $\tilde{q}_t(\mathbf{x}_t)$. There exists a unique

reverse-time SDE (Anderson, 1982)

$$\mathrm{d}\mathbf{x}_t = [\mathbf{f}(\mathbf{x}_t, t) - g^2(t)\nabla_{\mathbf{x}_t} \log \tilde{q}_t(\mathbf{x}_t)]\mathrm{d}t + g(t)\mathrm{d}\bar{\mathbf{w}}_t, \quad \mathbf{x}_T \sim p_{\mathrm{prior}}(\mathbf{x}_T), \tag{2}$$

where $\bar{\mathbf{w}}_t$ denotes a reverse-time Wiener process, $\nabla_{\mathbf{x}_t} \log \tilde{q}_t(\mathbf{x}_t)$ is the time-dependent score function, and $p_{\mathrm{prior}}(\mathbf{x}_T)$ stands for the prior distribution, which closely resembles a Gaussian distribution with the specific form of $\mathbf{f}$ and $g$ (Song et al., 2021c; Ho et al., 2020). Eq. (2) traces back from noise $\mathbf{x}_T$ to data $\mathbf{x}_0$. The reverse-time ordinary differential equation (ODE)

$$\mathrm{d}\mathbf{x}_t = [\mathbf{f}(\mathbf{x}_t, t) - \frac{1}{2}g^2(t)\nabla_{\mathbf{x}_t} \log \tilde{q}_t(\mathbf{x}_t)]\mathrm{d}t, \quad \mathbf{x}_T \sim p_{\mathrm{prior}}(\mathbf{x}_T), \tag{3}$$

produces a marginal distribution identical to Eq. (2) for all $t$, offering an alternative generative process while confining the stochasticity of the trajectory solely to $\mathbf{x}_T$. To construct both reverse SDE and ODE, the diffusion model estimates a time-dependent score function $\nabla_{\mathbf{x}_t} \log \tilde{q}_t(\mathbf{x}_t) \approx \mathbf{s}_\theta(\mathbf{x}_t, t)$ using a neural network and the score-matching objective (Vincent, 2011; Song & Ermon, 2019).

## 2.2 LATENT REPRESENTATION LEARNING WITH DIFFUSION MODELS

From the perspective of representation learning, the ODE in Eq. (3) (a.k.a DDIM (Song et al., 2021a) in discrete time diffusion formulation) provides a deterministic encoding from the data $\mathbf{x}_0$ to the latent $\mathbf{x}_T$. However, the latent representation $\mathbf{x}_T$ has some disadvantages. First, it is hard to learn its semantic meaning. This encoding is determined by the forward process $(\mathbf{f}, g)$ given a data distribution and assuming perfect optimization (Song et al., 2021c). The forward process $(\mathbf{f}, g)$ is set to a fixed noise injection process, but the noise is hard to consider as a semantically meaningful encoding. Second, the dimension cannot be reduced. According to the definition of the diffusion process in Eq. (1), the dimension of $\mathbf{x}_T$ must be the same as the data dimension. This hinders learning a compact representation, making it hard to facilitate downstream inference or attribute manipulation (Sinha et al., 2021). Finally, $\mathbf{x}_T$ is computationally expensive to obtain. To infer $\mathbf{x}_T$ from the data point $\mathbf{x}_0$, it is necessary to numerically solve the ODE in Eq. (3). This results in high time complexity for inferring $\mathbf{x}_T$, which makes it inefficient to exploit latent representations.

To resolve the problem in the latent endpoint $\mathbf{x}_T$, some previous literature, e.g., DiffAE (Preechakul et al., 2022), proposes an auxiliary latent space utilizing a learnable encoder $\mathrm{Enc}_\phi : \mathbb{R}^d \to \mathbb{R}^l$, which maps from data $\mathbf{x}_0$ to an auxiliary latent variable $\mathbf{z}$. Unlike DDIM, these approaches tractably obtain $\mathbf{z}$ from $\mathbf{x}_0$ without solving the ODE, and the encoder can directly learn the latent space in reduced dimensionality. Consequently, the generative ODE

$$\mathrm{d}\mathbf{x}_t = [\mathbf{f}(\mathbf{x}_t, t) - \frac{1}{2}g^2(t)\mathbf{s}_\theta(\mathbf{x}_t, \mathbf{z}, t)]\mathrm{d}t, \tag{4}$$

becomes associated with the $\mathbf{z}$-conditional score function $\mathbf{s}_\theta(\mathbf{x}_t, \mathbf{z}, t)$, which approximates $\nabla_{\mathbf{x}_t} \log q_\phi^t(\mathbf{x}_t|\mathbf{z})$. The generation starts from two distinct latent variables $\mathbf{z}$ and $\mathbf{x}_T$, and defines the conditional probability $p_\theta^{\mathrm{ODE}}(\mathbf{x}_0|\mathbf{z}, \mathbf{x}_T)$. The ODE also provides an encoding from $\mathbf{x}_0$ and $\mathbf{z}$ to $\mathbf{x}_T$, which defines the conditional probability $q_\theta^{\mathrm{ODE}}(\mathbf{x}_T|\mathbf{z}, \mathbf{x}_0)$. However, the auxiliary encoder framework encounters an *information split problem* which this paper raises in Section 3. This paper proposes a method to mitigate this problem.

## 2.3 DIFFUSION PROCESS WITH FIXED ENDPOINTS

To control the information regarding the diffusion endpoint $\mathbf{x}_T$, it is imperative to specify a forward SDE that terminates at the desired endpoint. We employ Doob's $h$-transform (Doob & Doob, 1984), which facilitates the conversion of the original forward SDE in Eq. (1) into

$$\mathrm{d}\mathbf{x}_t = [\mathbf{f}(\mathbf{x}_t, t) + g^2(t)\mathbf{h}(\mathbf{x}_t, t, \mathbf{y}, T)]\mathrm{d}t + g(t)\mathrm{d}\mathbf{w}_t, \quad \mathbf{x}_0 \sim q_{\mathrm{data}}(\mathbf{x}_0), \quad \mathbf{x}_T = \mathbf{y}, \tag{5}$$

where $\mathbf{h}(\mathbf{x}_t, t, \mathbf{y}, T) := \nabla_{\mathbf{x}_t} \log \tilde{q}_t(\mathbf{x}_T|\mathbf{x}_t)|_{\mathbf{x}_T=\mathbf{y}}$ is the score function of the perturbation kernel from the original forward SDE, and $\mathbf{y}$ denotes the desired endpoint. Let $q_t(\mathbf{x}_t)$ denote the marginal distribution of Eq. (5) at $t$. It is noteworthy that when both $\mathbf{x}_0$ and $\mathbf{x}_T$ are given, the conditional probability of $\mathbf{x}_t$ becomes identical to that of the original forward SDE, i.e., $q_t(\mathbf{x}_t|\mathbf{x}_T, \mathbf{x}_0) = \tilde{q}_t(\mathbf{x}_t|\mathbf{x}_T, \mathbf{x}_0)$. If the original forward SDE in Eq. (1) is set to be a specific form (e.g., variance preserving SDE (Ho et al., 2020)), then $q_t(\mathbf{x}_t|\mathbf{x}_T, \mathbf{x}_0)$ follows a Gaussian distribution. This means that sampling of $\mathbf{x}_t \sim q_t(\mathbf{x}_t|\mathbf{x}_T, \mathbf{x}_0)$ at any time $t$ is tractable with an exact density function.

Corresponding to the $h$-transformed forward SDE of Eq. (5), there also exist unique reverse-time SDE and ODE (Anderson, 1982; Zhou et al., 2024)

$$d\mathbf{x}_t = [\mathbf{f}(\mathbf{x}_t, t) - g^2(t)\nabla_{\mathbf{x}_t}\log q_t(\mathbf{x}_t|\mathbf{x}_T) + g^2(t)\mathbf{h}(\mathbf{x}_t, t, \mathbf{y}, T)]dt + g(t)d\bar{\mathbf{w}}_t, \mathbf{x}_T = \mathbf{y}, \quad (6)$$

$$d\mathbf{x}_t = [\mathbf{f}(\mathbf{x}_t, t) - \frac{1}{2}g^2(t)\nabla_{\mathbf{x}_t}\log q_t(\mathbf{x}_t|\mathbf{x}_T) + g^2(t)\mathbf{h}(\mathbf{x}_t, t, \mathbf{y}, T)]dt, \quad \mathbf{x}_T = \mathbf{y}, \quad (7)$$

where $q_t(\mathbf{x}_t|\mathbf{x}_T)$ is the conditional probability defined by Eq. (5). To construct the reverse SDE and ODE, it is necessary to estimate $\nabla_{\mathbf{x}_t}\log q_t(\mathbf{x}_t|\mathbf{x}_T) \approx \mathbf{s}_{\boldsymbol{\theta}}(\mathbf{x}_t, t, \mathbf{x}_T)$ through a neural network with a score matching objective (Zhou et al., 2024)

$$\frac{1}{2}\int_0^T \mathbb{E}_{q_t(\mathbf{x}_t, \mathbf{x}_T)}[g^2(t)||\mathbf{s}_{\boldsymbol{\theta}}(\mathbf{x}_t, t, \mathbf{x}_T) - \nabla_{\mathbf{x}_t}\log q_t(\mathbf{x}_t|\mathbf{x}_T)||_2^2]dt. \quad (8)$$

## 3 MOTIVATION: INFORMATION SPLIT PROBLEM

This paper raises a problem in diffusion-based representation learning with auxiliary encoders (Preechakul et al., 2022; Zhang et al., 2022; Wang et al., 2023; Yang et al., 2023; Yue et al., 2024; Wu & Zheng, 2024) introduced in Section 2.2. The latent variable $\mathbf{z}$ from the encoder has benefits compared to the latent endpoint $\mathbf{x}_T$, but the auxiliary encoder framework encounters an *information split problem*: the information of the data is split into two latent variables $\mathbf{z}$ and $\mathbf{x}_T$. The generative process in Eq. (4) initiates with two latent variables $\mathbf{z}$ and $\mathbf{x}_T$. If the framework only relies on the tractably inferred latent variable $\mathbf{z}$, the reconstruction outcomes depicted in Figure 1b appear to fluctuate depending on the choice of $\mathbf{x}_T$. This implies that $\mathbf{x}_T$ encompasses crucial information necessary for reconstructing $\mathbf{x}_0$. To represent all the information of $\mathbf{x}_0$, it is necessary to infer $\mathbf{x}_T$ by solving the ODE in Eq. (4) from input $\mathbf{x}_0$ to endpoint $\mathbf{x}_T$, enduring its computational costs. Consequently, the persisting issue within the latent variable $\mathbf{x}_T$ remains unresolved in this framework.

To learn an informative latent representation, the mutual information between the data and the latent variable needs to be maximized (Alemi et al., 2018). The *information split problem* hinders the maximization of the mutual information between the data $\mathbf{x}_0$ and the latent variable $\mathbf{z}$. The variational lower bound of the mutual information in the auxiliary encoder framework is

$$\mathbb{E}_{q_{\text{data}}(\mathbf{x}_0), q_{\boldsymbol{\phi}}(\mathbf{z}|\mathbf{x}_0)}[-CE(q_{\boldsymbol{\theta}}^{\text{ODE}}(\mathbf{x}_T|\mathbf{z}, \mathbf{x}_0)||p_{\text{prior}}(\mathbf{x}_T))] + H \leq MI(\mathbf{x}_0, \mathbf{z}), \quad (9)$$

where $MI(\mathbf{x}_0, \mathbf{z}) := \mathbb{E}_{q_{\boldsymbol{\phi}}(\mathbf{x}_0, \mathbf{z})}[\log\frac{q_{\boldsymbol{\phi}}(\mathbf{x}_0, \mathbf{z})}{q_{\text{data}}(\mathbf{x}_0)q_{\boldsymbol{\phi}}(\mathbf{z})}]$ represents the mutual information, $H := \mathcal{H}(q_{\text{data}}(\mathbf{x}_0))$ denotes the data entropy, and $CE(q_{\boldsymbol{\theta}}^{\text{ODE}}(\mathbf{x}_T|\mathbf{z}, \mathbf{x}_0)||p_{\text{prior}}(\mathbf{x}_T)) := \mathbb{E}_{q_{\boldsymbol{\theta}}^{\text{ODE}}(\mathbf{x}_T|\mathbf{z}, \mathbf{x}_0)}[-\log p_{\text{prior}}(\mathbf{x}_T)]$ is the cross-entropy. The cross-entropy term increases as the discrepancy between $q_{\boldsymbol{\theta}}^{\text{ODE}}(\mathbf{x}_T|\mathbf{z}, \mathbf{x}_0)$ and $p_{\text{prior}}(\mathbf{x}_T)$ increases, resulting in a looser lower bound on the mutual information. Since $q_{\boldsymbol{\theta}}^{\text{ODE}}(\mathbf{x}_T|\mathbf{z}, \mathbf{x}_0)$ inherently forms a Dirac delta distribution due to the nature of ODEs, the discrepancy between $q_{\boldsymbol{\theta}}^{\text{ODE}}(\mathbf{x}_T|\mathbf{z}, \mathbf{x}_0)$ and $p_{\text{prior}}(\mathbf{x}_T)$ is inevitable in this framework. For more details, please refer to Appendix A.4.1.

## 4 METHOD: DIFFUSION BRIDGE AUTOENCODERS

To resolve the *information split problem* in auxiliary encoder models, we introduce Diffusion Bridge AutoEncoders (DBAE) featuring $\mathbf{z}$-dependent endpoint $\mathbf{x}_T$ inference using a single network propagation. The endpoint $\mathbf{x}_T$ in DBAE only depends on $\mathbf{z}$, making $\mathbf{z}$ an information bottleneck. Figure 2 illustrates the overall schematic for DBAE. Section 4.1 explains the latent variable inference with the encoder-decoder structure and a learnable forward SDE utilizing Doob's $h$-transform. Section 4.2 delineates the generative process from the information bottleneck $\mathbf{z}$ to data $\mathbf{x}_0$. Section 4.3 analyzes the benefit of DBAE for mutual information maximization between $\mathbf{x}_0$ and $\mathbf{z}$. Section 4.4 elaborates on the objective function for reconstruction, unconditional generation, and its theoretical justifications.

### 4.1 ENCODING FROM $\mathbf{x}_0$ TO $\mathbf{x}_T$ CONDITIONED ON $\mathbf{z}$

We can access i.i.d. samples from $q_{\text{data}}(\mathbf{x}_0)$. The encoder $\text{Enc}_{\boldsymbol{\phi}}: \mathbb{R}^d \to \mathbb{R}^l$ maps data $\mathbf{x}_0$ to the latent variable $\mathbf{z}$, defining the conditional probability $q_{\boldsymbol{\phi}}(\mathbf{z}|\mathbf{x}_0)$. To condense the high-level representation of

Parametrized Forward Process ($\boldsymbol{\phi}, \boldsymbol{\psi}$)

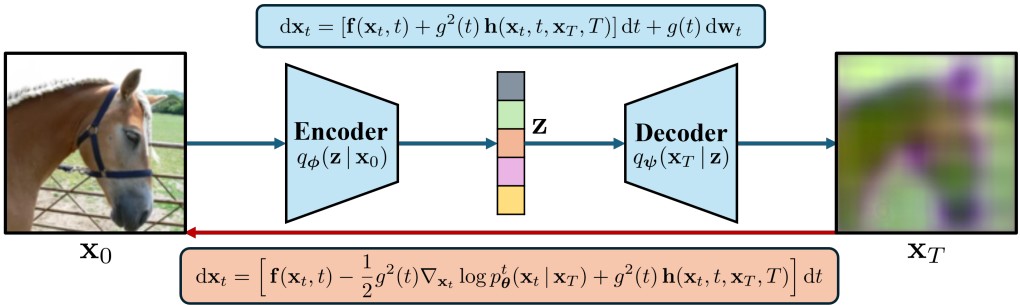

Reverse Process ($\boldsymbol{\theta}$)

Figure 2: A schematic for Diffusion Bridge AutoEncoders. The blue line shows the latent variable inference. DBAE infers the $\mathbf{z}$-dependent endpoint $\mathbf{x}_T$ to make $\mathbf{x}_T$ tractable and to establish $\mathbf{z}$ as an information bottleneck. The paired $\mathbf{x}_0$ and $\mathbf{x}_T$ define a new forward SDE utilizing the $h$-transform. The decoder and the red line show the generative process. The generation starts from the bottleneck latent variable $\mathbf{z}$ and decodes it to the endpoint $\mathbf{x}_T$. The reverse process generates $\mathbf{x}_0$ from $\mathbf{x}_T$.

$\mathbf{x}_0$, the latent dimension $l$ is set to be lower than the data dimension $d$. The decoder $\text{Dec}_\psi : \mathbb{R}^l \to \mathbb{R}^d$ maps from the latent variable $\mathbf{z}$ to the endpoint $\mathbf{x}_T$, defining the conditional probability $q_\psi(\mathbf{x}_T|\mathbf{z})$. The encoder and decoder can be deterministic (i.e., Dirac delta distribution) or stochastic (i.e., Gaussian distribution) depending on the experimental choice. Since the decoder generates the endpoint $\mathbf{x}_T$ solely based on the latent variable $\mathbf{z}$, $\mathbf{z}$ becomes a bottleneck for all the information in $\mathbf{x}_0$. The encoder-decoder structure provides the endpoint distribution $q_{\phi,\psi}(\mathbf{x}_T|\mathbf{x}_0) = \int q_\psi(\mathbf{x}_T|\mathbf{z})q_\phi(\mathbf{z}|\mathbf{x}_0)\mathrm{d}\mathbf{z}$ for a given starting point $\mathbf{x}_0$. We now discuss a new diffusion process $\{\mathbf{x}_t\}_{t=0}^T$ with a given starting point and endpoint pair.

To establish the relationship between the starting point and endpoint given by the encoder-decoder, we utilize Doob's $h$-transform to define a new forward SDE

$$\mathrm{d}\mathbf{x}_t = [\mathbf{f}(\mathbf{x}_t, t) + g^2(t)\mathbf{h}(\mathbf{x}_t, t, \mathbf{x}_T, T)]\mathrm{d}t + g(t)\mathrm{d}\mathbf{w}_t, \ \mathbf{x}_0 \sim q_{\text{data}}(\mathbf{x}_0), \ \mathbf{x}_T \sim q_{\phi,\psi}(\mathbf{x}_T|\mathbf{x}_0), \quad (10)$$

where $\mathbf{h}(\mathbf{x}_t, t, \mathbf{x}_T, T) := \nabla_{\mathbf{x}_t} \log \tilde{q}_t(\mathbf{x}_T|\mathbf{x}_t)$ is the score function of the perturbation kernel in the original forward SDE in Eq. (1). The forward SDE in Eq. (10) determines the distribution of $\mathbf{x}_t$, where $t \in (0, T)$. Let us denote the marginal distribution of Eq. (10) at time $t$ as $q_{\phi,\psi}^t(\mathbf{x}_t)$.

## 4.2 GENERATIVE PROCESS

The generative process begins with the bottleneck latent variable $\mathbf{z}$, which can be inferred from the input data $\mathbf{x}_0$ or is randomly drawn from the prior distribution $p_{\text{prior}}(\mathbf{z})$. The decoder $\text{Dec}_\psi : \mathbb{R}^l \to \mathbb{R}^d$ maps from the latent variable $\mathbf{z}$ to the endpoint $\mathbf{x}_T$ with the probability $p_\psi(\mathbf{x}_T|\mathbf{z})$.[1] Corresponding to a new forward SDE in Eq. (10), there exists a reverse ODE

$$\mathrm{d}\mathbf{x}_t = [\mathbf{f}(\mathbf{x}_t, t) - \frac{1}{2}g^2(t)\nabla_{\mathbf{x}_t} \log q_{\phi,\psi}^t(\mathbf{x}_t|\mathbf{x}_T) + g^2(t)\mathbf{h}(\mathbf{x}_t, t, \mathbf{x}_T, T)]\mathrm{d}t, \quad (11)$$

where the conditional probability $q_{\phi,\psi}^t(\mathbf{x}_t|\mathbf{x}_T)$ is defined by Eq. (10). However, computing the conditional probability $q_{\phi,\psi}^t(\mathbf{x}_t|\mathbf{x}_T)$ is intractable, so we parameterize our score model $\mathbf{s}_\theta(\mathbf{x}_t, t, \mathbf{x}_T) := \nabla_{\mathbf{x}_t} \log p_\theta^t(\mathbf{x}_t|\mathbf{x}_T)$ to approximate $\nabla_{\mathbf{x}_t} \log q_{\phi,\psi}^t(\mathbf{x}_t|\mathbf{x}_T)$. Our parametrized generative process becomes

$$\mathrm{d}\mathbf{x}_t = [\mathbf{f}(\mathbf{x}_t, t) - \frac{1}{2}g^2(t)\nabla_{\mathbf{x}_t} \log p_\theta^t(\mathbf{x}_t|\mathbf{x}_T) + g^2(t)\mathbf{h}(\mathbf{x}_t, t, \mathbf{x}_T, T)]\mathrm{d}t. \quad (12)$$

Stochastic sampling with an SDE is also naturally possible as shown in Section 2.3, but we describe only the ODE for convenience.

---

[1] The two distributions $p_\psi(\mathbf{x}_T|\mathbf{z})$ and $q_\psi(\mathbf{x}_T|\mathbf{z})$ are the same. However, to distinguish between inference and generation, they are respectively denoted as $p$ and $q$.

| **Algorithm 1:** DBAE Training Algorithm for Reconstruction | **Algorithm 2:** Reconstruction |
|---|---|
| **Input:** data distribution $q_{\text{data}}(\mathbf{x}_0)$, drift term $\mathbf{f}$, volatility term $g$ 
 **while** *not converges* **do** 
   Sample time $t$ from $[0, T]$ 
   $\mathbf{x}_0 \sim q_{\text{data}}(\mathbf{x}_0)$, 
   $\mathbf{z} = \text{Enc}_{\boldsymbol{\phi}}(\mathbf{x}_0)$ and $\mathbf{x}_T = \text{Dec}_{\boldsymbol{\psi}}(\mathbf{z})$ 
   $\mathbf{x}_t \sim \tilde{q}_t(\mathbf{x}_t \mid \mathbf{x}_0, \mathbf{x}_T)$ 
   $\mathcal{L}_{\text{AE}} \leftarrow \frac{1}{2} g^2(t) \|\mathbf{s}_{\boldsymbol{\theta}}(\mathbf{x}_t, t, \mathbf{x}_T) - \nabla_{\mathbf{x}_t} \log \tilde{q}_t(\mathbf{x}_t \mid \mathbf{x}_0, \mathbf{x}_T)\|_2^2$ 
   Update $\boldsymbol{\phi}, \boldsymbol{\psi}, \boldsymbol{\theta}$ by $\mathcal{L}_{\text{AE}}$ using the gradient descent method 
 **Output:** $\text{Enc}_{\boldsymbol{\phi}}$, $\text{Dec}_{\boldsymbol{\psi}}$, score network $\mathbf{s}_{\boldsymbol{\theta}}$ | **Input:** $\text{Enc}_{\boldsymbol{\phi}}$, $\text{Dec}_{\boldsymbol{\psi}}$, score network $\mathbf{s}_{\boldsymbol{\theta}}$, 
   sample $\mathbf{x}_0$, discretized time steps 
   $\{t_i\}_{i=0}^N$ 
 $\mathbf{z} = \text{Enc}_{\boldsymbol{\phi}}(\mathbf{x}_0)$ 
 $\mathbf{x}_T = \text{Dec}_{\boldsymbol{\psi}}(\mathbf{z})$ 
 **for** $i = N, ..., 1$ **do** 
   Update $\mathbf{x}_{t_i}$ using Eq. (12) 
 **Output:** Reconstructed sample $\hat{\mathbf{x}}_0$ |

## 4.3 MUTUAL INFORMATION ANALYSIS

From the definition of inference and generation of DBAE in Sections 4.1 and 4.2, the variational lower bound on the mutual information between $\mathbf{x}_0$ and $\mathbf{z}$ is

$$\mathbb{E}_{q_{\boldsymbol{\phi}}(\mathbf{x}_0, \mathbf{z})}[\mathbb{E}_{q_{\boldsymbol{\psi}}(\mathbf{x}_T \mid \mathbf{z})}[\log p_{\boldsymbol{\theta}}(\mathbf{x}_0 \mid \mathbf{x}_T)] - D_{KL}(q_{\boldsymbol{\psi}}(\mathbf{x}_T \mid \mathbf{z}) \| p_{\boldsymbol{\psi}}(\mathbf{x}_T \mid \mathbf{z}))] + H \le MI(\mathbf{x}_0, \mathbf{z}), \quad (13)$$

where $p_{\boldsymbol{\theta}}(\mathbf{x}_0 \mid \mathbf{x}_T)$ is defined by the generative process in Section 4.2. Please see Appendix A.4.2 for a detailed derivation. Here, the term $D_{KL}(q_{\boldsymbol{\psi}}(\mathbf{x}_T \mid \mathbf{z}) \| p_{\boldsymbol{\psi}}(\mathbf{x}_T \mid \mathbf{z}))$ becomes zero because both conditional probabilities of $\mathbf{x}_T$ given $\mathbf{z}$ are the same in the inference and the generation. The remaining term $\mathbb{E}_{q_{\boldsymbol{\phi}}(\mathbf{x}_0, \mathbf{z})}[\mathbb{E}_{q_{\boldsymbol{\psi}}(\mathbf{x}_T \mid \mathbf{z})}[\log p_{\boldsymbol{\theta}}(\mathbf{x}_0 \mid \mathbf{x}_T)]$ can be controlled by the optimization of $\boldsymbol{\phi}, \boldsymbol{\psi}$, and $\boldsymbol{\theta}$. The relation between an objective function and mutual information is declared in Theorem 2.

## 4.4 OBJECTIVE FUNCTION

The objective function bifurcates depending on the specific tasks. The model requires a reconstruction capability for downstream inference, attribute manipulation, and interpolation. To achieve reconstruction capability, the model needs 1) an encoding capability $(\mathbf{x}_0 \to \mathbf{z} \to \mathbf{x}_T)$ and 2) a regeneration capability $(\mathbf{x}_T \to \mathbf{x}_0)$. The encoding process should infer a distinct latent variable for each data point $\mathbf{x}_0$ to ensure that the original information is preserved during reconstruction. The regeneration capability needs to estimate the reverse process by approximating $\mathbf{s}_{\boldsymbol{\theta}}(\mathbf{x}_t, t, \mathbf{x}_T) \approx \nabla_{\mathbf{x}_t} \log q_{\boldsymbol{\phi}, \boldsymbol{\psi}}^t(\mathbf{x}_t \mid \mathbf{x}_T)$. For an unconditional generation, the model must possess the ability to generate random samples from the endpoint $\mathbf{x}_T$, which implies that the generative endpoint distribution $p_{\boldsymbol{\psi}}(\mathbf{x}_T) = \int p_{\boldsymbol{\psi}}(\mathbf{x}_T \mid \mathbf{z}) p_{\text{prior}}(\mathbf{z}) d\mathbf{z}$ should closely match the aggregated inferred distribution $q_{\boldsymbol{\phi}, \boldsymbol{\psi}}(\mathbf{x}_T) = \int q_{\boldsymbol{\psi}}(\mathbf{x}_T \mid \mathbf{z}) q_{\boldsymbol{\phi}}(\mathbf{z} \mid \mathbf{x}_0) q_{\text{data}}(\mathbf{x}_0) d\mathbf{x}_0 d\mathbf{z}$.

### 4.4.1 RECONSTRUCTION

For successful reconstruction, the model needs to fulfill two criteria: 1) encoding the latent variable $\mathbf{x}_T$ uniquely depending on the data point $\mathbf{x}_0$, and 2) regenerating from $\mathbf{x}_T$ to $\mathbf{x}_0$. The inferred latent distribution $q_{\boldsymbol{\phi}, \boldsymbol{\psi}}(\mathbf{x}_T \mid \mathbf{x}_0)$ should provide unique information for each $\mathbf{x}_0$. To achieve this, we aim to minimize the entropy $\mathcal{H}(q_{\boldsymbol{\phi}, \boldsymbol{\psi}}(\mathbf{x}_T \mid \mathbf{x}_0))$ to embed $\mathbf{x}_0$-dependent $\mathbf{x}_T$ with minimum uncertainty. On the other hand, we maximize the entropy $\mathcal{H}(q_{\boldsymbol{\phi}, \boldsymbol{\psi}}(\mathbf{x}_T))$ to embed different $\mathbf{x}_T$ for each $\mathbf{x}_0$. Since the posterior entropy $\mathcal{H}(q_{\boldsymbol{\phi}, \boldsymbol{\psi}}(\mathbf{x}_0 \mid \mathbf{x}_T)) = \mathcal{H}(q_{\boldsymbol{\phi}, \boldsymbol{\psi}}(\mathbf{x}_T \mid \mathbf{x}_0)) - \mathcal{H}(q_{\boldsymbol{\phi}, \boldsymbol{\psi}}(\mathbf{x}_T)) + \mathcal{H}(q_{\text{data}}(\mathbf{x}_0))$ naturally includes the aforementioned terms, we use this term as a regularization. Minimizing the gap between Eqs. (11) and (12) is necessary for regenerating from $\mathbf{x}_T$ to $\mathbf{x}_0$. This requires alignment between the inferred score function $\nabla_{\mathbf{x}_t} \log q_{\boldsymbol{\phi}, \boldsymbol{\psi}}^t(\mathbf{x}_t \mid \mathbf{x}_T)$ and the model score function $\mathbf{s}_{\boldsymbol{\theta}}(\mathbf{x}_t, t, \mathbf{x}_T)$. Similarly to Eq. (8), we propose the score-matching objective function $\mathcal{L}_{\text{SM}}$ described as

$$\mathcal{L}_{\text{SM}} := \frac{1}{2} \int_0^T \mathbb{E}_{q_{\boldsymbol{\phi}, \boldsymbol{\psi}}^t(\mathbf{x}_t, \mathbf{x}_T)}[g^2(t) \| \mathbf{s}_{\boldsymbol{\theta}}(\mathbf{x}_t, t, \mathbf{x}_T) - \nabla_{\mathbf{x}_t} \log q_{\boldsymbol{\phi}, \boldsymbol{\psi}}^t(\mathbf{x}_t \mid \mathbf{x}_T)\|_2^2] dt. \quad (14)$$

We train DBAE with the entropy-regularized score matching objective $\mathcal{L}_{\text{AE}}$ described as

$$\mathcal{L}_{\text{AE}} := \mathcal{L}_{\text{SM}} + \mathcal{H}(q_{\boldsymbol{\phi}, \boldsymbol{\psi}}(\mathbf{x}_0 \mid \mathbf{x}_T)). \quad (15)$$

The detailed training and testing procedures are outlined in algorithms 1 and 2, respectively. Theorem 1 demonstrates that the entropy-regularized score matching objective in $\mathcal{L}_{\text{AE}}$ becomes a tractable form of objective, and it is equivalent to the reconstruction formulation. The inference distribution $q_{\boldsymbol{\phi}, \boldsymbol{\psi}}(\mathbf{x}_t, \mathbf{x}_T \mid \mathbf{x}_0)$ is optimized to provide the best information about $\mathbf{x}_0$ for easy reconstruction.

**Theorem 1.** *For the objective function $\mathcal{L}_{AE}$, the following equality holds.*

$$\mathcal{L}_{AE} = \frac{1}{2} \int_0^T \mathbb{E}_{q_{\boldsymbol{\phi}, \boldsymbol{\psi}}^t(\mathbf{x}_0, \mathbf{x}_t, \mathbf{x}_T)}[g^2(t) \| \mathbf{s}_{\boldsymbol{\theta}}(\mathbf{x}_t, t, \mathbf{x}_T) - \nabla_{\mathbf{x}_t} \log \tilde{q}_t(\mathbf{x}_t \mid \mathbf{x}_0, \mathbf{x}_T)\|_2^2] dt \quad (16)$$

*Moreover, if Eq. (1) is a linear SDE.[2], there exists $\alpha(t)$, $\beta(t)$, $\gamma(t)$, $\lambda(t)$, such that*

$$\mathcal{L}_{AE} = \frac{1}{2} \int_0^T \mathbb{E}_{q^t_{\phi,\psi}(\mathbf{x}_0,\mathbf{x}_t,\mathbf{x}_T)}[\lambda(t)||\mathbf{x}^0_{\boldsymbol{\theta}}(\mathbf{x}_t,t,\mathbf{x}_T) - \mathbf{x}_0||^2_2]dt, \tag{17}$$

*where* $\mathbf{x}^0_{\boldsymbol{\theta}}(\mathbf{x}_t,t,\mathbf{x}_T) := \alpha(t)\mathbf{x}_t + \beta(t)\mathbf{x}_T + \gamma(t)\mathbf{s}_{\boldsymbol{\theta}}(\mathbf{x}_t,t,\mathbf{x}_T)$, *and* $q_{\phi,\psi}(\mathbf{x}_0,\mathbf{x}_t,\mathbf{x}_T) = \int q_{data}(\mathbf{x}_0)q_\phi(\mathbf{z}|\mathbf{x}_0)q_\psi(\mathbf{x}_T|\mathbf{z})q_t(\mathbf{x}_t|\mathbf{x}_T,\mathbf{x}_0)d\mathbf{z}$, *following the graphical model in Fig. 1c.*

The assumptions and proof of Theorem 1 are in Appendix A.1. Moreover, Theorem 2 shows the objective functions $\mathcal{L}_{AE}$ is the upper bound of the negative mutual information between $\mathbf{x}_0$ and $\mathbf{z}$ up to a constant. Since the optimization direction of $\mathcal{L}_{AE}$ is aligned with maximizing the mutual information, our objective function makes the mutual information higher, which can make $\mathbf{z}$ informative. The proof of Theorem 2 is in Appendix A.5.

**Theorem 2.** $-MI(\mathbf{x}_0, \mathbf{z}) \leq \mathcal{L}_{AE} - H$, *where* $H = \mathcal{H}(q_{data}(\mathbf{x}_0))$ *is a constant w.r.t.* $\phi, \psi, \boldsymbol{\theta}$.

### 4.4.2 GENERATIVE MODELING

In Section 4.4.1, the discussion focused on the objective function for reconstruction. The distribution of $\mathbf{x}_T$ should be considered for generative modeling. This section addresses the discrepancy between the inferred distribution $q_{\phi,\psi}(\mathbf{x}_T)$ and the generative prior distribution $p_\psi(\mathbf{x}_T)$. To address this, we propose the objective $\mathcal{L}_{PR}$ related to the generative prior.

$$\mathcal{L}_{PR} := \mathbb{E}_{q_{data}(\mathbf{x}_0)}[D_{KL}(q_{\phi,\psi}(\mathbf{x}_T|\mathbf{x}_0)||p_\psi(\mathbf{x}_T))] \tag{18}$$

Theorem 3 demonstrates that the autoencoding objective $\mathcal{L}_{AE}$ and prior objective $\mathcal{L}_{PR}$ bound the Kullback-Leibler divergence between data distribution $q_{data}(\mathbf{x}_0)$ and the generative model distribution $p_{\psi,\boldsymbol{\theta}}(\mathbf{x}_0) = \int p_{\boldsymbol{\theta}}(\mathbf{x}_0|\mathbf{x}_T)p_\psi(\mathbf{x}_T|\mathbf{z})p_{prior}(\mathbf{z})d\mathbf{z}d\mathbf{x}_T$ up to a constant. The proof is in Appendix A.2.

**Theorem 3.** $D_{KL}(q_{data}(\mathbf{x}_0)||p_{\psi,\boldsymbol{\theta}}(\mathbf{x}_0)) \leq \mathcal{L}_{AE} + \mathcal{L}_{PR} - H$, *where* $H = \mathcal{H}(q_{data}(\mathbf{x}_0))$ *is a constant w.r.t.* $\phi, \psi, \boldsymbol{\theta}$.

For generative modeling, we separately minimize the terms $\mathcal{L}_{AE}$ and $\mathcal{L}_{PR}$, following (Esser et al., 2021; Preechakul et al., 2022; Zhang et al., 2022). The separate training of the generative prior distribution with a powerful generative model effectively reduces the mismatch between the prior and the aggregated posterior (Sinha et al., 2021; Aneja et al., 2021). Initially, we optimize $\mathcal{L}_{AE}$ with respect to the parameters of encoder ($\phi$), decoder ($\psi$), and score network ($\boldsymbol{\theta}$), and fix the parameters $\boldsymbol{\theta}, \phi, \psi$. Subsequently, we newly parameterize the generative prior $p_{prior}(\mathbf{z}) := p_{\boldsymbol{\omega}}(\mathbf{z})$ using a shallow latent diffusion models, and optimize $\mathcal{L}_{PR}$ w.r.t $\boldsymbol{\omega}$. See Appendix A.3 for further details.

## 5 EXPERIMENT

This section empirically validates the effectiveness of the intended design of the proposed model, DBAE. We utilize the U-Net architecture for the score network ($\boldsymbol{\theta}$), as shown in Fig. 7b. Since our score network needs to account for the additional input $\mathbf{x}_T$, we concatenate $\mathbf{x}_t$ and $\mathbf{x}_T$ as the U-Net input. We employ half of the U-Net architecture as the encoder ($\phi$) and use a CNN-based upsampler as the decoder ($\psi$), adopted from (Liu et al., 2021). The encoder and decoder architectures are detailed in Fig. 7a. To compare DBAE with previous diffusion-based representation learning approaches, we adopt the remaining experimental configurations (e.g., batch size, learning rate) from DiffAE (Preechakul et al., 2022) as closely as possible. Detailed experimental configurations are provided in Appendix C. We evaluate both latent encoding capability and generation quality across various tasks. We quantitatively assess the performance of downstream inference, reconstruction, disentanglement, and unconditional generation. Additionally, we qualitatively demonstrate interpolation and attribute manipulation capabilities. Finally, we conduct experiments with two variations of the proposed model's encoder: 1) a Gaussian stochastic encoder (DBAE) and 2) a deterministic encoder (DBAE-d) for ablation studies. We use a deterministic structure for the decoder.

### 5.1 DOWNSTREAM INFERENCE

To examine the learned latent representation capability of $\text{Enc}_\phi$, we perform a linear-probe attribute prediction following DiTi (Yue et al., 2024). We train a linear classifier with parameters $(\mathbf{w}, b)$ using

---

[2]Eq. (1) is a linear SDE when the drift function $\mathbf{f}$ is linear with respect to $\mathbf{x}_t$.

Table 1: Linear-probe attribute prediction quality comparison for models trained on CelebA and FFHQ with $\dim(\mathbf{z}) = 512$. 'Gen' indicates the generation capability. The best and second-best results are highlighted in **bold** and underline, respectively. We evaluate 5 times and report the average.

| Method | Gen | CelebA | | | FFHQ | | |
|---|---|---|---|---|---|---|---|
| | | AP (↑) | Pearson's r (↑) | MSE (↓) | AP (↑) | Pearson's r (↑) | MSE (↓) |
| SimCLR (Chen et al., 2020) | ✗ | 0.597 | 0.474 | 0.603 | 0.608 | 0.481 | 0.638 |
| $\beta$-TCVAE (Chen et al., 2018) | ✓ | 0.450 | 0.378 | 0.573 | 0.432 | 0.335 | 0.608 |
| IB-GAN (Jeon et al., 2021) | ✓ | 0.442 | 0.307 | 0.597 | 0.428 | 0.260 | 0.644 |
| DiffAE (Preechakul et al., 2022) | ✓ | 0.603 | 0.598 | 0.421 | 0.605 | 0.606 | 0.410 |
| PDAE (Zhang et al., 2022) | ✓ | 0.602 | 0.596 | 0.410 | 0.597 | 0.603 | 0.416 |
| DiTi (Yue et al., 2024) | ✓ | 0.623 | 0.617 | 0.392 | 0.614 | 0.622 | 0.384 |
| DBAE-d | ✓ | 0.650 | 0.635 | 0.413 | 0.656 | 0.638 | 0.404 |
| DBAE | ✓ | **0.655** | **0.643** | **0.369** | **0.664** | **0.675** | **0.332** |

data-attribute pairs $(\mathbf{x}_0, y)$. The attribute prediction $\hat{y} = \mathbf{w}^T \mathbf{z} + b$ is based on the learned latent representation $\mathbf{z} = \text{Enc}_\phi(\mathbf{x}_0)$, which is fitted to predict the ground-truth label $y$. An informative latent representation allows the linear classifier to predict the ground-truth label $y$ more effectively. We evaluate $\text{Enc}_\phi(\mathbf{x}_0)$ trained on CelebA (Liu et al., 2015) and FFHQ (Karras et al., 2019). We train a linear classifier on 1) CelebA with 40 binary labels, measuring accuracy as AP, and 2) LFW (Kumar et al., 2009) for attribute regression, measuring accuracy using Pearson's $r$ and MSE. Table 1 shows that DBAE outperforms other diffusion-based representation learning baselines. Since DiffAE, PDAE, and DiTi suffer from the *information split problem*, they produce a $\mathbf{z}$ that is less informative than DBAE. Figure 3 presents statistics for 100 reconstructions of the same image with inferred $\mathbf{z}$. Because PDAE's reconstruction varies depending on the selection of $\mathbf{x}_T$, it suggests that intricate details, such as hair and facial features, are contained in $\mathbf{x}_T$, which $\mathbf{z}$ fails to capture. This observation aligns with Figure 8, where significant performance gains are observed for attributes related to facial details, such as shadows and hair. A comparison between DBAE-d and DBAE reveals that the stochastic encoder performs slightly better. We conjecture that the stochastic encoder leverages a broader latent space, which benefits discriminative downstream inference.

## 5.2 RECONSTRUCTION

Table 2: Autoencoding reconstruction quality comparison. Among tractable and 512-dimensional latent variable models, the one yielding the best performance is highlighted in **bold**, underline for the next best performer.

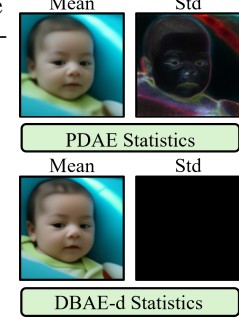

| Method | Tractability | Latent dim (↓) | SSIM (↑) | LPIPS (↓) | MSE (↓) |
|---|---|---|---|---|---|
| StyleGAN2 ($\mathcal{W}$) (Karras et al., 2020) | ✗ | 512 | 0.677 | 0.168 | 0.016 |
| StyleGAN2 ($\mathcal{W}+$) (Abdal et al., 2019) | ✗ | 7,168 | 0.827 | 0.114 | 0.006 |
| VQ-GAN (Esser et al., 2021) | ✓ | 65,536 | 0.782 | 0.109 | 3.61e-3 |
| VQ-VAE2 (Razavi et al., 2019) | ✓ | 327,680 | 0.947 | 0.012 | 4.87e-4 |
| NVAE (Vahdat & Kautz, 2020) | ✓ | 6,005,760 | 0.984 | 0.001 | 4.85e-5 |
| DDIM (Inferred $\mathbf{x}_T$) (Song et al., 2021a) | ✗ | 49,152 | 0.917 | 0.063 | 0.002 |
| DiffAE (Inferred $\mathbf{x}_T$) (Preechakul et al., 2022) | ✗ | 49,664 | 0.991 | 0.011 | 6.07e-5 |
| PDAE (Inferred $\mathbf{x}_T$) (Zhang et al., 2022) | ✗ | 49,664 | 0.994 | 0.007 | 3.84e-5 |
| DiffAE (Random $\mathbf{x}_T$) (Preechakul et al., 2022) | ✓ | 512 | 0.677 | 0.073 | 0.007 |
| PDAE (Random $\mathbf{x}_T$) (Zhang et al., 2022) | ✓ | 512 | 0.689 | 0.098 | 5.01e-3 |
| DBAE | ✓ | 512 | 0.920 | 0.094 | 4.81e-3 |
| DBAE-d | ✓ | 512 | **0.953** | **0.072** | **2.49e-3** |

Figure 3: Reconstruction w/ inferred $\mathbf{z}$.

We examine the reconstruction quality following DiffAE (Preechakul et al., 2022) to quantify information loss in the latent variable. For a test sample $\mathbf{x}_0$, the procedure in algorithm 2 provides a reconstructed sample $\hat{\mathbf{x}}_0$. The reconstruction error is the distance $d(\mathbf{x}_0, \hat{\mathbf{x}}_0)$, where the distance function can be SSIM (Wang et al., 2003), LPIPS (Zhang et al., 2018), or MSE. Table 2 reports the averaged reconstruction error over the test dataset $\mathbb{E}_{p_{\text{test}}(\mathbf{x}_0)}[d(\mathbf{x}_0, \hat{\mathbf{x}}_0)]$. We trained DBAE on FFHQ and evaluated it on CelebA-HQ (Karras et al., 2018). Tractability refers to the ability to perform inference on latent variables without repeated neural network evaluations. Tractability is crucial for regularizing the latent variable to achieve specific goals (e.g., disentanglement) during the training phase. The latent dimension refers to the dimension of the bottleneck latent variable during inference. A lower dimension is advantageous for applications such as downstream inference or attribute manipulation. The third block in Table 2 compares performance under the same qualitative conditions. DBAE-d exhibits performance that surpasses both DiffAE and PDAE. Naturally, DiffAE and PDAE exhibit worse performance because the information is split between $\mathbf{x}_T$ and $\mathbf{z}$. Unlike the downstream inference experiments in Section 5.1, the deterministic encoder performs better.

## 5.3 DISENTANGLEMENT

Table 3: Disentanglment and sample quality comparisons on CelebA.

| Method | Reg $\mathbf{z}$ | TAD ($\uparrow$) | ATTRS ($\uparrow$) | FID ($\downarrow$) |
|---|---|---|---|---|
| AE | ✗ | 0.042±0.004 | 1.0±0.0 | 90.4±1.8 |
| DiffAE (Preechakul et al., 2022) | ✗ | **0.155**±0.010 | 2.0±0.0 | 22.7±2.1 |
| DBAE | ✗ | 0.124±0.078 | **2.2**±1.3 | **11.8**±0.2 |
| VAE (Kingma & Welling, 2014) | ✓ | 0.000±0.000 | 0.0±0.0 | 94.3±2.8 |
| $\beta$-VAE (Higgins et al., 2017) | ✓ | 0.088±0.051 | 1.6±0.8 | 99.8±2.4 |
| InfoVAE (Zhao et al., 2019) | ✓ | 0.000±0.000 | 0.0±0.0 | 77.8±1.6 |
| InfoDiffusion (Wang et al., 2023) | ✓ | 0.299±0.006 | 3.0±0.0 | 22.3±1.2 |
| DisDiff (Yang et al., 2023) | ✓ | 0.305±0.010 | - | 18.3±2.1 |
| DBAE+TC | ✓ | **0.362**±0.036 | **3.8**±0.8 | 13.4±0.2 |

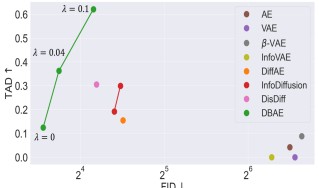

Figure 4: TAD-FID tradeoffs compared to the baselines.

Unsupervised disentanglement of the latent variable $\mathbf{z}$ is an important application of generative representation learning, as it enables controllable generation without supervision. The goal of disentanglement is to ensure that each dimension of the latent variable captures distinct information. To achieve this, we apply regularization to minimize total correlation (TC), i.e., $D_{\mathrm{KL}}(q_\phi(\mathbf{z})||\Pi_{i=1}^{l}q_\phi(\mathbf{z}_i))$, adopted from (Chen et al., 2018). TC regularization decouples the correlation between the dimensions of $\mathbf{z}$, allowing different information to be captured in each dimension. Following InfoDiffusion (Wang et al., 2023), we measure TAD and ATTRS (Yeats et al., 2022) to quantify disentanglement in $\mathbf{z}$. Since sample quality and disentanglement often involve a trade-off, we also measure FID (Heusel et al., 2017) between 10k samples. Table 3 shows the performance comparison, where DBAE outperforms all the baselines. Figure 4 demonstrates the effects of coefficients on TC regularization, showing that DBAE envelops all the baselines. To disentangle information, a well-encoded representation must first be achieved. The informative representation capability of DBAE supports this application.

## 5.4 UNCONDITIONAL GENERATION

Table 4: Unconditional generation on FFHQ. '+AE' indicates the use of the inferred distribution $q_\phi(\mathbf{z})$ instead of $p_\omega(\mathbf{z})$.

| Method | Prec ($\uparrow$) | IS ($\uparrow$) | FID 50k ($\downarrow$) | Rec ($\uparrow$) |
|---|---|---|---|---|
| DDIM (Song et al., 2021a) | 0.697 | 3.14 | 11.27 | 0.451 |
| DDPM (Ho et al., 2020) | 0.768 | 3.11 | **9.14** | 0.335 |
| DiffAE (Preechakul et al., 2022) | 0.762 | 2.98 | 9.40 | **0.458** |
| PDAE (Zhang et al., 2022) | 0.695 | 2.23 | 47.42 | 0.153 |
| DBAE | **0.780** | **3.87** | 11.25 | 0.392 |
| DiffAE+AE | 0.750 | **3.63** | 2.84 | 0.685 |
| PDAE+AE | 0.709 | 3.55 | 7.42 | 0.602 |
| DBAE+AE | **0.751** | 3.57 | **1.77** | **0.687** |

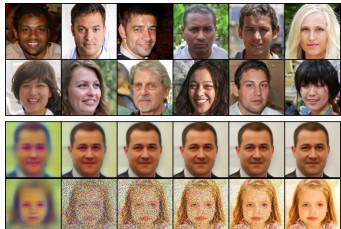

Figure 5: Top two rows: uncurated samples. Bottom two rows: the sampling trajectory with ODE and SDE.

To generate a sample unconditionally, the generation starts from the learned prior distribution $\mathbf{z} \sim p_\omega(\mathbf{z})$. The latent variable $\mathbf{z}$ is decoded into $\mathbf{x}_T = \mathrm{Dec}_\psi(\mathbf{z})$, and the sample $\mathbf{x}_0$ is finally obtained through the generative process described in Eq. (12). For CelebA, a comparison with DiffAE in Table 3 shows that DBAE surpasses DiffAE by a large margin in FID (Heusel et al., 2017) (22.7 vs. 11.8). Table 4 shows the performance on FFHQ, which is known to be more diverse than CelebA. DBAE still performs the best among the baselines in terms of Precision (Kynkäänniemi et al., 2019) and Inception Score (IS) (Salimans et al., 2016), both of which are highly influenced by image fidelity. However, DBAE shows slightly worse FID (Heusel et al., 2017) and Recall (Kynkäänniemi et al., 2019), which are more affected by sample diversity. To analyze this, we alter the learned generative prior $p_\omega(\mathbf{z})$ to the inferred distribution $q_\phi(\mathbf{z})$ as shown in the second block of Table 4. In this autoencoding case, DBAE captures both image fidelity and diversity. We speculate that it is more sensitive to the gap between $q_\phi(\mathbf{z})$ and $p_\omega(\mathbf{z})$ since the information depends solely on $\mathbf{z}$, not on the joint condition of $\mathbf{x}_T$ and $\mathbf{z}$. A complex generative prior model $\omega$ could potentially solve this issue (Esser et al., 2021; Vahdat et al., 2021). Figure 5 shows the randomly generated samples and sampling trajectories on FFHQ from DBAE.

## 5.5 INTERPOLATION

For the two images $\mathbf{x}_0^1$ and $\mathbf{x}_0^2$, DBAE can mix the styles by exploring the intermediate points in the latent space. We encode images into $\mathbf{z}^1 = \mathrm{Enc}_\phi(\mathbf{x}_0^1)$ and $\mathbf{z}^2 = \mathrm{Enc}_\phi(\mathbf{x}_0^2)$. We then regenerate

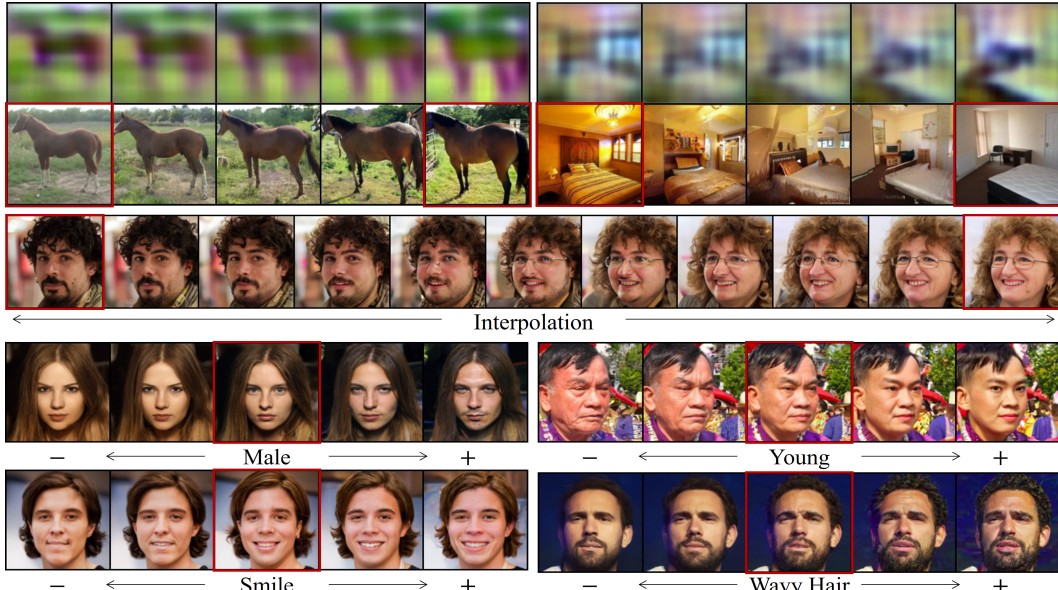

Figure 6: Interpolation (top) and attribute manipulation (bottom) with DBAE. (Red box: input image)

from $\mathbf{z}^\lambda = \lambda \mathbf{z}^1 + (1-\lambda)\mathbf{z}^2$ to data $\mathbf{x}_0$ using the generative process specified in Eq. (12). The unique properties of DBAE offer distinct benefits here: 1) DiffAE (Preechakul et al., 2022) and PDAE (Zhang et al., 2022) need to infer $\mathbf{x}_T^1$, $\mathbf{x}_T^2$ by solving the ODE in Eq. (4) with hundreds of score function evaluations (Preechakul et al., 2022; Zhang et al., 2022). They then geometrically interpolate between $\mathbf{x}_T^1$ and $\mathbf{x}_T^2$ to obtain $\mathbf{x}_T^\lambda$, regardless of the correspondence between $\mathbf{z}^\lambda$ and $\mathbf{x}_T^\lambda$. 2) DBAE directly obtains an intermediate value of $\mathbf{x}_T^\lambda = \text{Dec}_\psi(\mathbf{z}^\lambda)$. This does not require solving the ODE, and the correspondence between $\mathbf{x}_T^\lambda$ and $\mathbf{z}^\lambda$ is also naturally determined by the decoder ($\psi$). Figure 6 shows the interpolation results on the LSUN Horse, Bedroom (Yu et al., 2015) and FFHQ datasets. The top row shows the corresponding endpoints $\mathbf{x}_T^\lambda$ in the interpolation, which changes smoothly between $\mathbf{x}_T^1$ and $\mathbf{x}_T^2$. The bottom row shows the interpolation results on FFHQ, which smoothly changes semantic information such as gender, glasses, and hair color.

## 5.6 ATTRIBUTE MANIPULATION

The linear classifier used in Section 5.1 can also be utilized to identify the manipulation direction of $\mathbf{z}$. From the prediction of a linear classifier $\hat{y} = \mathbf{w}^T \mathbf{z} + b$, traversing in the direction $\frac{dy}{d\mathbf{z}} = \mathbf{w}$ increases or decreases the logit. For a image $\mathbf{x}_0$, this is encoded as $\mathbf{z} = \text{Enc}_\phi(\mathbf{x}_0)$. The encoded representation $\mathbf{z}$ is manipulated as $\mathbf{z}^{\text{new}} = \mathbf{z} + \lambda \mathbf{w}$. The manipulated image $\mathbf{x}_0^{\text{new}}$ is obtained by decoding $\mathbf{x}_T^{\text{new}} = \text{Dec}_\psi(\mathbf{z}^{\text{new}})$, and the reverse process in Eq. (12). DiffAE and PDAE additionally infer from $\mathbf{x}_0$ to $\mathbf{x}_T$ by solving Eq. (4) with hundreds of score function evaluations, fixing $\mathbf{x}_T$ to prevent undesirable variations in $\mathbf{x}_T$. Table 8 describes the long inference time for $\mathbf{x}_T$ in previous approaches. Moreover, if some information is split into $\mathbf{x}_T$, these methods cannot handle this information. On the other hand, DBAE infers $\mathbf{x}_T$ directly from manipulated $\mathbf{z}$, ensuring that the endpoint $\mathbf{x}_T$ is also controlled through the decoder ($\psi$). Figure 6 shows the manipulation results for both CelebA-HQ images and FFHQ images with various attributes.

## 6 CONCLUSION

This paper identifies the *information split problem* in diffusion-based representation learning, stemming from separate inferences of the forward process and the auxiliary encoder. This issue hinders the representation capabilities of the tractable latent variable $\mathbf{z}$. The proposed method, Diffusion Bridge AutoEncoders, systematically addresses these challenges by constructing $\mathbf{z}$-dependent endpoint $\mathbf{x}_T$ inference. By transforming $\mathbf{z}$ into an information bottleneck, DBAE extracts more meaningful representations within the tractable latent space. The notable enhancements in the latent quality of DBAE improve downstream inference and image manipulation applications. This work lays a solid foundation for further exploration of effective representation in learnable diffusion inference.

ACKNOWLEDGMENTS

This work was supported by Institute of Information & Communication Technology Planning & Evaluation (IITP) grant funded by the Korea government (MSIT) (No. RS-2024-00361319, Development of technology to generate synthetic data in exceptional situations and advance artificial intelligence prediction models). This research was supported in part by the NAVER-Intel Co-Lab. The work was conducted by KAIST and reviewed by both NAVER and Intel.

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

CONTENTS

# A   PROOFS AND MATHEMATICAL EXPLANATIONS

In this section, we follow the assumptions from Appendix A in (Song et al., 2021b), and we also assume that both $\mathbf{s}_{\boldsymbol{\theta}}$ and $q_{\phi,\psi}^t$ have continuous second-order derivatives and finite second moments, which are the same assumptions of Theorems 2 and 4 in (Song et al., 2021b).

## A.1   PROOF OF THEOREM 1

**Theorem 1.** *For the objective function $\mathcal{L}_{AE}$, the following equality holds.*

$$\mathcal{L}_{AE} = \frac{1}{2}\int_0^T \mathbb{E}_{q_{\phi,\psi}^t(\mathbf{x}_0,\mathbf{x}_t,\mathbf{x}_T)}[g^2(t)||\mathbf{s}_{\boldsymbol{\theta}}(\mathbf{x}_t,t,\mathbf{x}_T) - \nabla_{\mathbf{x}_t}\log\tilde{q}_t(\mathbf{x}_t|\mathbf{x}_0,\mathbf{x}_T)||_2^2]dt \qquad (16)$$

*Moreover, if Eq. (1) is a linear SDE.[3], there exists $\alpha(t)$, $\beta(t)$, $\gamma(t)$, $\lambda(t)$, such that*

$$\mathcal{L}_{AE} = \frac{1}{2}\int_0^T \mathbb{E}_{q_{\phi,\psi}^t(\mathbf{x}_0,\mathbf{x}_t,\mathbf{x}_T)}[\lambda(t)||\mathbf{x}_{\boldsymbol{\theta}}^0(\mathbf{x}_t,t,\mathbf{x}_T) - \mathbf{x}_0||_2^2]dt, \qquad (17)$$

*where* $\mathbf{x}_{\boldsymbol{\theta}}^0(\mathbf{x}_t,t,\mathbf{x}_T) := \alpha(t)\mathbf{x}_t + \beta(t)\mathbf{x}_T + \gamma(t)\mathbf{s}_{\boldsymbol{\theta}}(\mathbf{x}_t,t,\mathbf{x}_T)$, *and* $q_{\phi,\psi}^t(\mathbf{x}_0,\mathbf{x}_t,\mathbf{x}_T) = \int q_{data}(\mathbf{x}_0)q_{\phi}(\mathbf{z}|\mathbf{x}_0)q_{\psi}(\mathbf{x}_T|\mathbf{z})q_t(\mathbf{x}_t|\mathbf{x}_T,\mathbf{x}_0)d\mathbf{z}$, *following the graphical model in Fig. 1c.*

*Proof.* Note that the definitions of the objective functions are

$$\mathcal{L}_{\text{SM}} := \frac{1}{2}\int_0^T \mathbb{E}_{q_{\phi,\psi}^t(\mathbf{x}_t,\mathbf{x}_T)}[g^2(t)||\mathbf{s}_{\boldsymbol{\theta}}(\mathbf{x}_t,t,\mathbf{x}_T) - \nabla_{\mathbf{x}_t}\log q_{\phi,\psi}^t(\mathbf{x}_t|\mathbf{x}_T)||_2^2]dt, \qquad (19)$$

$$\mathcal{L}_{\text{AE}} := \mathcal{L}_{\text{SM}} + \mathcal{H}(q_{\phi,\psi}(\mathbf{x}_0|\mathbf{x}_T)). \qquad (20)$$

We derive the score-matching objective $\mathcal{L}_{\text{SM}}$ with the denoising version for tractability. First, $\mathcal{L}_{\text{SM}}$ is derived as follows.

$$\mathcal{L}_{\text{SM}} = \frac{1}{2}\int_0^T \mathbb{E}_{q_{\phi,\psi}^t(\mathbf{x}_t,\mathbf{x}_T)}[g^2(t)||\mathbf{s}_{\boldsymbol{\theta}}(\mathbf{x}_t,t,\mathbf{x}_T)||_2^2 + g^2(t)||\nabla_{\mathbf{x}_t}\log q_{\phi,\psi}^t(\mathbf{x}_t|\mathbf{x}_T)||_2^2$$
$$- 2g^2(t)\mathbf{s}_{\boldsymbol{\theta}}(\mathbf{x}_t,t,\mathbf{x}_T)^T\nabla_{\mathbf{x}_t}\log q_{\phi,\psi}^t(\mathbf{x}_t|\mathbf{x}_T)]dt. \qquad (21)$$

Then, the last inner product term of Eq. (21) can be deduced in a similar approach to (Vincent, 2011):

$$\mathbb{E}_{q_{\phi,\psi}^t(\mathbf{x}_t,\mathbf{x}_T)}[\mathbf{s}_{\boldsymbol{\theta}}(\mathbf{x}_t,t,\mathbf{x}_T)^T\nabla_{\mathbf{x}_t}\log q_{\phi,\psi}^t(\mathbf{x}_t|\mathbf{x}_T)] \qquad (22)$$

$$= \int q_{\phi,\psi}^t(\mathbf{x}_t,\mathbf{x}_T)\mathbf{s}_{\boldsymbol{\theta}}(\mathbf{x}_t,t,\mathbf{x}_T)^T\nabla_{\mathbf{x}_t}\log q_{\phi,\psi}^t(\mathbf{x}_t|\mathbf{x}_T)d\mathbf{x}_t d\mathbf{x}_T \qquad (23)$$

$$= \int q_{\phi,\psi}^t(\mathbf{x}_T)q_{\phi,\psi}^t(\mathbf{x}_t|\mathbf{x}_T)\mathbf{s}_{\boldsymbol{\theta}}(\mathbf{x}_t,t,\mathbf{x}_T)^T\nabla_{\mathbf{x}_t}\log q_{\phi,\psi}^t(\mathbf{x}_t|\mathbf{x}_T)d\mathbf{x}_t d\mathbf{x}_T \qquad (24)$$

$$= \mathbb{E}_{q_{\phi,\psi}^t(\mathbf{x}_T)}\left[\int q_{\phi,\psi}^t(\mathbf{x}_t|\mathbf{x}_T)\mathbf{s}_{\boldsymbol{\theta}}(\mathbf{x}_t,t,\mathbf{x}_T)^T\nabla_{\mathbf{x}_t}\log q_{\phi,\psi}^t(\mathbf{x}_t|\mathbf{x}_T)d\mathbf{x}_t\right] \qquad (25)$$

$$= \mathbb{E}_{q_{\phi,\psi}^t(\mathbf{x}_T)}\left[\int \mathbf{s}_{\boldsymbol{\theta}}(\mathbf{x}_t,t,\mathbf{x}_T)^T\nabla_{\mathbf{x}_t}q_{\phi,\psi}^t(\mathbf{x}_t|\mathbf{x}_T)d\mathbf{x}_t\right] \qquad (26)$$

$$= \mathbb{E}_{q_{\phi,\psi}^t(\mathbf{x}_T)}\left[\int \mathbf{s}_{\boldsymbol{\theta}}(\mathbf{x}_t,t,\mathbf{x}_T)^T\left\{\nabla_{\mathbf{x}_t}\int q_{\phi,\psi}^t(\mathbf{x}_0|\mathbf{x}_T)q_{\phi,\psi}^t(\mathbf{x}_t|\mathbf{x}_T,\mathbf{x}_0)d\mathbf{x}_0\right\}d\mathbf{x}_t\right] \qquad (27)$$

$$= \mathbb{E}_{q_{\phi,\psi}^t(\mathbf{x}_T)}\left[\int \mathbf{s}_{\boldsymbol{\theta}}(\mathbf{x}_t,t,\mathbf{x}_T)^T\left\{\int q_{\phi,\psi}^t(\mathbf{x}_0|\mathbf{x}_T)\nabla_{\mathbf{x}_t}q_{\phi,\psi}^t(\mathbf{x}_t|\mathbf{x}_T,\mathbf{x}_0)d\mathbf{x}_0\right\}d\mathbf{x}_t\right] \qquad (28)$$

$$= \mathbb{E}_{q_{\phi,\psi}^t(\mathbf{x}_T)}\left[\int \mathbf{s}_{\boldsymbol{\theta}}(\mathbf{x}_t,t,\mathbf{x}_T)^T\left\{\int q_{\phi,\psi}^t(\mathbf{x}_0|\mathbf{x}_T)q_{\phi,\psi}^t(\mathbf{x}_t|\mathbf{x}_T,\mathbf{x}_0)\nabla_{\mathbf{x}_t}\log q_{\phi,\psi}^t(\mathbf{x}_t|\mathbf{x}_T,\mathbf{x}_0)d\mathbf{x}_0\right\}d\mathbf{x}_t\right] \qquad (29)$$

$$= \mathbb{E}_{q_{\phi,\psi}^t(\mathbf{x}_T)}\left[\int\int q_{\phi,\psi}^t(\mathbf{x}_0|\mathbf{x}_T)q_{\phi,\psi}^t(\mathbf{x}_t|\mathbf{x}_T,\mathbf{x}_0)\mathbf{s}_{\boldsymbol{\theta}}(\mathbf{x}_t,t,\mathbf{x}_T)^T\nabla_{\mathbf{x}_t}\log q_{\phi,\psi}^t(\mathbf{x}_t|\mathbf{x}_T,\mathbf{x}_0)d\mathbf{x}_0 d\mathbf{x}_t\right] \qquad (30)$$

$$= \mathbb{E}_{q_{\phi,\psi}^t(\mathbf{x}_0,\mathbf{x}_t,\mathbf{x}_T)}[\mathbf{s}_{\boldsymbol{\theta}}(\mathbf{x}_t,t,\mathbf{x}_T)^T\nabla_{\mathbf{x}_t}\log q_{\phi,\psi}^t(\mathbf{x}_t|\mathbf{x}_T,\mathbf{x}_0)] \qquad (31)$$

---

[3]Eq. (1) is a linear SDE when the drift function $\mathbf{f}$ is linear with respect to $\mathbf{x}_t$.

Next, we rewrite the second term of Eq. (21). To begin, we express the entropy $\mathcal{H}(q_{\phi,\psi}(\mathbf{x}_0|\mathbf{x}_T))$ with $\nabla_{\mathbf{x}_t} \log q_{\phi,\psi}^t(\mathbf{x}_t|\mathbf{x}_T)$, which is similar to the proof of Theorem 4 in (Song et al., 2021b). Let $\mathcal{H}(q_{\phi,\psi}(\mathbf{x}_t, \mathbf{x}_T)) := -\int q_{\phi,\psi}(\mathbf{x}_t, \mathbf{x}_T) \log q_{\phi,\psi}(\mathbf{x}_t, \mathbf{x}_T) \mathrm{d}\mathbf{x}_t \mathrm{d}\mathbf{x}_T$ be the joint entropy function of $q_{\phi,\psi}(\mathbf{x}_t, \mathbf{x}_T)$. Note that $\mathcal{H}(q_{\phi,\psi}(\mathbf{x}_T, \mathbf{x}_T)) = \mathcal{H}(q_{\phi,\psi}(\mathbf{x}_T))$. Then, we have

$$\mathcal{H}(q_{\phi,\psi}(\mathbf{x}_0, \mathbf{x}_T)) = \mathcal{H}(q_{\phi,\psi}(\mathbf{x}_T, \mathbf{x}_T)) + \int_T^0 \frac{\partial \mathcal{H}_t(\mathbf{x}_t, \mathbf{x}_T)}{\partial t} \mathrm{d}t. \tag{32}$$

We can expand the integrand of Eq. (32) as follows.

$$\frac{\partial \mathcal{H}_t(\mathbf{x}_t, \mathbf{x}_T)}{\partial t} = \frac{\partial}{\partial t} \left[ -\int q_{\phi,\psi}(\mathbf{x}_t, \mathbf{x}_T) \log q_{\phi,\psi}(\mathbf{x}_t, \mathbf{x}_T) \mathrm{d}\mathbf{x}_t \mathrm{d}\mathbf{x}_T \right] \tag{33}$$

$$= \frac{\partial}{\partial t} \left[ -\int q_{\phi,\psi}(\mathbf{x}_T) q_{\phi,\psi}(\mathbf{x}_t|\mathbf{x}_T)[\log q_{\phi,\psi}(\mathbf{x}_T) + \log q_{\phi,\psi}(\mathbf{x}_t|\mathbf{x}_T)] \mathrm{d}\mathbf{x}_t \mathrm{d}\mathbf{x}_T \right] \tag{34}$$

$$= -\int q_{\phi,\psi}(\mathbf{x}_T) \frac{\partial}{\partial t} \{ q_{\phi,\psi}(\mathbf{x}_t|\mathbf{x}_T)[\log q_{\phi,\psi}(\mathbf{x}_T) + \log q_{\phi,\psi}(\mathbf{x}_t|\mathbf{x}_T)] \} \mathrm{d}\mathbf{x}_t \mathrm{d}\mathbf{x}_T \tag{35}$$

$$= -\mathbb{E}_{q_{\phi,\psi}(\mathbf{x}_T)} \left[ \int \frac{\partial}{\partial t} \{ q_{\phi,\psi}(\mathbf{x}_t|\mathbf{x}_T)[\log q_{\phi,\psi}(\mathbf{x}_T) + \log q_{\phi,\psi}(\mathbf{x}_t|\mathbf{x}_T)] \} \mathrm{d}\mathbf{x}_t \right] \tag{36}$$

We further expand the integration in the last term as follows.

$$\int \frac{\partial}{\partial t} \{ q_{\phi,\psi}(\mathbf{x}_t|\mathbf{x}_T)[\log q_{\phi,\psi}(\mathbf{x}_T) + \log q_{\phi,\psi}(\mathbf{x}_t|\mathbf{x}_T)] \} \mathrm{d}\mathbf{x}_t \tag{37}$$

$$= \int \frac{\partial}{\partial t} \{ q_{\phi,\psi}(\mathbf{x}_t|\mathbf{x}_T) \} [\log q_{\phi,\psi}(\mathbf{x}_T) + \log q_{\phi,\psi}(\mathbf{x}_t|\mathbf{x}_T)] + q_{\phi,\psi}(\mathbf{x}_t|\mathbf{x}_T) \frac{\partial \log q_{\phi,\psi}(\mathbf{x}_t|\mathbf{x}_T)}{\partial t} \mathrm{d}\mathbf{x}_t \tag{38}$$

$$= \int \frac{\partial}{\partial t} \{ q_{\phi,\psi}(\mathbf{x}_t|\mathbf{x}_T) \} [\log q_{\phi,\psi}(\mathbf{x}_T) + \log q_{\phi,\psi}(\mathbf{x}_t|\mathbf{x}_T)] + \frac{\partial q_{\phi,\psi}(\mathbf{x}_t|\mathbf{x}_T)}{\partial t} \mathrm{d}\mathbf{x}_t \tag{39}$$

$$= \int \frac{\partial}{\partial t} \{ q_{\phi,\psi}(\mathbf{x}_t|\mathbf{x}_T) \} [\log q_{\phi,\psi}(\mathbf{x}_T) + \log q_{\phi,\psi}(\mathbf{x}_t|\mathbf{x}_T)] \mathrm{d}\mathbf{x}_t + \frac{\partial}{\partial t} \int q_{\phi,\psi}(\mathbf{x}_t|\mathbf{x}_T) \mathrm{d}\mathbf{x}_t \tag{40}$$

$$= \int \frac{\partial}{\partial t} \{ q_{\phi,\psi}(\mathbf{x}_t|\mathbf{x}_T) \} [\log q_{\phi,\psi}(\mathbf{x}_T) + \log q_{\phi,\psi}(\mathbf{x}_t|\mathbf{x}_T)] \mathrm{d}\mathbf{x}_t \tag{41}$$

$$= \int \frac{\partial}{\partial t} \{ q_{\phi,\psi}(\mathbf{x}_t|\mathbf{x}_T) \} \log q_{\phi,\psi}(\mathbf{x}_T) \mathrm{d}\mathbf{x}_t + \int \frac{\partial}{\partial t} \{ q_{\phi,\psi}(\mathbf{x}_t|\mathbf{x}_T) \} \log q_{\phi,\psi}(\mathbf{x}_t|\mathbf{x}_T) \mathrm{d}\mathbf{x}_t \tag{42}$$

$$= \log q_{\phi,\psi}(\mathbf{x}_T) \frac{\partial}{\partial t} \int q_{\phi,\psi}(\mathbf{x}_t|\mathbf{x}_T) \mathrm{d}\mathbf{x}_t + \int \frac{\partial}{\partial t} \{ q_{\phi,\psi}(\mathbf{x}_t|\mathbf{x}_T) \} \log q_{\phi,\psi}(\mathbf{x}_t|\mathbf{x}_T) \mathrm{d}\mathbf{x}_t \tag{43}$$

$$= \int \frac{\partial}{\partial t} \{ q_{\phi,\psi}(\mathbf{x}_t|\mathbf{x}_T) \} \log q_{\phi,\psi}(\mathbf{x}_t|\mathbf{x}_T) \mathrm{d}\mathbf{x}_t \tag{44}$$

Note that we use $\int q_{\phi,\psi}(\mathbf{x}_t|\mathbf{x}_T) \mathrm{d}\mathbf{x}_t = 1$ in Eqs. (41) and (44).

By eq. (51) in (Zhou et al., 2024), the Fokker-Plank equation for $q_{\phi,\psi}(\mathbf{x}_t|\mathbf{x}_T)$ follows

$$\frac{\partial}{\partial t} q_{\phi,\psi}(\mathbf{x}_t|\mathbf{x}_T) = -\nabla_{\mathbf{x}_t} \cdot \left[ (\mathbf{f}(\mathbf{x}_t, t) + g^2(t)\mathbf{h}(\mathbf{x}_t, t, \mathbf{x}_T, T)) q_{\phi,\psi}(\mathbf{x}_t|\mathbf{x}_T) \right]$$

$$+ \frac{1}{2} g^2(t) \nabla_{\mathbf{x}_t} \cdot \nabla_{\mathbf{x}_t} q_{\phi,\psi}(\mathbf{x}_t|\mathbf{x}_T) \tag{45}$$

$$= -\nabla_{\mathbf{x}_t} \cdot [\tilde{\mathbf{f}}_{\phi,\psi}(\mathbf{x}_t, t) q_{\phi,\psi}(\mathbf{x}_t|\mathbf{x}_T)], \tag{46}$$

where $\tilde{\mathbf{f}}_{\phi,\psi}(\mathbf{x}_t, t) := \mathbf{f}(\mathbf{x}_t, t) + g^2(t)\mathbf{h}(\mathbf{x}_t, t, \mathbf{x}_T, T) - \frac{1}{2}g^2(t)\nabla_{\mathbf{x}_t} \log q_{\phi,\psi}(\mathbf{x}_t|\mathbf{x}_T)$.

Combining Eqs. (36), (44) and (46), we have

$$\frac{\partial \mathcal{H}_t(\mathbf{x}_t, \mathbf{x}_T)}{\partial t} \tag{47}$$

$$= -\mathbb{E}_{q_{\phi,\psi}(\mathbf{x}_T)}\left[\int -\nabla_{\mathbf{x}_t} \cdot [\tilde{\mathbf{f}}_{\phi,\psi}(\mathbf{x}_t, t) q_{\phi,\psi}(\mathbf{x}_t|\mathbf{x}_T)] \log q_{\phi,\psi}(\mathbf{x}_t|\mathbf{x}_T) \mathrm{d}\mathbf{x}_t\right] \tag{48}$$

$$= \mathbb{E}_{q_{\phi,\psi}(\mathbf{x}_T)}\left[\int \nabla_{\mathbf{x}_t} \cdot [\tilde{\mathbf{f}}_{\phi,\psi}(\mathbf{x}_t, t) q_{\phi,\psi}(\mathbf{x}_t|\mathbf{x}_T)] \log q_{\phi,\psi}(\mathbf{x}_t|\mathbf{x}_T) \mathrm{d}\mathbf{x}_t\right] \tag{49}$$

$$= \mathbb{E}_{q_{\phi,\psi}(\mathbf{x}_T)}\Big[\tilde{\mathbf{f}}_{\phi,\psi}(\mathbf{x}_t, t) q_{\phi,\psi}(\mathbf{x}_t|\mathbf{x}_T) \log q_{\phi,\psi}(\mathbf{x}_t|\mathbf{x}_T)$$
$$- \int q_{\phi,\psi}(\mathbf{x}_t|\mathbf{x}_T) \tilde{\mathbf{f}}_{\phi,\psi}(\mathbf{x}_t, t)^T \nabla_{\mathbf{x}_t} \log q_{\phi,\psi}(\mathbf{x}_t|\mathbf{x}_T) \mathrm{d}\mathbf{x}_t\Big] \tag{50}$$

$$= \mathbb{E}_{q_{\phi,\psi}(\mathbf{x}_T)}\Big[-\int q_{\phi,\psi}(\mathbf{x}_t|\mathbf{x}_T) \tilde{\mathbf{f}}_{\phi,\psi}(\mathbf{x}_t, t)^T \nabla_{\mathbf{x}_t} \log q_{\phi,\psi}(\mathbf{x}_t|\mathbf{x}_T) \mathrm{d}\mathbf{x}_t\Big] \tag{51}$$

$$= \mathbb{E}_{q_{\phi,\psi}(\mathbf{x}_T)}\Big[-\int \{\mathbf{f}(\mathbf{x}_t, t) + g^2(t)\mathbf{h}(\mathbf{x}_t, t, \mathbf{x}_T, T) - \frac{1}{2}g^2(t)\nabla_{\mathbf{x}_t} \log q_{\phi,\psi}(\mathbf{x}_t|\mathbf{x}_T)\}^T$$
$$\nabla_{\mathbf{x}_t} \log q_{\phi,\psi}(\mathbf{x}_t|\mathbf{x}_T) q_{\phi,\psi}(\mathbf{x}_t|\mathbf{x}_T) \mathrm{d}\mathbf{x}_t\Big] \tag{52}$$

$$= \mathbb{E}_{q_{\phi,\psi}(\mathbf{x}_t, \mathbf{x}_T)}\Big[\{-\mathbf{f}(\mathbf{x}_t, t) - g^2(t)\mathbf{h}(\mathbf{x}_t, t, \mathbf{x}_T, T) + \frac{1}{2}g^2(t)\nabla_{\mathbf{x}_t} \log q_{\phi,\psi}(\mathbf{x}_t|\mathbf{x}_T)\}^T$$
$$\nabla_{\mathbf{x}_t} \log q_{\phi,\psi}(\mathbf{x}_t|\mathbf{x}_T)\Big] \tag{53}$$

$$= \mathbb{E}_{q_{\phi,\psi}(\mathbf{x}_t, \mathbf{x}_T)}\Big[\frac{1}{2}g^2(t)\|\nabla_{\mathbf{x}_t} \log q_{\phi,\psi}(\mathbf{x}_t|\mathbf{x}_T)\|_2^2$$
$$- \{\mathbf{f}(\mathbf{x}_t, t) + g^2(t)\mathbf{h}(\mathbf{x}_t, t, \mathbf{x}_T, T)\}^T \nabla_{\mathbf{x}_t} \log q_{\phi,\psi}(\mathbf{x}_t|\mathbf{x}_T)\Big]. \tag{54}$$

Therefore, the joint entropy function $\mathcal{H}(q_{\phi,\psi}(\mathbf{x}_0, \mathbf{x}_T))$ can be expressed as

$$\mathcal{H}(q_{\phi,\psi}(\mathbf{x}_0, \mathbf{x}_T)) = \mathcal{H}(q_{\phi,\psi}(\mathbf{x}_T)) + \int_T^0 \mathbb{E}_{q_{\phi,\psi}^t(\mathbf{x}_t, \mathbf{x}_T)}\Big[\frac{1}{2}g^2(t)\|\nabla_{\mathbf{x}_t} \log q_{\phi,\psi}^t(\mathbf{x}_t|\mathbf{x}_T)\|_2^2$$
$$- \mathbf{f}(\mathbf{x}_t, t)^T \nabla_{\mathbf{x}_t} \log q_{\phi,\psi}^t(\mathbf{x}_t|\mathbf{x}_T) - g^2(t)\mathbf{h}(\mathbf{x}_t, t, \mathbf{x}_T, T)^T \nabla_{\mathbf{x}_t} \log q_{\phi,\psi}^t(\mathbf{x}_t|\mathbf{x}_T)\Big]\mathrm{d}t. \tag{55}$$

We can re-write the above equation as follows.

$$\int_0^T \mathbb{E}_{q_{\phi,\psi}^t(\mathbf{x}_t, \mathbf{x}_T)}[g^2(t)\|\nabla_{\mathbf{x}_t} \log q_{\phi,\psi}^t(\mathbf{x}_t|\mathbf{x}_T)\|_2^2]\mathrm{d}t \tag{56}$$

$$= -2\mathcal{H}(q_{\phi,\psi}(\mathbf{x}_0|\mathbf{x}_T)) \tag{57}$$

$$+ \int_0^T \mathbb{E}_{q_{\phi,\psi}^t(\mathbf{x}_t, \mathbf{x}_T)}[2\mathbf{f}(\mathbf{x}_t, t)^T \nabla_{\mathbf{x}_t} \log q_{\phi,\psi}^t(\mathbf{x}_t|\mathbf{x}_T) + 2g^2(t)\mathbf{h}(\mathbf{x}_t, t, \mathbf{x}_T, T)^T \nabla_{\mathbf{x}_t} \log q_{\phi,\psi}^t(\mathbf{x}_t|\mathbf{x}_T)]\mathrm{d}t$$

$$= -2\mathcal{H}(q_{\phi,\psi}(\mathbf{x}_0|\mathbf{x}_T)) + 2\int_0^T \mathbb{E}_{q_{\phi,\psi}^t(\mathbf{x}_t, \mathbf{x}_T)}[\nabla_{\mathbf{x}_t} \cdot \{\mathbf{f}(\mathbf{x}_t, t) + g^2(t)\mathbf{h}(\mathbf{x}_t, t, \mathbf{x}_T, T)\}]\mathrm{d}t \tag{58}$$

Similar to the process above, we can obtain the following results for the following joint entropy function $\mathcal{H}(q_{\phi,\psi}(\mathbf{x}_0, \mathbf{x}_t, \mathbf{x}_T)) := -\int q_{\phi,\psi}(\mathbf{x}_0, \mathbf{x}_t, \mathbf{x}_T) \log q_{\phi,\psi}(\mathbf{x}_0, \mathbf{x}_t, \mathbf{x}_T) \mathrm{d}\mathbf{x}_0 \mathrm{d}\mathbf{x}_t \mathrm{d}\mathbf{x}_T$.

$$\mathcal{H}(q_{\phi,\psi}(\mathbf{x}_0, \mathbf{x}_0, \mathbf{x}_T)) = \mathcal{H}(q_{\phi,\psi}(\mathbf{x}_0, \mathbf{x}_T, \mathbf{x}_T)) + \int_T^0 \frac{\partial \mathcal{H}(\mathbf{x}_0, \mathbf{x}_t, \mathbf{x}_T)}{\partial t}\mathrm{d}t \tag{59}$$

In the following results, we utilize the Fokker-Plank equation for $q_{\phi,\psi}(\mathbf{x}_t|\mathbf{x}_0, \mathbf{x}_T)$, which comes from eq. (49) in (Zhou et al., 2024):

$$\frac{\partial}{\partial t}q_{\phi,\psi}(\mathbf{x}_t|\mathbf{x}_0, \mathbf{x}_T) = -\nabla_{\mathbf{x}_t} \cdot \Big[(\mathbf{f}(\mathbf{x}_t, t) + g^2(t)\mathbf{h}(\mathbf{x}_t, t, \mathbf{x}_T, T))q_{\phi,\psi}(\mathbf{x}_t|\mathbf{x}_0, \mathbf{x}_T)\Big]$$
$$+ \frac{1}{2}g^2(t)\nabla_{\mathbf{x}_t} \cdot \nabla_{\mathbf{x}_t} q_{\phi,\psi}(\mathbf{x}_t|\mathbf{x}_0, \mathbf{x}_T) \tag{60}$$

$$= -\nabla_{\mathbf{x}_t} \cdot [\hat{\mathbf{f}}_{\phi,\psi}(\mathbf{x}_t, t)q_{\phi,\psi}(\mathbf{x}_t|\mathbf{x}_0, \mathbf{x}_T)], \tag{61}$$

where $\hat{\mathbf{f}}_{\phi,\psi}(\mathbf{x}_t, t) := \mathbf{f}(\mathbf{x}_t, t) + g^2(t)\mathbf{h}(\mathbf{x}_t, t, \mathbf{x}_T, T) - \frac{1}{2}g^2(t)\nabla_{\mathbf{x}_t}\log q_{\phi,\psi}(\mathbf{x}_t|\mathbf{x}_0, \mathbf{x}_T)$.

Then, we have

$$0 = \int_T^0 \mathbb{E}_{q_{\phi,\psi}^t(\mathbf{x}_0, \mathbf{x}_t, \mathbf{x}_T)}\Big[\frac{1}{2}g^2(t)||\nabla_{\mathbf{x}_t}\log q_{\phi,\psi}^t(\mathbf{x}_t|\mathbf{x}_T, \mathbf{x}_0)||_2^2 - \mathbf{f}(\mathbf{x}_t, t)\nabla_{\mathbf{x}_t}\log q_{\phi,\psi}^t(\mathbf{x}_t|\mathbf{x}_T, \mathbf{x}_0)$$
$$- g^2(t)\mathbf{h}(\mathbf{x}_t, t, \mathbf{x}_T, T)\nabla_{\mathbf{x}_t}\log q_{\phi,\psi}^t(\mathbf{x}_t|\mathbf{x}_T, \mathbf{x}_0)\Big]\mathrm{d}t, \tag{62}$$

where the left hand side is from $0 = \mathcal{H}(q_{\phi,\psi}(\mathbf{x}_0, \mathbf{x}_0, \mathbf{x}_T)) - \mathcal{H}(q_{\phi,\psi}(\mathbf{x}_0, \mathbf{x}_T, \mathbf{x}_T))$, and right hand side is from $\int_T^0 \frac{\partial \mathcal{H}(\mathbf{x}_0, \mathbf{x}_t, \mathbf{x}_T)}{\partial t}\mathrm{d}t$. We can further derive as follows.

$$\int_0^T \mathbb{E}_{q_{\phi,\psi}^t(\mathbf{x}_0, \mathbf{x}_t, \mathbf{x}_T)}[g^2(t)||\nabla_{\mathbf{x}_t}\log q_{\phi,\psi}^t(\mathbf{x}_t|\mathbf{x}_T, \mathbf{x}_0)||_2^2]\mathrm{d}t$$
$$= 2\int_0^T \mathbb{E}_{q_{\phi,\psi}^t(\mathbf{x}_0, \mathbf{x}_t, \mathbf{x}_T)}[\nabla_{\mathbf{x}_t}\cdot\{\mathbf{f}(\mathbf{x}_t, t) + g^2(t)\mathbf{h}(\mathbf{x}_t, t, \mathbf{x}_T, T)\}]\mathrm{d}t \tag{63}$$

Combining Eqs. (58) and (63), we have

$$\int_0^T \mathbb{E}_{q_{\phi,\psi}^t(\mathbf{x}_t, \mathbf{x}_T)}[g^2(t)||\nabla_{\mathbf{x}_t}\log q_{\phi,\psi}^t(\mathbf{x}_t|\mathbf{x}_T)||_2^2]\mathrm{d}t$$
$$= -2\mathcal{H}(q_{\phi,\psi}(\mathbf{x}_0|\mathbf{x}_T)) + \int_0^T \mathbb{E}_{q_{\phi,\psi}^t(\mathbf{x}_0, \mathbf{x}_t, \mathbf{x}_T)}[g^2(t)||\nabla_{\mathbf{x}_t}\log q_{\phi,\psi}^t(\mathbf{x}_t|\mathbf{x}_T, \mathbf{x}_0)||_2^2]\mathrm{d}t \tag{64}$$

Combining all results, the score-matching objective $\mathcal{L}_{\mathrm{SM}}$ can be expressed as

$$\mathcal{L}_{\mathrm{SM}} = \frac{1}{2}\int_0^T \mathbb{E}_{q_{\phi,\psi}^t(\mathbf{x}_0, \mathbf{x}_t, \mathbf{x}_T)}[g^2(t)||\mathbf{s}_{\boldsymbol{\theta}}(\mathbf{x}_t, t, \mathbf{x}_T)||_2^2 + g^2(t)||\nabla_{\mathbf{x}_t}\log q_{\phi,\psi}^t(\mathbf{x}_t|\mathbf{x}_T, \mathbf{x}_0)||_2^2$$
$$- 2g^2(t)\mathbf{s}_{\boldsymbol{\theta}}(\mathbf{x}_t, t, \mathbf{x}_T)^T\nabla_{\mathbf{x}_t}\log q_{\phi,\psi}^t(\mathbf{x}_t|\mathbf{x}_T, \mathbf{x}_0)]\mathrm{d}t - \mathcal{H}(q_{\phi,\psi}(\mathbf{x}_0|\mathbf{x}_T)) \tag{65}$$
$$= \frac{1}{2}\int_0^T \mathbb{E}_{q_{\phi,\psi}^t(\mathbf{x}_0, \mathbf{x}_t, \mathbf{x}_T)}[g^2(t)||\mathbf{s}_{\boldsymbol{\theta}}(\mathbf{x}_t, t, \mathbf{x}_T) - \nabla_{\mathbf{x}_t}\log \tilde{q}_t(\mathbf{x}_t|\mathbf{x}_0, \mathbf{x}_T)||_2^2]\mathrm{d}t - \mathcal{H}(q_{\phi,\psi}(\mathbf{x}_0|\mathbf{x}_T)) \tag{66}$$

The last equality comes from $q_{\phi,\psi}^t(\mathbf{x}_t|\mathbf{x}_0, \mathbf{x}_T) = \tilde{q}_t(\mathbf{x}_t|\mathbf{x}_0, \mathbf{x}_T)$, which is based on the Doob's $h$-transform (Doob & Doob, 1984; Rogers & Williams, 2000; Zhou et al., 2024). Finally, we have

$$\mathcal{L}_{\mathrm{AE}} = \mathcal{L}_{\mathrm{SM}} + \mathcal{H}(q_{\phi,\psi}(\mathbf{x}_0|\mathbf{x}_T)) \tag{67}$$
$$= \frac{1}{2}\int_0^T \mathbb{E}_{q_{\phi,\psi}^t(\mathbf{x}_0, \mathbf{x}_t, \mathbf{x}_T)}[g^2(t)||\mathbf{s}_{\boldsymbol{\theta}}(\mathbf{x}_t, t, \mathbf{x}_T) - \nabla_{\mathbf{x}_t}\log \tilde{q}_t(\mathbf{x}_t|\mathbf{x}_0, \mathbf{x}_T)||_2^2]\mathrm{d}t. \tag{68}$$

From here, we show that the objective $\mathcal{L}_{\mathrm{AE}}$ is equivalent to the reconstruction objective. Assume that the forward SDE in Eq. (1) is a linear SDE in terms of $\mathbf{x}_t$ (e.g. VP (Ho et al., 2020), VE (Song et al., 2021c)). Then the transition kernel $\tilde{q}(\mathbf{x}_t|\mathbf{x}_0)$ becomes Gaussian distribution. Then, we can represent reparametrized form $\mathbf{x}_t = \alpha_t\mathbf{x}_0 + \sigma_t\boldsymbol{\epsilon}$, where $\alpha_t$ and $\sigma_t$ are time-dependent constants determined by drift $\mathbf{f}$ and volatility $g$, and $\boldsymbol{\epsilon} \sim \mathcal{N}(\mathbf{0}, \mathbf{I})$. The time-dependent constant signal-to-noise ratio $SNR(t) := \frac{\alpha_t^2}{\sigma_t^2}$ often define to discuss on diffusion process (Kingma et al., 2021). We define SNR ratio, $R(t) := \frac{SNR(T)}{SNR(t)}$ for convenient derivation.

Zhou et al. (2024) show the exact form of $\tilde{q}_t(\mathbf{x}_t|\mathbf{x}_0, \mathbf{x}_T) := \mathcal{N}(\hat{\mu}_t, \hat{\sigma}_t^2\mathbf{I})$, where $\hat{\mu}_t = R(t)\frac{\alpha_t}{\alpha_T}\mathbf{x}_T + \alpha_t\mathbf{x}_0(1 - R(t))$ and $\hat{\sigma}_t = \sigma_t\sqrt{1 - R(t)}$. This Gaussian form determines the exact analytic form of the score function $\nabla_{\mathbf{x}_t}\log \tilde{q}_t(\mathbf{x}_t|\mathbf{x}_0, \mathbf{x}_T)$. We plug this into our objective $\mathcal{L}_{\mathrm{AE}}$.

$$\mathcal{L}_{\text{AE}} = \frac{1}{2} \int_0^T \mathbb{E}_{q_{\phi,\psi}^t(\mathbf{x}_0,\mathbf{x}_t,\mathbf{x}_T)} [g^2(t) || \mathbf{s}_{\boldsymbol{\theta}}(\mathbf{x}_t, t, \mathbf{x}_T) - \nabla_{\mathbf{x}_t} \log \tilde{q}_t(\mathbf{x}_t|\mathbf{x}_0,\mathbf{x}_T) ||_2^2] \mathrm{d}t \tag{69}$$

$$= \frac{1}{2} \int_0^T \mathbb{E}_{q_{\phi,\psi}^t(\mathbf{x}_0,\mathbf{x}_t,\mathbf{x}_T)} [g^2(t) || \mathbf{s}_{\boldsymbol{\theta}}(\mathbf{x}_t, t, \mathbf{x}_T) - \frac{-\mathbf{x}_t + (R(t)\frac{\alpha_t}{\alpha_T}\mathbf{x}_T + \alpha_t\mathbf{x}_0(1 - R(t)))}{\sigma_t^2(1 - R(t))} ||_2^2] \mathrm{d}t \tag{70}$$

$$= \frac{1}{2} \int_0^T \mathbb{E}_{q_{\phi,\psi}^t(\mathbf{x}_0,\mathbf{x}_t,\mathbf{x}_T)} [\lambda(t) || \mathbf{x}_{\boldsymbol{\theta}}^0(\mathbf{x}_t, t, \mathbf{x}_T) - \mathbf{x}_0 ||_2^2] \mathrm{d}t, \tag{71}$$

where

$$\lambda(t) = \frac{\alpha_t}{\sigma_t^2} g^2(t), \tag{72}$$

$$\mathbf{x}_{\boldsymbol{\theta}}^0(\mathbf{x}_t, t, \mathbf{x}_T) := \alpha(t)\mathbf{x}_t + \beta(t)\mathbf{x}_T + \gamma(t)\mathbf{s}_{\boldsymbol{\theta}}(\mathbf{x}_t, t, \mathbf{x}_T), \tag{73}$$

$$\alpha(t) = \frac{1}{\alpha_t(1 - R(t))}, \quad \beta(t) = -\frac{R(t)}{\alpha_T(1 - R(t))}, \quad \gamma(t) = \frac{\sigma_t^2}{\alpha_t}. \tag{74}$$

$\square$

## A.2 PROOF OF THEOREM 3

**Theorem 3.** $D_{KL}(q_{data}(\mathbf{x}_0)||p_{\psi,\theta}(\mathbf{x}_0)) \leq \mathcal{L}_{AE} + \mathcal{L}_{PR} - H$, where $H = \mathcal{H}(q_{data}(\mathbf{x}_0))$ is a constant w.r.t. $\phi, \psi, \theta$.

*Proof.* From the data processing inequality with our graphical model, we have the following result, similar to eq. (14) in (Song et al., 2021a).

$$D_{\text{KL}}(q_{\text{data}}(\mathbf{x}_0)||p_{\psi,\theta}(\mathbf{x}_0)) \leq D_{\text{KL}}(q_{\phi,\psi}(\mathbf{x}_{0:T}, \mathbf{z})||p_{\psi,\theta}(\mathbf{x}_{0:T}, \mathbf{z})) \tag{75}$$

Also, the chain rule of KL divergences, we have

$$D_{\text{KL}}(q_{\phi,\psi}(\mathbf{x}_{0:T}, \mathbf{z})||p_{\psi,\theta}(\mathbf{x}_{0:T}, \mathbf{z})) \tag{76}$$
$$= D_{\text{KL}}(q_{\phi,\psi}(\mathbf{x}_T, \mathbf{z})||p_{\psi,\theta}(\mathbf{x}_T, \mathbf{z})) + \mathbb{E}_{q_{\phi,\psi}(\mathbf{x}_T, \mathbf{z})}[D_{\text{KL}}(\mu_{\phi,\psi}(\cdot|\mathbf{x}_T, \mathbf{z})||\nu_{\theta,\psi}(\cdot|\mathbf{x}_T, \mathbf{z}))], \tag{77}$$

where $\mu_{\phi,\psi}$ and $\nu_{\theta,\psi}$ are the path measures of the SDEs in Eqs. (78) and (79), respectively:

$$\mathrm{d}\mathbf{x}_t = [\mathbf{f}(\mathbf{x}_t, t) + g^2(t)\mathbf{h}(\mathbf{x}_t, t, \mathbf{y}, T)]\mathrm{d}t + g(t)\mathrm{d}\mathbf{w}_t, \quad \mathbf{x}_0 \sim q_{\text{data}}(\mathbf{x}_0), \quad \mathbf{x}_T \sim q_{\phi,\psi}(\mathbf{x}_T|\mathbf{x}_0), \tag{78}$$

$$\mathrm{d}\mathbf{x}_t = [\mathbf{f}(\mathbf{x}_t, t) - g^2(t)[\nabla_{\mathbf{x}_t} \log p_{\boldsymbol{\theta}}(\mathbf{x}_t|\mathbf{x}_T) - \mathbf{h}(\mathbf{x}_t, t, \mathbf{y}, T)]]\mathrm{d}t + g(t)\mathrm{d}\bar{\mathbf{w}}_t, \quad \mathbf{x}_T \sim p_{\psi}(\mathbf{x}_T). \tag{79}$$

By our graphical modeling, $\mathbf{z}$ is independent of $\{\mathbf{x}_t\}$ given $\mathbf{x}_T$. Therefore, we have

$$\mathbb{E}_{q_{\phi,\psi}(\mathbf{x}_T, \mathbf{z})}[D_{\text{KL}}(\mu_{\phi,\psi}(\cdot|\mathbf{x}_T, \mathbf{z})||\nu_{\boldsymbol{\theta}}(\cdot|\mathbf{x}_T, \mathbf{z}))] = \mathbb{E}_{q_{\phi,\psi}(\mathbf{x}_T)}[D_{\text{KL}}(\mu_{\phi,\psi}(\cdot|\mathbf{x}_T)||\nu_{\boldsymbol{\theta}}(\cdot|\mathbf{x}_T))], \tag{80}$$

where $\mu_{\phi,\psi}(\cdot|\mathbf{x}_T)$ and $\nu_{\boldsymbol{\theta}}(\cdot|\mathbf{x}_T)$ are the path measures of the SDEs in Eqs. (81) and (82), respectively:

$$\mathrm{d}\mathbf{x}_t = [\mathbf{f}(\mathbf{x}_t, t) - g^2(t)[\nabla_{\mathbf{x}_t} \log q_{\phi,\psi}(\mathbf{x}_t|\mathbf{x}_T) - \mathbf{h}(\mathbf{x}_t, t, \mathbf{y}, T)]]\mathrm{d}t + g(t)\mathrm{d}\bar{\mathbf{w}}_t, \quad \mathbf{x}(T) = \mathbf{x}_T, \tag{81}$$

$$\mathrm{d}\mathbf{x}_t = [\mathbf{f}(\mathbf{x}_t, t) - g^2(t)[\nabla_{\mathbf{x}_t} \log p_{\boldsymbol{\theta}}(\mathbf{x}_t|\mathbf{x}_T) - \mathbf{h}(\mathbf{x}_t, t, \mathbf{y}, T)]]\mathrm{d}t + g(t)\mathrm{d}\bar{\mathbf{w}}_t, \quad \mathbf{x}(T) = \mathbf{x}_T \tag{82}$$

Similar to eq. (17) in (Song et al., 2021a), this KL divergence can be expressed using the Girsanov theorem (Oksendal, 2013) and martingale property.

$$D_{\text{KL}}(\mu_{\phi,\psi}(\cdot|\mathbf{x}_T)||\nu_{\boldsymbol{\theta}}(\cdot|\mathbf{x}_T)) = \frac{1}{2} \int_0^T \mathbb{E}_{q_{\phi,\psi}(\mathbf{x}_t|\mathbf{x}_T)}[g^2(t)||\mathbf{s}_{\boldsymbol{\theta}}(\mathbf{x}_t, t, \mathbf{x}_T) - \nabla_{\mathbf{x}_t} \log q_{\phi,\psi}^t(\mathbf{x}_t|\mathbf{x}_T)||_2^2]\mathrm{d}t \tag{83}$$

From Eqs. (75), (77) and (83) and Theorem 1, we have:

$$D_{\mathrm{KL}}(q_{\mathrm{data}}(\mathbf{x}_0)||p_{\psi,\theta}(\mathbf{x}_0)) \leq D_{\mathrm{KL}}(q_{\phi,\psi}(\mathbf{x}_T,\mathbf{z})||p_{\psi,\theta}(\mathbf{x}_T,\mathbf{z})) + \mathcal{L}_{\mathrm{AE}} - \mathcal{H}(q_{\phi,\psi}(\mathbf{x}_0|\mathbf{x}_T)) \quad (84)$$

Furthermore, the first and third terms of RHS in Eq. (84) can be expressed as follows.

$$D_{\mathrm{KL}}(q_{\phi,\psi}(\mathbf{x}_T,\mathbf{z})||p_{\psi,\theta}(\mathbf{x}_T,\mathbf{z})) - \mathcal{H}(q_{\phi,\psi}(\mathbf{x}_0|\mathbf{x}_T)) \tag{85}$$

$$= \int q_{\phi,\psi}(\mathbf{x}_T,\mathbf{z})) \log \frac{q_{\phi,\psi}(\mathbf{x}_T,\mathbf{z})}{p_{\psi,\theta}(\mathbf{x}_T,\mathbf{z})} \mathrm{d}\mathbf{x}_T \mathrm{d}\mathbf{z} + \int q_{\phi,\psi}(\mathbf{x}_0,\mathbf{x}_T) \log q_{\phi,\psi}(\mathbf{x}_0|\mathbf{x}_T) \mathrm{d}\mathbf{x}_0 \mathrm{d}\mathbf{x}_T \tag{86}$$

$$= \int q_{\phi,\psi}(\mathbf{x}_0,\mathbf{x}_T,\mathbf{z}) \Big[ \log \frac{q_{\phi,\psi}(\mathbf{x}_T,\mathbf{z})}{p_{\psi,\theta}(\mathbf{x}_T,\mathbf{z})} + \log q_{\phi,\psi}(\mathbf{x}_0|\mathbf{x}_T) \Big] \mathrm{d}\mathbf{x}_0 \mathrm{d}\mathbf{x}_T \mathrm{d}\mathbf{z} \tag{87}$$

$$= \int q_{\phi,\psi}(\mathbf{x}_0,\mathbf{x}_T,\mathbf{z}) \Big[ \log \frac{q_{\phi,\psi}(\mathbf{x}_T)q_\psi(\mathbf{z}|\mathbf{x}_T)}{p_\psi(\mathbf{x}_T)p_\psi(\mathbf{z}|\mathbf{x}_T)} + \log q_{\phi,\psi}(\mathbf{x}_0|\mathbf{x}_T) \Big] \mathrm{d}\mathbf{x}_0 \mathrm{d}\mathbf{x}_T \mathrm{d}\mathbf{z} \tag{88}$$

$$= \int q_{\phi,\psi}(\mathbf{x}_0,\mathbf{x}_T,\mathbf{z}) \Big[ \log \frac{q_{\phi,\psi}(\mathbf{x}_T)}{p_\psi(\mathbf{x}_T)} + \log q_{\phi,\psi}(\mathbf{x}_0|\mathbf{x}_T) \Big] \mathrm{d}\mathbf{x}_0 \mathrm{d}\mathbf{x}_T \mathrm{d}\mathbf{z} \tag{89}$$

$$= \int q_{\phi,\psi}(\mathbf{x}_0,\mathbf{x}_T) \Big[ \log \frac{q_{\phi,\psi}(\mathbf{x}_T)q_{\phi,\psi}(\mathbf{x}_0|\mathbf{x}_T)}{p_\psi(\mathbf{x}_T)} \Big] \mathrm{d}\mathbf{x}_0 \mathrm{d}\mathbf{x}_T \tag{90}$$

$$= \int q_{\phi,\psi}(\mathbf{x}_0,\mathbf{x}_T) \Big[ \log \frac{q_{\mathrm{data}}(\mathbf{x}_0)q_{\phi,\psi}(\mathbf{x}_T|\mathbf{x}_0)}{p_\psi(\mathbf{x}_T)} \Big] \mathrm{d}\mathbf{x}_0 \mathrm{d}\mathbf{x}_T \tag{91}$$

$$= \int q_{\mathrm{data}}(\mathbf{x}_0)q_{\phi,\psi}(\mathbf{x}_T|\mathbf{x}_0) \Big[ \log \frac{q_{\phi,\psi}(\mathbf{x}_T|\mathbf{x}_0)}{p_\psi(\mathbf{x}_T)} + \log q_{\mathrm{data}}(\mathbf{x}_0) \Big] \mathrm{d}\mathbf{x}_0 \mathrm{d}\mathbf{x}_T \tag{92}$$

$$= \mathbb{E}_{q_{\mathrm{data}}(\mathbf{x}_0)}[D_{\mathrm{KL}}(q_{\phi,\psi}(\mathbf{x}_T|\mathbf{x}_0)||p_\psi(\mathbf{x}_T))] - \mathcal{H}(q_{\mathrm{data}}(\mathbf{x}_0)) \tag{93}$$

$$= \mathcal{L}_{\mathrm{PR}} - \mathcal{H}(q_{\mathrm{data}}(\mathbf{x}_0)) \tag{94}$$

To sum up, we have

$$D_{\mathrm{KL}}(q_{\mathrm{data}}(\mathbf{x}_0)||p_{\psi,\theta}(\mathbf{x}_0)) \leq \mathcal{L}_{\mathrm{AE}} + \mathcal{L}_{\mathrm{PR}} - \mathcal{H}(q_{\mathrm{data}}(\mathbf{x}_0)). \tag{95}$$

$\square$

## A.3 PRIOR OPTIMIZATION OBJECTIVE

This section explains the details of the prior related objective function mentioned in Section 4.4.2. The proposed objective is $\mathcal{L}_{\mathrm{PR}}$ as shown in Eq. (96).

$$\mathcal{L}_{\mathrm{PR}} = \mathbb{E}_{q_{\mathrm{data}}(\mathbf{x}_0)}[D_{\mathrm{KL}}(q_{\phi,\psi}(\mathbf{x}_T|\mathbf{x}_0)||p_\psi(\mathbf{x}_T))] \tag{96}$$

To optimize this term, we fix the parameters of the encoder ($\phi \to \phi^*$), the decoder ($\psi \to \psi^*$), and score network ($\theta \to \theta^*$), which is optimized by $\mathcal{L}_{AE}$. And we newly parameterize the generative prior $p_{\mathrm{prior}}(\mathbf{z}) \to p_\omega(\mathbf{z})$, so the generative endpoint distribution becomes $p_\psi(\mathbf{x}_T) \to p_{\psi^*,\omega}(\mathbf{x}_T)$. We utilize MLP-based latent diffusion models following (Preechakul et al., 2022; Zhang et al., 2022).

The objective function in Eq. (96) with respect to $\omega$ is described in Eq. (97) and extends to Eq. (99) with equality. Equation (100) is derived from the same optimality condition. In other words, it reduces the problem of training an unconditional generative prior $p_\omega(\mathbf{z})$ to matching the aggregated posterior distribution $q_{\phi^*}(\mathbf{z})$.

$$\arg\min_{\omega} \mathbb{E}_{q_{\mathrm{data}}(\mathbf{x}_0)}[D_{\mathrm{KL}}(q_{\phi^*,\psi^*}(\mathbf{x}_T|\mathbf{x}_0)||p_{\psi^*,\omega}(\mathbf{x}_T))] \tag{97}$$

$$\Leftrightarrow \arg\min_{\omega} \int q_{\mathrm{data}}(\mathbf{x}_0)q_{\phi^*,\psi^*}(\mathbf{x}_T|\mathbf{x}_0) \log \frac{q_{\phi^*,\psi^*}(\mathbf{x}_T|\mathbf{x}_0)}{p_{\psi^*,\omega}(\mathbf{x}_T)} \mathrm{d}\mathbf{x}_0 \mathrm{d}\mathbf{x}_T \tag{98}$$

$$\Leftrightarrow \arg\min_{\omega} D_{\mathrm{KL}}(q_{\phi^*,\psi^*}(\mathbf{x}_T)||p_{\psi^*,\omega}(\mathbf{x}_T)) + C \tag{99}$$

$$\Leftrightarrow \arg\min_{\omega} D_{\mathrm{KL}}(q_{\phi^*}(\mathbf{z})||p_\omega(\mathbf{z})) \tag{100}$$

## A.4 MUTUAL INFORMATION ANALYSIS

Alemi et al. (2018) shows the *distortion*; reconstruction error with inferred $\mathbf{z}$ is the variational bound of mutual information between $\mathbf{x}_0$ and $\mathbf{z}$ in the autoencoding framework. We explain the functional form of *distortion* in both the auxiliary encoder framework (Appendix A.4.1) and DBAE (Appendix A.4.2).

### A.4.1 AUXILIARY ENCODER FRAMEWORK

In the auxiliary encoder framework (e.g., DiffAE (Preechakul et al., 2022)), the *distortion* := $\mathbb{E}_{q_{\text{data}}(\mathbf{x}_0),q_\phi(\mathbf{z}|\mathbf{x}_0)}[-\log p_{\boldsymbol{\theta}}(\mathbf{x}_0|\mathbf{z})]$ and mutual information $MI(\mathbf{x}_0,\mathbf{z}) := \mathbb{E}_{q_\phi(\mathbf{x}_0,\mathbf{z})}[\log \frac{q_\phi(\mathbf{x}_0,\mathbf{z})}{q_{\text{data}}(\mathbf{x}_0)q_\phi(\mathbf{z})}]$ has a relation

$$-\mathbb{E}_{q_{\text{data}}(\mathbf{x}_0),q_\phi(\mathbf{z}|\mathbf{x}_0)}[-\log p_{\boldsymbol{\theta}}(\mathbf{x}_0|\mathbf{z})] + \mathcal{H}(q_{\text{data}}(\mathbf{x}_0)) \le MI(\mathbf{x}_0,\mathbf{z}), \tag{101}$$

where $p_{\boldsymbol{\theta}}(\mathbf{x}_0|\mathbf{z}) = \int p_{\text{prior}}(\mathbf{x}_T)p_{\boldsymbol{\theta}}^{\text{ODE}}(\mathbf{x}_0|\mathbf{z},\mathbf{x}_T)d\mathbf{x}_T$, when this framework reconstruct only with inferred $\mathbf{z}$.

We have the followings

$$\log p_{\boldsymbol{\theta}}(\mathbf{x}_0|\mathbf{z}) \tag{102}$$

$$= \log \int p_{\text{prior}}(\mathbf{x}_T)p_{\boldsymbol{\theta}}^{\text{ODE}}(\mathbf{x}_0|\mathbf{z},\mathbf{x}_T)d\mathbf{x}_T \tag{103}$$

$$= \log \int p_{\text{prior}}(\mathbf{x}_T)p_{\boldsymbol{\theta}}^{\text{ODE}}(\mathbf{x}_0|\mathbf{z},\mathbf{x}_T)\frac{q_{\boldsymbol{\theta}}^{\text{ODE}}(\mathbf{x}_T|\mathbf{z},\mathbf{x}_0)}{q_{\boldsymbol{\theta}}^{\text{ODE}}(\mathbf{x}_T|\mathbf{z},\mathbf{x}_0)}d\mathbf{x}_T \tag{104}$$

$$\ge \int q_{\boldsymbol{\theta}}^{\text{ODE}}(\mathbf{x}_T|\mathbf{z},\mathbf{x}_0) \log \frac{p_{\text{prior}}(\mathbf{x}_T)p_{\boldsymbol{\theta}}^{\text{ODE}}(\mathbf{x}_0|\mathbf{z},\mathbf{x}_T)}{q_{\boldsymbol{\theta}}^{\text{ODE}}(\mathbf{x}_T|\mathbf{z},\mathbf{x}_0)}d\mathbf{x}_T \tag{105}$$

$$= \mathbb{E}_{q_{\boldsymbol{\theta}}^{\text{ODE}}(\mathbf{x}_T|\mathbf{x}_0,\mathbf{z})}[\log p_{\boldsymbol{\theta}}^{\text{ODE}}(\mathbf{x}_0|\mathbf{z},\mathbf{x}_T)] - D_{KL}(q_{\boldsymbol{\theta}}^{\text{ODE}}(\mathbf{x}_T|\mathbf{x}_0,\mathbf{z})||p_{\text{prior}}(\mathbf{x}_T)). \tag{106}$$

$$= \int q_{\boldsymbol{\theta}}^{\text{ODE}}(\mathbf{x}_T|\mathbf{z},\mathbf{x}_0) \log \frac{p_{\text{prior}}(\mathbf{x}_T)\cancel{p_{\boldsymbol{\theta}}^{\text{ODE}}(\mathbf{x}_0|\mathbf{z},\mathbf{x}_T)}}{\cancel{q_{\boldsymbol{\theta}}^{\text{ODE}}(\mathbf{x}_T|\mathbf{z},\mathbf{x}_0)}}d\mathbf{x}_T \tag{107}$$

$$= \int q_{\boldsymbol{\theta}}^{\text{ODE}}(\mathbf{x}_T|\mathbf{z},\mathbf{x}_0) \log p_{\text{prior}}(\mathbf{x}_T)d\mathbf{x}_T \tag{108}$$

$$= -CE(q_{\boldsymbol{\theta}}^{\text{ODE}}(\mathbf{x}_T|\mathbf{z},\mathbf{x}_0)||p_{\text{prior}}(\mathbf{x}_T)) \tag{109}$$

Note that $p_{\boldsymbol{\theta}}^{\text{ODE}}(\mathbf{x}_0|\mathbf{z},\mathbf{x}_T) = q_{\boldsymbol{\theta}}^{\text{ODE}}(\mathbf{x}_T|\mathbf{z},\mathbf{x}_0)$ because the deterministic coupling of $(\mathbf{x}_0,\mathbf{x}_T)$ is given by the ODE in Eq. (110). When the coupling $(\mathbf{x}_0,\mathbf{x}_T)$ lies on the ODE path, both probabilities $p_{\boldsymbol{\theta}}^{\text{ODE}}(\mathbf{x}_0|\mathbf{z},\mathbf{x}_T)$ and $q_{\boldsymbol{\theta}}^{\text{ODE}}(\mathbf{x}_T|\mathbf{z},\mathbf{x}_0)$ become infinite. When the coupling $(\mathbf{x}_0,\mathbf{x}_T)$ is outside the ODE path, both probabilities $p_{\boldsymbol{\theta}}^{\text{ODE}}(\mathbf{x}_0|\mathbf{z},\mathbf{x}_T)$ and $q_{\boldsymbol{\theta}}^{\text{ODE}}(\mathbf{x}_T|\mathbf{z},\mathbf{x}_0)$ become zero.

$$d\mathbf{x}_t = [\mathbf{f}(\mathbf{x}_t,t) - \frac{1}{2}g^2(t)\mathbf{s}_{\boldsymbol{\theta}}(\mathbf{x}_t,\mathbf{z},t)]dt. \tag{110}$$

From Eq. (101) and Eq. (109), we have the following.

$$\mathbb{E}_{q_{\text{data}}(\mathbf{x}_0),q_\phi(\mathbf{z}|\mathbf{x}_0)}[-CE(q_{\boldsymbol{\theta}}^{\text{ODE}}(\mathbf{x}_T|\mathbf{z},\mathbf{x}_0)||p_{\text{prior}}(\mathbf{x}_T))] + \mathcal{H}(q_{\text{data}}(\mathbf{x}_0)) \le MI(\mathbf{x}_0,\mathbf{z}) \tag{111}$$

The discrepancy between $q_{\boldsymbol{\theta}}^{\text{ODE}}(\mathbf{x}_T|\mathbf{x}_0,\mathbf{z})$ and $p_{\text{prior}}(\mathbf{x}_T)$ makes the lower bound of mutual information between $\mathbf{x}_0$ and $\mathbf{z}$ loose. This discrepancy is inevitable from the deterministic nature of $q_{\boldsymbol{\theta}}^{\text{ODE}}(\mathbf{x}_T|\mathbf{z},\mathbf{x}_0)$.

This discrepancy is empirically observed in Table 2, providing two cases of $\mathbf{x}_T$ draw (random $\mathbf{x}_T$, inferred $\mathbf{x}_T$) in the auxiliary encoder models. The reconstruction gap between (random $\mathbf{x}_T$, inferred $\mathbf{x}_T$) is significant in practice. However, the inference of $\mathbf{x}_T$ is computationally expensive and inflexible in terms of dimensionality. If we only consider $\mathbf{z}$ inference, the information leakage is inevitable due to the functional form of diffusion models with an auxiliary encoder.

### A.4.2 DIFFUSION BRIDGE AUTOENCODERS

In the DBAE, the *distortion* $:= \mathbb{E}_{q_{\text{data}}(\mathbf{x}_0), q_\phi(\mathbf{z}|\mathbf{x}_0)}[-\log p_{\boldsymbol{\theta}, \boldsymbol{\psi}}(\mathbf{x}_0|\mathbf{z})]$ term and mutual information between $\mathbf{x}_0$ and $\mathbf{z}$ has relation in Eq. (112).

$$-\mathbb{E}_{q_{\text{data}}(\mathbf{x}_0), q_\phi(\mathbf{z}|\mathbf{x}_0)}[-\log p_{\boldsymbol{\theta}, \boldsymbol{\psi}}(\mathbf{x}_0|\mathbf{z})] + \mathcal{H}(q_{\text{data}}(\mathbf{x}_0)) \leq MI(\mathbf{x}_0, \mathbf{z}), \tag{112}$$

where $p_{\boldsymbol{\theta}, \boldsymbol{\psi}}(\mathbf{x}_0|\mathbf{z}) = \int p_{\boldsymbol{\theta}}(\mathbf{x}_0|\mathbf{x}_T) p_{\boldsymbol{\psi}}(\mathbf{x}_T|\mathbf{z}) d\mathbf{x}_T$. We have followings

$$\log p_{\boldsymbol{\theta}, \boldsymbol{\psi}}(\mathbf{x}_0|\mathbf{z}) \tag{113}$$

$$= \log \int p_{\boldsymbol{\theta}}(\mathbf{x}_0|\mathbf{x}_T) p_{\boldsymbol{\psi}}(\mathbf{x}_T|\mathbf{z}) d\mathbf{x}_T \tag{114}$$

$$= \log \int p_{\boldsymbol{\theta}}(\mathbf{x}_0|\mathbf{x}_T) p_{\boldsymbol{\psi}}(\mathbf{x}_T|\mathbf{z}) \frac{q_{\boldsymbol{\psi}}(\mathbf{x}_T|\mathbf{z})}{q_{\boldsymbol{\psi}}(\mathbf{x}_T|\mathbf{z})} d\mathbf{x}_T \tag{115}$$

$$\geq \int q_{\boldsymbol{\psi}}(\mathbf{x}_T|\mathbf{z}) \log \frac{p_{\boldsymbol{\theta}}(\mathbf{x}_0|\mathbf{x}_T) p_{\boldsymbol{\psi}}(\mathbf{x}_T|\mathbf{z})}{q_{\boldsymbol{\psi}}(\mathbf{x}_T|\mathbf{z})} d\mathbf{x}_T \tag{116}$$

$$= \mathbb{E}_{q_{\boldsymbol{\psi}}(\mathbf{x}_T|\mathbf{z})}[\log p_{\boldsymbol{\theta}}(\mathbf{x}_0|\mathbf{x}_T)] - D_{KL}(q_{\boldsymbol{\psi}}(\mathbf{x}_T|\mathbf{z})||p_{\boldsymbol{\psi}}(\mathbf{x}_T|\mathbf{z})) \tag{117}$$

Since $D_{KL}(q_{\boldsymbol{\psi}}(\mathbf{x}_T|\mathbf{z})||p_{\boldsymbol{\psi}}(\mathbf{x}_T|\mathbf{z})) = 0$, we have followings from Eq. (112) and Eq. (117).

$$\mathbb{E}_{q_{\text{data}}(\mathbf{x}_0), q_\phi(\mathbf{z}|\mathbf{x}_0)}[\mathbb{E}_{q_{\boldsymbol{\psi}}(\mathbf{x}_T|\mathbf{z})}[\log p_{\boldsymbol{\theta}}(\mathbf{x}_0|\mathbf{x}_T)]] + \mathcal{H}(q_{\text{data}}(\mathbf{x}_0)) \leq MI(\mathbf{x}_0, \mathbf{z}). \tag{118}$$

Unlike in Eq. (111), the $\mathbf{x}_T$ related term does not hinder maximizing mutual information between $\mathbf{x}_0$ and $\mathbf{z}$. Moreover, the remaining term $\mathbb{E}_{q_{\text{data}}(\mathbf{x}_0), q_\phi(\mathbf{z}|\mathbf{x}_0)}[\mathbb{E}_{q_{\boldsymbol{\psi}}(\mathbf{x}_T|\mathbf{z})}[\log p_{\boldsymbol{\theta}}(\mathbf{x}_0|\mathbf{x}_T)]]$ can maximized by our training, as we explain in Theorem 2.

### A.5 PROOF OF THEOREM 2

**Theorem 2.** $-MI(\mathbf{x}_0, \mathbf{z}) \leq \mathcal{L}_{AE} - H$, *where* $H = \mathcal{H}(q_{data}(\mathbf{x}_0))$ *is a constant w.r.t.* $\boldsymbol{\phi}, \boldsymbol{\psi}, \boldsymbol{\theta}$.

*Proof.* From data processing inequality similar in Eq. (75),

$$\mathbb{E}_{q_{\boldsymbol{\phi}, \boldsymbol{\psi}}(\mathbf{x}_T)}[D_{\text{KL}}(q_{\boldsymbol{\phi}, \boldsymbol{\psi}}(\mathbf{x}_0|\mathbf{x}_T)||p_{\boldsymbol{\theta}}(\mathbf{x}_0|\mathbf{x}_T))] \leq \mathbb{E}_{q_{\boldsymbol{\phi}, \boldsymbol{\psi}}(\mathbf{x}_T)}[D_{\text{KL}}(\mu_{\boldsymbol{\phi}, \boldsymbol{\psi}}(\cdot|\mathbf{x}_T)||\nu_{\boldsymbol{\theta}}(\cdot|\mathbf{x}_T))] \tag{119}$$

The LHS of Eq. (119) becomes followings,

$$\mathbb{E}_{q_{\boldsymbol{\phi}, \boldsymbol{\psi}}(\mathbf{x}_T)}[D_{\text{KL}}(q_{\boldsymbol{\phi}, \boldsymbol{\psi}}(\mathbf{x}_0|\mathbf{x}_T)||p_{\boldsymbol{\theta}}(\mathbf{x}_0|\mathbf{x}_T))] = \mathbb{E}_{q_{\boldsymbol{\phi}, \boldsymbol{\psi}}(\mathbf{x}_0, \mathbf{x}_T)}[-\log p_{\boldsymbol{\theta}}(\mathbf{x}_0|\mathbf{x}_T)] - \mathcal{H}(q_{\boldsymbol{\phi}, \boldsymbol{\psi}}(\mathbf{x}_0|\mathbf{x}_T)) \tag{120}$$

The RHS of Eq. (119) becomes followings from the result of Eq. (83),

$$\mathbb{E}_{q_{\boldsymbol{\phi}, \boldsymbol{\psi}}(\mathbf{x}_T)}[D_{\text{KL}}(\mu_{\boldsymbol{\phi}, \boldsymbol{\psi}}(\cdot|\mathbf{x}_T)||\nu_{\boldsymbol{\theta}}(\cdot|\mathbf{x}_T))] = \mathcal{L}_{\text{SM}} = \mathcal{L}_{\text{AE}} - \mathcal{H}(q_{\boldsymbol{\phi}, \boldsymbol{\psi}}(\mathbf{x}_0|\mathbf{x}_T)) \tag{121}$$

From Eqs. (119) to (121), we have the followings

$$\mathbb{E}_{q_{\boldsymbol{\phi}, \boldsymbol{\psi}}(\mathbf{x}_0, \mathbf{x}_T)}[-\log p_{\boldsymbol{\theta}}(\mathbf{x}_0|\mathbf{x}_T)] \leq \mathcal{L}_{\text{AE}} \tag{122}$$

We have the following to sum up Eq. (122) and Eq. (118).

$$-MI(\mathbf{x}_0, \mathbf{z}) \leq \mathcal{L}_{\text{AE}} - \mathcal{H}(q_{\text{data}}(\mathbf{x}_0)) \tag{123}$$

$\square$

## B RELATED WORK

### B.1 REPRESENTATION LEARNING IN DIFFUSION MODELS

Expanding the applicability of generative models to various downstream tasks depends on exploring meaningful latent variables through representation learning. Methods within both variational autoencoders (VAEs) (Kingma & Welling, 2014; Rezende et al., 2014; Higgins et al., 2017; Zhao et al., 2019; Kim & Mnih, 2018) and generative adversarial networks (GANs) (Jeon et al., 2021; Karras et al., 2020; Abdal et al., 2019; 2020; Chen et al., 2016) have been proposed; however, VAEs suffer

from low sample quality, limiting their practical deployment in real-world scenarios. Conversely, GANs are known for their ability to produce high-quality samples with fast sampling speeds but face challenges in accessing latent variables due to their intractable model structure. This leads to computationally expensive inference methods like GAN inversion (Xia et al., 2022; Voynov & Babenko, 2020; Zhu et al., 2016; Karras et al., 2020; Abdal et al., 2019). Additionally, the adversarial training objective of GANs introduces instability during the training.

In contrast, recent research has delved into representation learning within diffusion probabilistic models (DPMs), which offer stable training and high sample quality. In early studies, the diffusion endpoint $\mathbf{x}_T$ was introduced as a latent variable (Song et al., 2021a;c) with an invertible path defined by an ordinary differential equation (ODE). However, $\mathbf{x}_T$ is difficult to consider as a semantically meaningful encoding. Additionally, the dimension of $\mathbf{x}_T$ matches that of the original data $\mathbf{x}_0$, limiting the ability to learn condensed feature representation for downstream tasks (e.g., downstream inference, attribute manipulation with linear classifier). The inference of latent variables also relies on solving ODE, rendering inference intractable. This intractability not only hinders the desired regularization (e.g. disentanglment (Higgins et al., 2017; Kim & Mnih, 2018; Chen et al., 2018)) of the latent variable but also slows down the downstream applications.

Diffusion AutoEncoder (DiffAE) (Preechakul et al., 2022) introduces a new framework for learning tractable latent variables in DPMs. DiffAE learns representation in the latent variable $\mathbf{z}$ through an auxiliary encoder, with a $\mathbf{z}$-conditional score network (Ronneberger et al., 2015). The encoder-generated latent variable $\mathbf{z}$ can learn a semantic representation with a flexible dimensionality. Pre-trained DPM AutoEncoding (PDAE) (Zhang et al., 2022) proposes a method to learn unsupervised representation from pre-trained unconditional DPMs. PDAE also employs an auxiliary encoder to define $\mathbf{z}$ and introduces a decoder to represent $\nabla_{\mathbf{x}_t} \log p(\mathbf{z}|\mathbf{x}_t)$. PDAE can parameterize the $\mathbf{z}$-conditional model score combined with a pre-trained unconditional score network, utilizing the idea of classifier guidance (Dhariwal & Nichol, 2021). PDAE can use the pre-trained checkpoint from publicly available sources, but its complex decoder architecture slows down the sampling speed.

Subsequent studies have imposed additional assumptions or constraints on the encoder based on specific objectives. DiTi (Yue et al., 2024) introduces a time-dependent latent variable on the top of PDAE to enable feature learning that depends on diffusion time. InfoDiffusion (Wang et al., 2023) regularizes the latent space of DiffAE to foster an informative and disentangled representation of $\mathbf{z}$. It should be noted that such proposed regularization in (Wang et al., 2023) is also applicable with DBAE, and Section 5.3 demonstrates that the tradeoff between disentanglement and sample quality is better managed in DBAE than in DiffAE. FDAE (Wu & Zheng, 2024) learns disentangled latent representation by masking image pixel content with DiffAE. DisDiff (Yang et al., 2023) learns disentangled latent variable $\mathbf{z}$ by minimizing mutual information between each latent variable from different dimensions atop PDAE. LCG-DM (Kim et al., 2022b) adopts a pre-trained disentangled encoder and trains DiffAE structure with fixed encoder parameters to enable unsupervised controllable generation. SODA (Hudson et al., 2023) improves the network architectures of DiffAE and training for novel image reconstruction.

All the frameworks (Preechakul et al., 2022; Zhang et al., 2022) and applications (Yue et al., 2024; Wang et al., 2023; Wu & Zheng, 2024; Yang et al., 2023; Hudson et al., 2023) utilize the encoder and do not consider the diffusion endpoint $\mathbf{x}_T$, leading to an *information split problem*. In contrast, DBAE constructs an $\mathbf{z}$-dependent endpoint $\mathbf{x}_T$ inference with feed-forward architecture to induce $\mathbf{z}$ as an information bottleneck. Our framework makes $\mathbf{z}$ more informative, which is orthogonal to advancements in downstream applications (Kim et al., 2022b; Yue et al., 2024; Wang et al., 2023; Wu & Zheng, 2024; Yang et al., 2023; Hudson et al., 2023), as exemplified in Section 5.3.

### B.2 PARAMETRIZED FORWARD DIFFUSION

The forward diffusion process with learnable parameters is a key technique in DBAE to resolve *information split problem*. We summarize several other methods that proposed a learnable forward process. Note that DBAE has clear technical differences from those methods.

Schödinger bridge problem (SBP) (De Bortoli et al., 2021; Chen et al., 2022) learns the pair of SDEs that have forward and reverse dynamics relationships. SBP identifies the joint distribution in the form of a diffusion path between two given marginal distributions. The optimization is reduced to entropy-regularized optimal transport (Schrödinger, 1932; Genevay et al., 2018), which is often

solved by Iterative Proportional Fitting (Ruschendorf, 1995). For this optimization, samples are required at any given time $t$ from the forward SDE; however, these samples are not from a Gaussian kernel like Eq. (1) or Eq. (5), resulting in longer training times needed to solve the SDE numerically with intermediate particles. The formulation is also not suitable for our case, as we learn the given joint distribution through an encoder-decoder framework.

Diffusion normalizing flow (DiffFlow) (Zhang & Chen, 2021) parameterizes the drift term in Eq. (1) using a normalizing flow, making the endpoint of DiffFlow learnable. However, both training and endpoint inference are intractable because the parametrized forward SDE does not provide a Gaussian kernel similar to that in SBP. Implicit nonlinear diffusion model (INDM) (Kim et al., 2022a) learns a diffusion model that is defined in the latent space of a normalizing flow, implicitly parameterizing both the drift and volatility terms in Eq. (1). A unique benefit is its tractable training, allowing direct sampling from any diffusion time $t$. However, INDM merely progresses the existing diffusion process in the flow latent space, making it unsuitable for encoding due to technical issues such as dimensionality. The inference also requires solving the ODE for encoding.

Unlike other studies, we parameterize the endpoint $\mathbf{x}_T$ rather than the drift or volatility terms. The forward process is naturally influenced by the endpoint determined from Doob's $h$-transform. Unlike other parameterized diffusions, our approach ensures tractable learning and $\mathbf{x}_T$ inference, making it particularly advantageous for encoding tasks.

## C  IMPLEMENTATION DETAILS

### C.1  TRAINING CONFIGURATION

**Model Architecture** We use the score network ($\boldsymbol{\theta}$) backbone U-Net (Ronneberger et al., 2015), which are modified for diffusion models (Dhariwal & Nichol, 2021) with time-embedding. Dif-fAE (Preechakul et al., 2022), PDAE (Zhang et al., 2022), and DiTi (Yue et al., 2024) also utilize the same score network architecture. The only difference for DBAE is the endpoint $\mathbf{x}_T$ conditioning. We follow DDBM (Zhou et al., 2024) which concatenate $\mathbf{x}_t$ and $\mathbf{x}_T$ for the inputs as described in Figure 7b. This modification only increases the input channels, so the complexity increase is marginal. While the endpoint $\mathbf{x}_T$ contains all the information from $\mathbf{z}$, we design a score network also conditioning on $\mathbf{z}$ for implementation to effectively utilize the latent information in the generative process. For the encoder ($\phi$), we utilize the same structure from DiffAE (Preechakul et al., 2022). For the decoder ($\psi$), we adopt the upsampling structure from the generator of FastGAN (Liu et al., 2021), while removing the intermediate stochastic element. For the generative prior ($\boldsymbol{\omega}$), we utilize latent ddim from (Preechakul et al., 2022). Tables 5 and 6 explains the network configurations for the aforementioned structures.

**Optimization** We follow the optimization argument from DDBM (Zhou et al., 2024) with Variance Preserving (VP) SDE. We utilize the preconditioning and time-weighting proposed in DDBM, with the pred-x parameterization (Karras et al., 2022). Table 5 shows the remaining optimization hyperparameters. While DDBM does not include the encoder ($\phi$) and the decoder ($\psi$), we optimize jointly the parameters $\phi$, $\psi$, and $\boldsymbol{\theta}$ to minimize $\mathcal{L}_{\text{AE}}$.

### C.2  EVALUATION CONFIGURATION AND METRIC

**Downstream Inference** In Table 1, we use Average Precision (AP), Pearson Correlation Coefficient (Pearson's r), and Mean Squared Error (MSE) as metrics for comparison. For AP measurement, we train a linear classifier ($\mathbb{R}^l \rightarrow [0, 1]^{40}$) to classify 40 binary attribute labels from the CelebA (Liu et al., 2015) training dataset. The output of the encoder, $\text{Enc}_\phi(\mathbf{x}_0) = \mathbf{z}$, serves as the input for a linear classifier. We examine the CelebA test dataset. Precision and recall for each attribute label are calculated by computing true positives (TP), false positives (FP), and false negatives (FN) for each threshold interval divided by predicted values. The area under the precision-recall curve is obtained as AP. For Pearson's r and MSE, we train a linear regressor ($\mathbb{R}^l \rightarrow \mathbb{R}^{73}$) using LFW (Huang et al., 2007; Kumar et al., 2009) dataset. The regressor predicts the value of 73 attributes based on the latent variable $\mathbf{z}$. Pearson's r is evaluated by calculating the variance and covariance between the ground truth and predicted values for each attribute, while MSE is assessed by measuring the differences between two values. We borrow the baseline results from the DiTi (Yue et al., 2024) paper and adhere to the evaluation protocol found at `https://github.com/yue-zhongqi/diti`.

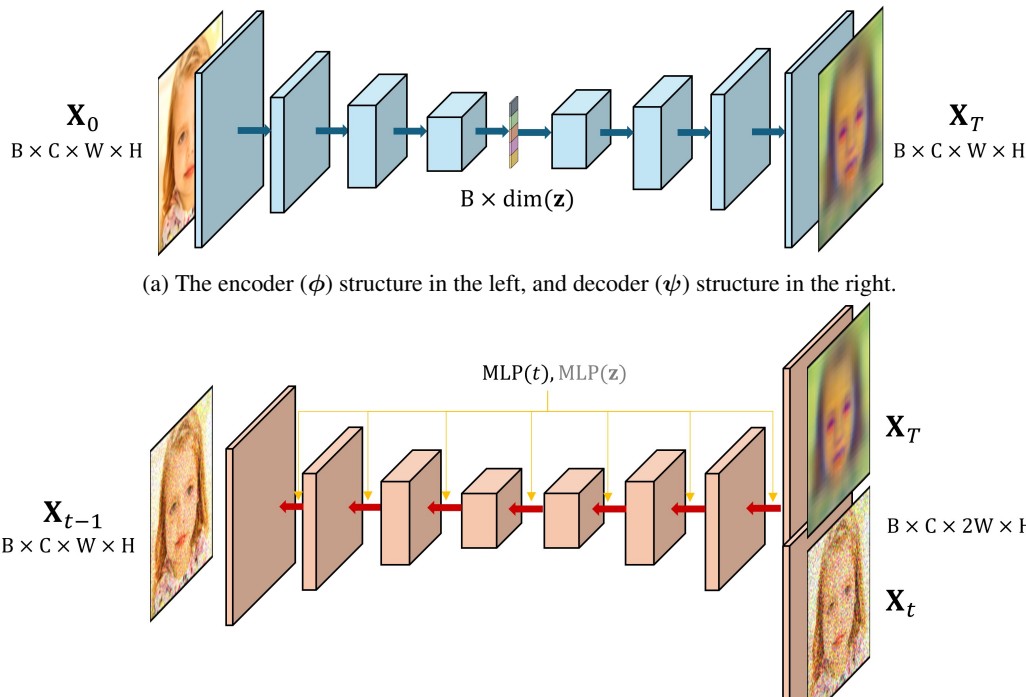

(a) The encoder ($\phi$) structure in the left, and decoder ($\psi$) structure in the right.

(b) The score network ($\boldsymbol{\theta}$) structure. While the model output is not directly one-step denoised sample $\mathbf{x}_t$, the output is equivalent to $\mathbf{x}_{t-1}$ with time-dependent constant operation with accessible information.

Figure 7: The architecture overview of Diffusion Bridge AutoEncoder.

Table 5: Network architecture and training configuration of DBAE.

| Parameter | CelebA 64 | FFHQ 128 | Horse 128 | Bedroom 128 |
|---|---|---|---|---|
| Base channels | 64 | 128 | 128 | 128 |
| Channel multipliers | [1,2,4,8] | [1,1,2,3,4] | [1,1,2,3,4] | [1,1,2,3,4] |
| Attention resolution | [16] | [16] | [16] | [16] |
| Encoder base ch | 64 | 128 | 128 | 128 |
| Enc. attn. resolution | [16] | [16] | [16] | [16] |
| Encoder ch. mult. | [1,2,4,8,8] | [1,1,2,3,4,4] | [1,1,2,3,4,4] | [1,1,2,3,4,4] |
| latent variable $\mathbf{z}$ dimension | 32, 256, 512 | 512 | 512 | 512 |
| Vanilla forward SDE | VP | VP | VP | VP |
| Images trained | 72M, 130M | 130M | 130M | 130M |
| Batch size | 128 | 128 | 128 | 128 |
| Learning rate | 1e-4 | 1e-4 | 1e-4 | 1e-4 |
| Optimizer | RAdam | RAdam | RAdam | RAdam |
| Weight decay | 0.0 | 0.0 | 0.0 | 0.0 |
| EMA rate | 0.9999 | 0.9999 | 0.9999 | 0.9999 |

Table 6: Network architecture and training configuration of latent diffusion models $p_{\boldsymbol{\omega}}(\mathbf{z})$ for an unconditional generation, following (Preechakul et al., 2022).

| Parameter | CelebA 64 | FFHQ 128 |
|---|---|---|
| Batch size | 512 | 256 |
| **z** trained | 600M | 600M |
| MLP layers ($N$) | 10, 15 | 10 |
| MLP hidden size | 2048 | |
| latent variable **z** dimension | 512 | |
| SDE | VP | |
| $\beta$ scheduler | Constant 0.008 | |
| Learning rate | 1e-4 | |
| Optimizer | AdamW (weight decay = 0.01) | |
| Train Diff $T$ | 1000 | |
| Diffusion loss | L1, L2 | |

**Reconstruction** We quantify reconstruction error in Table 2 though the Structural Similarity Index Measure (SSIM) (Wang et al., 2003), Learned Perceptual Image Patch Similarity (LPIPS) (Zhang et al., 2018) and Mean Squared Error (MSE). This metric measures the distance between original images in CelebA-HQ and their reconstructions across all 30K samples and averages them. SSIM compares the luminance, contrast, and structure between images to measure the differences on a scale from 0 to 1, like human visual perception. LPIPS measures the distance in the feature space of a neural network that learns the similarity between two images. We borrow the baseline results from DiffAE (Preechakul et al., 2022) and PDAE (Zhang et al., 2022). We use Heun's ODE sampler (99 NFE) to evaluate SSIM and MSE and use a stochastic sampler (Zhou et al., 2024) (998 NFE) to evaluate LPIPS for Table 2. We also present performance metrics for various NFE values in Tables 12 and 13.

**Disentanglment** The metric Total AUROC Difference (TAD) (Yeats et al., 2022) measures how effectively the latent space is disentangled, utilizing a dataset with multiple binary ground truth labels. It calculates the correlation between attributes based on the proportion of entropy reduction given any other single attribute. Attributes that show an entropy reduction greater than 0.2 when conditioned on another attribute are considered highly correlated and therefore entangled. For each remaining attribute that is not considered entangled, we calculate the AUROC score for each dimension of the latent variable **z**. To calculate the AUROC score, first determine the dimension-wise minimum and maximum values of **z**. We increment the threshold from the minimum to the maximum for each dimension, converting **z** to a one-hot vector by comparing each dimension's value against the threshold. This one-hot vector is then compared to the true labels to compute the AUROC score. An attribute is considered disentangled if at least one dimension of **z** can detect it with an AUROC score of 0.75 or higher. The sub-metric ATTRS denotes the number of such captured attributes. The TAD score is calculated as the sum of the differences between the two highest AUROC scores for each captured attribute. We randomly selected 1000 samples from the CelebA training, validation, and test sets to perform the measurement following (Yeats et al., 2022). We borrow the baseline results expect DisDiff from the InfoDiffusion (Wang et al., 2023), and we follow their setting that the $\dim(\mathbf{z}) = 32$. DisDiff (Yang et al., 2023) utilizes the $\dim(\mathbf{z}) = 192$ and we borrow its performance from the original paper. We use evaluation code from https://github.com/ericyeats/nashae-beamsynthesis.

**Unconditional Generation** To measure unconditional generative modeling, we quantify Precision, Recall (Kynkäänniemi et al., 2019), Inception Score (IS) (Salimans et al., 2016) and the Fréchet Inception Distance (FID) (Heusel et al., 2017). Precision and Recall are measured by 10k real images and 10k generated images following (Dhariwal & Nichol, 2021). Precision is the ratio of generated images belonging to real images' manifold. Recall is the ratio of real images belonging to the generated images' manifold. The manifold is constructed in a pre-trained feature space using the nearest neighborhoods. Precision quantifies sample fidelity, and Recall quantifies sample diversity. Both IS and FID are influenced by fidelity and diversity. IS is calculated using an Inception Network (Szegedy et al., 2016) pre-trained on ImageNet (Russakovsky et al., 2015), and it computes the logits for generated samples. If an instance is predicted with high confidence for a specific class,

and predictions are made for multiple classes across all samples, then the IS will be high. On the other hand, for samples generated from FFHQ or CelebA, predictions cannot be made for multiple classes, which does not allow for diversity to be reflected. Therefore, a good Inception Score (IS) can only result from high-confidence predictions based solely on sample fidelity. We measure IS for 10k generated samples. FID approximates the generated and real samples as Gaussians in the feature space of an Inception Network and measures the Wasserstein distance between them. Since it measures the distance between distributions, it emphasizes the importance of sample diversity and sample fidelity. For Table 4 we measure FID between 50k random samples from the FFHQ dataset and 50k randomly generated samples. For 'AE', we measure the FID between 50k random samples from the FFHQ dataset and generate samples that reconstruct the other 50k random samples from FFHQ. In Table 3, we measure the FID between 10k random samples from the CelebA and 10k randomly generated samples. We utilize `https://github.com/openai/guided-diffusion` to measure Precision, Recall and IS. We utilize `https://github.com/GaParmar/clean-fid` to measure FID. In Table 4, we loaded checkpoints for all baselines (except the generative prior of PDAE, we train it to fill performance) and conducted evaluations in the same NFEs. Table 14 shows the performance under various NFEs. For CelebA training, we use a $\dim(\mathbf{z}) = 256$ following (Wang et al., 2023), while FFHQ training employs a $\dim(\mathbf{z}) = 512$ following (Preechakul et al., 2022; Zhang et al., 2022).

## C.3 ALGORITHM

This section presents the training and utilization algorithms of DBAE. Algorithm 1 outlines the procedure for minimizing the autoencoding objective, $\mathcal{L}_{\text{AE}}$. Algorithm 2 explains the method for reconstruction using the trained DBAE. Algorithm 3 describes the steps for training the generative prior, $p_{\boldsymbol{\omega}}$. Algorithm 4 explains the procedure for unconditional generation using the trained DBAE and generative prior.

---

**Algorithm 3:** Latent DPM Training Algorithm

**Input:** $\text{Enc}_{\phi}$, data distribution $q_{\text{data}}(\mathbf{x}_0)$, drift term $\mathbf{f}$, volatility term $g$
**Output:** Latent DPM score network $\mathbf{s}_{\boldsymbol{\omega}}$
**while** *not converges* **do**
    Sample time $t$ from $[0, T]$
    $\mathbf{x}_0 \sim q_{\text{data}}(\mathbf{x}_0)$
    $\mathbf{z} = \text{Enc}_{\phi}(\mathbf{x}_0)$
    $\mathbf{z}_t \sim \tilde{q}_t(\mathbf{z}_t | \mathbf{z}_0)$
    $\mathcal{L} \leftarrow g^2(t) || \mathbf{s}_{\boldsymbol{\omega}}(\mathbf{z}_t, t) - \nabla_{\mathbf{z}_t} \log p_t(\mathbf{z}_t | \mathbf{z}) ||_2^2$
    Update $\boldsymbol{\omega}$ by $\mathcal{L}$ using the gradient descent method
**end**

---

**Algorithm 4:** Unconditional Generation Algorithm

**Input:** $\text{Dec}_{\psi}$, latent score network $\mathbf{s}_{\boldsymbol{\omega}}$, score network $\mathbf{s}_{\boldsymbol{\theta}}$, latent discretized time steps $\{t_j^*\}_{j=0}^{N_{\mathbf{z}}}$, discretized
       time steps $\{t_i\}_{i=0}^{N}$
$\mathbf{z}_T \sim \mathcal{N}(\mathbf{0}, \mathbf{I})$
**for** $j = N_{\mathbf{z}}, ..., 1$ **do**
    Update $\mathbf{z}_{t_j}$ using Eq. (3)
$\mathbf{x}_T = \text{Dec}_{\psi}(\mathbf{z}_0)$
**for** $i = N, ..., 1$ **do**
    Update $\mathbf{x}_{t_i}$ using Eq. (12)
**Output:** Unconditioned sample $\mathbf{x}_0$

---

## C.4 COMPUTATIONAL COST

This section presents a computational cost comparison among diffusion-based representation learning baselines. Table 7 compares DDIM (Song et al., 2021a), DiffAE (Preechakul et al., 2022), PDAE (Zhang et al., 2022), and DBAE in terms of parameter size, training time, and testing time. DDIM requires only a score network (99M), resulting in minimal parameter size. DiffAE involves

Table 7: Computational cost comparison for FFHQ128. Training time is measured in milliseconds per image per NVIDIA A100 (ms/img/A100), and testing time is reported in milliseconds per one sampling step per NVIDIA A100 (ms/one sampling step/A100).

|  | Parameter Size | Training | Testing |
|---|---|---|---|
| DDIM (Song et al., 2021a) | 99M | 9.687 | 0.997 |
| DiffAE (Preechakul et al., 2022) | 129M | 12.088 | 1.059 |
| PDAE (Zhang et al., 2022) | 280M | 12.163 | 1.375 |
| DBAE | 161M | 13.190 | 1.024 |

Table 8: Computing costs for $\mathbf{x}_T$ inference.

| Method | NFE ($\downarrow$) | | | Total time ($\downarrow$) (ms) |
|---|---|---|---|---|
|  | $\text{Enc}_\phi$ | $\text{Dec}_\psi$ | $\mathbf{s}_\theta$ |  |
| PDAE | 1 | 500 | 500 | 688 |
| DiffAE | 1 | - | 250 | 265 |
| DBAE | 1 | 1 | 0 | 0.31 |

a $\mathbf{z}$-conditional score network (105M) and an encoder (24M), leading to an increase in parameter size. PDAE incorporates both a heavy decoder and an encoder, further increasing the parameter size. Conversely, although DBAE also includes a decoder, it is less complex (32M), resulting in a smaller relative increase in parameter size compared to PDAE. From a training time perspective, DiffAE, PDAE, and DBAE all require longer durations compared to DDIM due to their increased model sizes. DBAE's training time is 9% longer than that of DiffAE because of the decoder module. However, the decoder does not repeatedly affect the sampling time, making it similar to DiffAE's. In contrast, PDAE, which utilizes a decoder at every sampling step, has a longer sampling time.

# D    ADDITIONAL EXPERIMENTS

## D.1    DOWNSTREAM INFERENCE

Figure 8 shows the attribute-wise Average Precision (AP) gap between PDAE (Zhang et al., 2022) and DBAE. As discussed in Section 5.1, PDAE suffers from an *information split problem* that $\mathbf{x}_T$ contains facial or hair details. The resulting attribute-wise gain aligns with that analysis with Figure 3. Figure 9d shows the absolute attribute-wise AP of DBAE performance across the training setting varies on the encoder (deterministic/stochastic) and training datasets (CelebA training set / FFHQ). The attribute-wise performance is similar across the training configurations. Table 9 shows the comparsion to the other baseline DiffuseVAE (Pandey et al., 2022). From the two-stage paradigm of DiffuseVAE, its latent quality is only from the latent representation capability of the VAE module. This is an aligned result from the poor performance of $\beta$-TCVAE in Table 1. Note that the image crop for CelebA in DiffuseVAE is not exactly the same as our setting.

Table 9: Linear-probe attribute prediction quality comparison for models trained on CelebA and CIFAR-10 with dim($\mathbf{z}$) = 512. The best and second-best results are highlighted in **bold**. We evaluate 5 times and report the average.

| Method | CelebA | | | CIFAR-10 |
|---|---|---|---|---|
|  | AP ($\uparrow$) | Pearson's r ($\uparrow$) | MSE ($\downarrow$) | AUROC ($\uparrow$) |
| DiffuseVAE (Pandey et al., 2022) | 0.395 | 0.325 | 0.618 | 0.736 |
| DBAE | **0.655** | **0.643** | **0.369** | **0.836** |

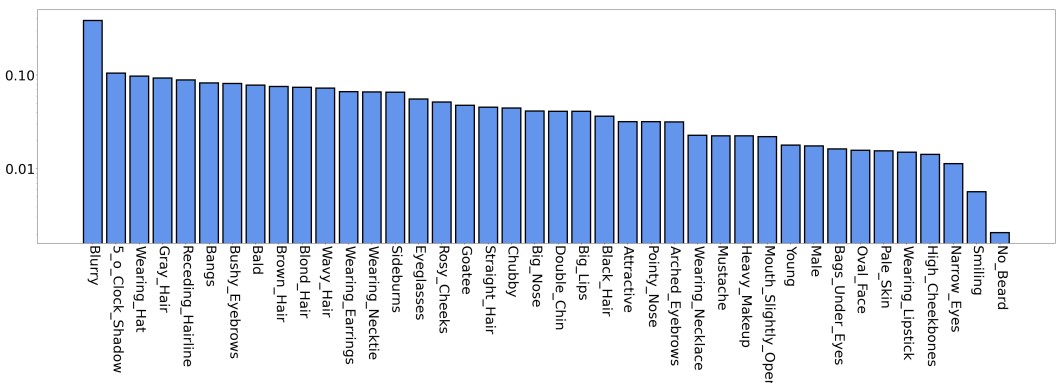

Figure 8: Attribute-wise AP gap between PDAE and DBAE-d trained on CelebA. DBAE-d performs better for all 40 attributes.

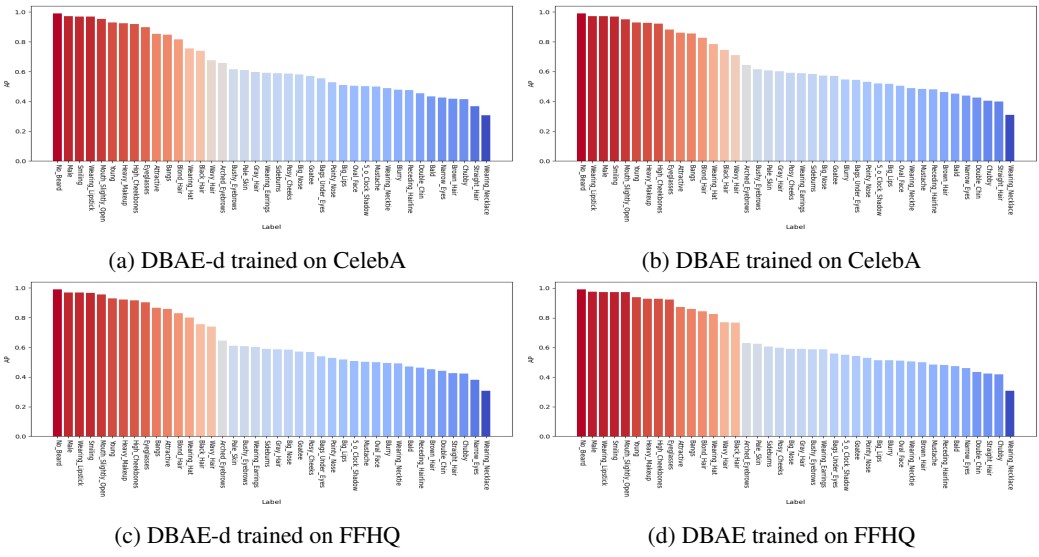

(a) DBAE-d trained on CelebA

(b) DBAE trained on CelebA

(c) DBAE-d trained on FFHQ

(d) DBAE trained on FFHQ

Figure 9: Attribute-wise Average Precision across the training configuration of DBAE.

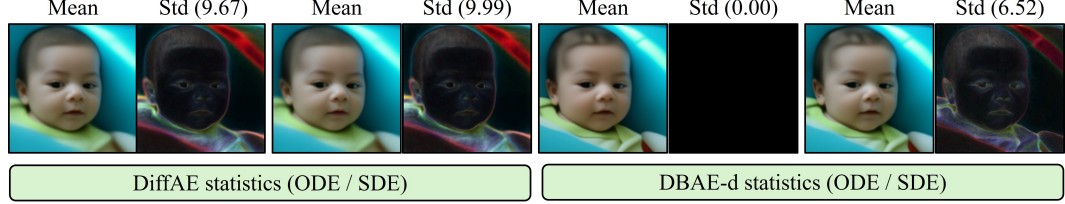

Figure 10: Reconstruction statistics with inferred **z**. We quantify the mean and standard deviation of the reconstruction in the pixel space. The number in parentheses represents the dimension-wise averaged standard deviation in the pixel space.

## D.2 RECONSTRUCTION

The sampling step is important for practical applications (Lu et al., 2022; Zheng et al., 2024). We compare the reconstruction results across various sampling steps among the baselines. Tables 12 and 13 shows the results. The proposed model performs the best results among all NFEs in $(10, 20, 50, 100)$. We borrow the performance of DDIM, DiffAE from (Preechakul et al., 2022). We manually measure for PDAE (Zhang et al., 2022) using an official checkpoint in `https://github.com/ckczzj/PDAE`. Figure 10 shows the reconstruction statistics for a single image with inferred **z**. Due to the information split on $\mathbf{x}_T$, DiffAE shows substantial variations even when utilizing ODE sampling. When DBAE also performs stochastic sampling, information is split across the sampling path, but it has less variation compared to DiffAE (9.99 vs 6.52), and DBAE induce information can be stored solely at $\mathbf{x}_T$ through the ODE path. Table 10 shows that the reconstruction quality compare to DiffuseVAE (Pandey et al., 2022). Since DiffuseVAE also requires to sample random $\mathbf{x}_T$ for the generation, this framework also suffers from *information split problem*. That is the reason for poor reconstruction quality. Table 11 shows the reconstruction quality for Horse and Bedroom datasets, which surpasses the DiffAE.

Table 10: Autoencoding reconstruction quality comparison with DiffuseVAE with 512-dimensional latent variable, the one yielding the best performance is highlighted in **bold**.

|  | **CelebA** | | |
| Method | SSIM ($\uparrow$) | LPIPS ($\downarrow$) | MSE ($\downarrow$) |
| --- | --- | --- | --- |
| DiffuseVAE (Pandey et al., 2022) | 0.836 | 0.134 | 0.018 |
| DBAE | **0.990** | **0.014** | **4.86e-4** |

Table 11: More results on autoencoding reconstruction quality comparison with DiffAE with 512-dimensional latent variable, the one yielding the best performance is highlighted in **bold**.

|  | **Horse** | | **Bedroom** | |
| Method | SSIM ($\uparrow$) | MSE ($\downarrow$) | SSIM ($\uparrow$) | MSE ($\downarrow$) |
| --- | --- | --- | --- | --- |
| DiffAE (Preechakul et al., 2022) | 0.857 | 0.025 | 0.910 | 0.017 |
| DBAE | **0.902** | **0.012** | **0.948** | **0.007** |

## D.3 UNCONDITIONAL GENERATION

The sampling step is also important for unconditional generation (Lu et al., 2022; Zheng et al., 2024). We reduce the NFE=1000 in Table 4 to NFE=500 and NFE=250 in Table 14. As the number of function evaluations (NFE) decreased, DDPM (Ho et al., 2020) showed a significant drop in performance, while DBAE and the other baselines maintained a similar performance trend.

Although DBAE improves sample fidelity which is crucial for practical uses, sample diversity remains an important virtue depending on the specific application scenarios (Kim et al., 2024; Sadat et al., 2024). In the area of generative models, there is a trade-off between fidelity and diversity (Dhariwal

Table 12: Autoencoding reconstruction quality comparison. All the methods are trained on the FFHQ dataset and evaluated on the 30K CelebA-HQ dataset. Among tractable and compact 512-dimensional latent variable models, the one yielding the best performance was highlighted in **bold**, followed by an underline for the next best performer. All the metric is SSIM.

| Method | Tractability | Latent dim ($\downarrow$) | NFE=10 | NFE=20 | NFE=50 | NFE=100 |
|---|---|---|---|---|---|---|
| DDIM (Inferred $\mathbf{x}_T$) (Song et al., 2021a) | ✗ | 49,152 | 0.600 | 0.760 | 0.878 | 0.917 |
| DiffAE (Inferred $\mathbf{x}_T$) (Preechakul et al., 2022) | ✗ | 49,664 | 0.827 | 0.927 | 0.978 | 0.991 |
| PDAE (Inferred $\mathbf{x}_T$) (Zhang et al., 2022) | ✗ | 49,664 | 0.822 | 0.901 | 0.966 | 0.987 |
| DiffAE (Random $\mathbf{x}_T$) (Preechakul et al., 2022) | ✓ | 512 | 0.707 | 0.695 | 0.683 | 0.677 |
| PDAE (Random $\mathbf{x}_T$) (Zhang et al., 2022) | ✓ | 512 | 0.728 | 0.713 | 0.697 | 0.689 |
| DBAE | ✓ | 512 | **0.904** | 0.909 | 0.916 | 0.920 |
| DBAE-d | ✓ | 512 | 0.884 | **0.920** | **0.945** | **0.954** |

Table 13: Autoencoding reconstruction quality comparison. All the methods are trained on the FFHQ dataset and evaluated on the 30K CelebA-HQ dataset. Among tractable and compact 512-dimensional latent variable models, the one yielding the best performance was highlighted in **bold**, followed by an underline for the next best performer. All the metric is MSE.

| Method | Tractability | Latent dim ($\downarrow$) | NFE=10 | NFE=20 | NFE=50 | NFE=100 |
|---|---|---|---|---|---|---|
| DDIM (Inferred $\mathbf{x}_T$) (Song et al., 2021a) | ✗ | 49,152 | 0.019 | 0.008 | 0.003 | 0.002 |
| DiffAE (Inferred $\mathbf{x}_T$) (Preechakul et al., 2022) | ✗ | 49,664 | 0.001 | 0.001 | 0.000 | 0.000 |
| PDAE (Inferred $\mathbf{x}_T$) (Zhang et al., 2022) | ✗ | 49,664 | 0.001 | 0.001 | 0.000 | 0.000 |
| DiffAE (Random $\mathbf{x}_T$) (Preechakul et al., 2022) | ✓ | 512 | 0.006 | 0.007 | 0.007 | 0.007 |
| PDAE (Random $\mathbf{x}_T$) (Zhang et al., 2022) | ✓ | 512 | **0.004** | 0.005 | 0.005 | 0.005 |
| DBAE | ✓ | 512 | 0.005 | 0.005 | 0.005 | 0.005 |
| DBAE-d | ✓ | 512 | 0.006 | **0.003** | **0.002** | **0.002** |

& Nichol, 2021). Therefore, providing a balance between these two virtues is important. We offer an option based on DBAE. The $h$-transformed forward SDE we designed in Eq. (10) is governed by the determination of the endpoint distribution. If we set endpoint distribution as Eq. (124), we can achieve smooth transitions between DiffAE and DBAE in terms of $\mathbf{x}_T$ distribution. Modeling $q_{\phi,\psi}(\mathbf{x}_T|\mathbf{x}_0)$ as a Gaussian distribution (with learnable mean and covariance) with a certain variance or higher can also be considered as an indirect approach.

$$\mathbf{x}_T \sim \lambda \times q_{\phi,\psi}(\mathbf{x}_T|\mathbf{x}_0) + (1-\lambda) \times \mathcal{N}(\mathbf{0}, \mathbf{I}) \tag{124}$$

Table 14: Unconditional generation with reduced NFE $\in \{250, 500\}$ on FFHQ. '+AE' indicates the use of the inferred distribution $q_\phi(\mathbf{z})$ instead of $p_\omega(\mathbf{z})$

| Method | NFE = 500 | | | | NFE = 250 | | | |
|---|---|---|---|---|---|---|---|---|
| | Prec ($\uparrow$) | IS ($\uparrow$) | FID 50k ($\downarrow$) | Rec ($\uparrow$) | Prec ($\uparrow$) | IS ($\uparrow$) | FID 50k ($\downarrow$) | Rec ($\uparrow$) |
| DDIM (Song et al., 2021a) | 0.705 | 3.16 | 11.33 | 0.439 | 0.706 | 3.16 | 11.48 | **0.453** |
| DDPM (Ho et al., 2020) | 0.589 | 2.92 | 22.10 | 0.251 | 0.390 | 2.76 | 39.55 | 0.093 |
| DiffAE (Preechakul et al., 2022) | 0.755 | 2.98 | **9.71** | **0.451** | 0.755 | 3.04 | **10.24** | 0.443 |
| PDAE (Zhang et al., 2022) | 0.687 | 2.24 | 46.67 | 0.175 | 0.709 | 2.25 | 44.82 | 0.189 |
| DBAE | **0.774** | **3.91** | 11.71 | 0.391 | **0.758** | **3.90** | 13.88 | 0.381 |
| DiffAE+AE | **0.750** | **3.61** | 3.21 | 0.689 | **0.750** | **3.61** | 3.87 | 0.666 |
| PDAE+AE | 0.710 | 3.53 | 7.11 | 0.598 | 0.721 | 3.54 | 6.58 | 0.608 |
| DBAE+AE | 0.748 | 3.57 | **1.99** | **0.702** | 0.731 | 3.58 | **3.36** | **0.694** |

## D.4 Results with Intel Gaudi v2 hardware.

We conducted evaluations across various infrastructures to assess experimental reproducibility. The performance of the trained model (DBAE-d) was evaluated on both the Nvidia A100 and Intel Gaudi v2 chips. The reconstruction results for both chips are presented in Table 15. Reconstruction performance on each chip was assessed using various metrics, revealing negligible errors across all metrics. To facilitate reproducibility, we provide the code at `https://github.com/NAVER-INTEL-Co-Lab/gaudi-dbae` for reproducing our experiments on Intel Gaudi v2 chips.

Table 15: Regenerated results of Table 2 across multiple hardwares.

| Hardware | SSIM ($\uparrow$) | LPIPS ($\downarrow$) | MSE ($\downarrow$) |
|---|---|---|---|
| Nvidia A100 | 0.953 | 0.072 | 2.49e-3 |
| Intel Gaudi v2 | 0.956 | 0.073 | 2.47e-3 |

## D.5 Additional Samples

**Interpolation** Figures 11 and 12 shows the interpolation results of DBAE trained on FFHQ, Horse, and Bedroom. The two paired rows indicate the endpoints $x_T$ and generated image $x_0$ each. Figure 13 compares the interpolation results with PDAE (Zhang et al., 2022) and DiffAE (Preechakul et al., 2022) under tractable inference condition. PDAE and DiffAE result in unnatural interpolations without inferring $x_T$, compared to DBAE.

**Attribute Manipulation** Figure 15 shows additional manipulation results using a linear classifier, including multiple attributes editing on a single image. Figure 14 provides the variations in the manipulation method within DBAE. The top row utilizes the manipulated $x_T$ both for the starting point of the generative process and score network condition input. The bottom row utilizes the manipulated $x_T$ only for the score network condition input, while the starting point remains the original image's $x_T$. Using manipulated $x_T$ both for starting and conditioning results in more dramatic editing, and we expect to be able to adjust this according to the user's desires.

**Generation Trajectory** Figure 16 shows the sampling trajectory of DBAE from $x_T$ to $x_0$ with stochastic sampling for FFHQ, Horse, and Bedroom.

**Unconditional Generation** Figures 17 and 18 show the randomly generated uncurated samples from DBAE for FFHQ and CelebA.

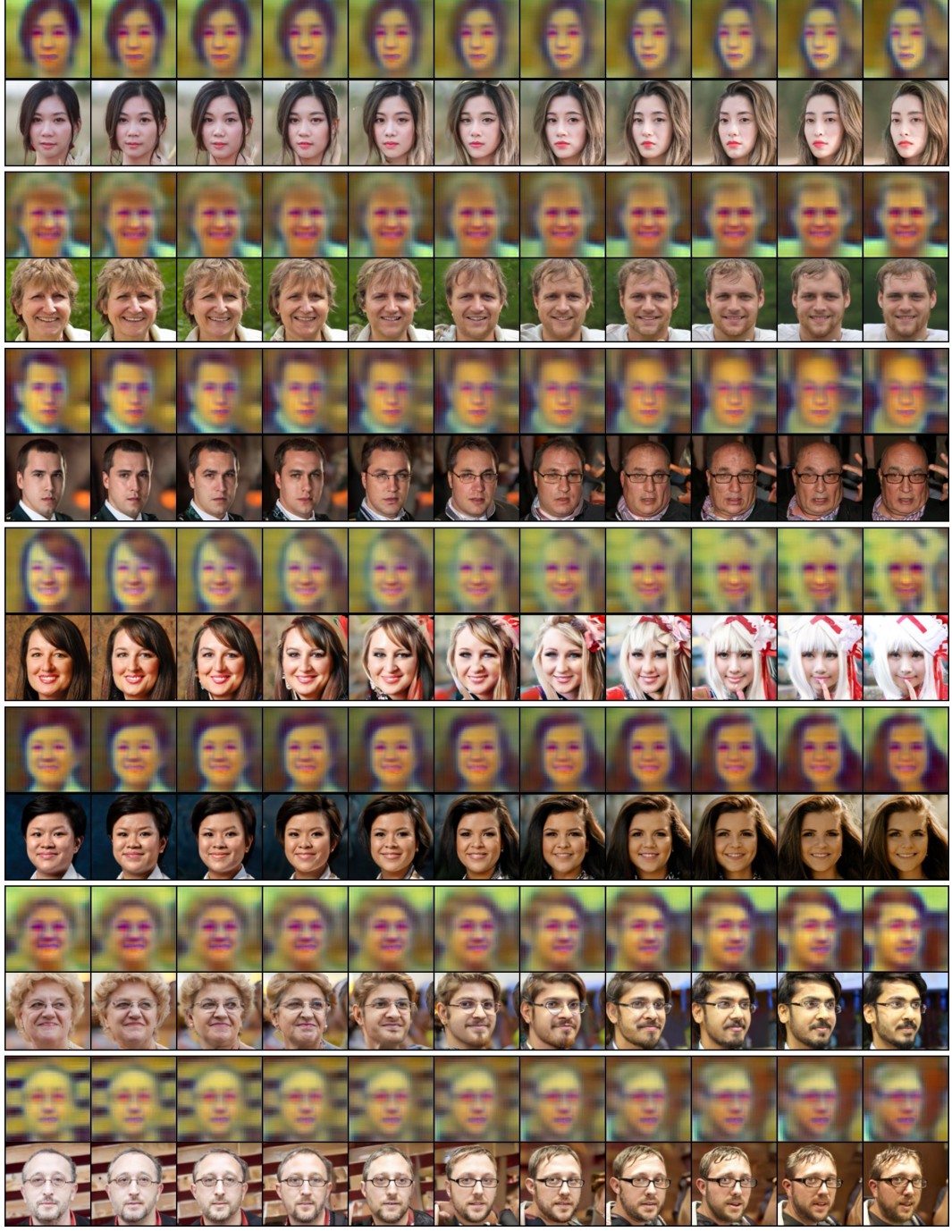

Figure 11: FFHQ interpolations results with corresponding endpoints $\mathbf{x}_T$. The leftmost and rightmost images are real images.

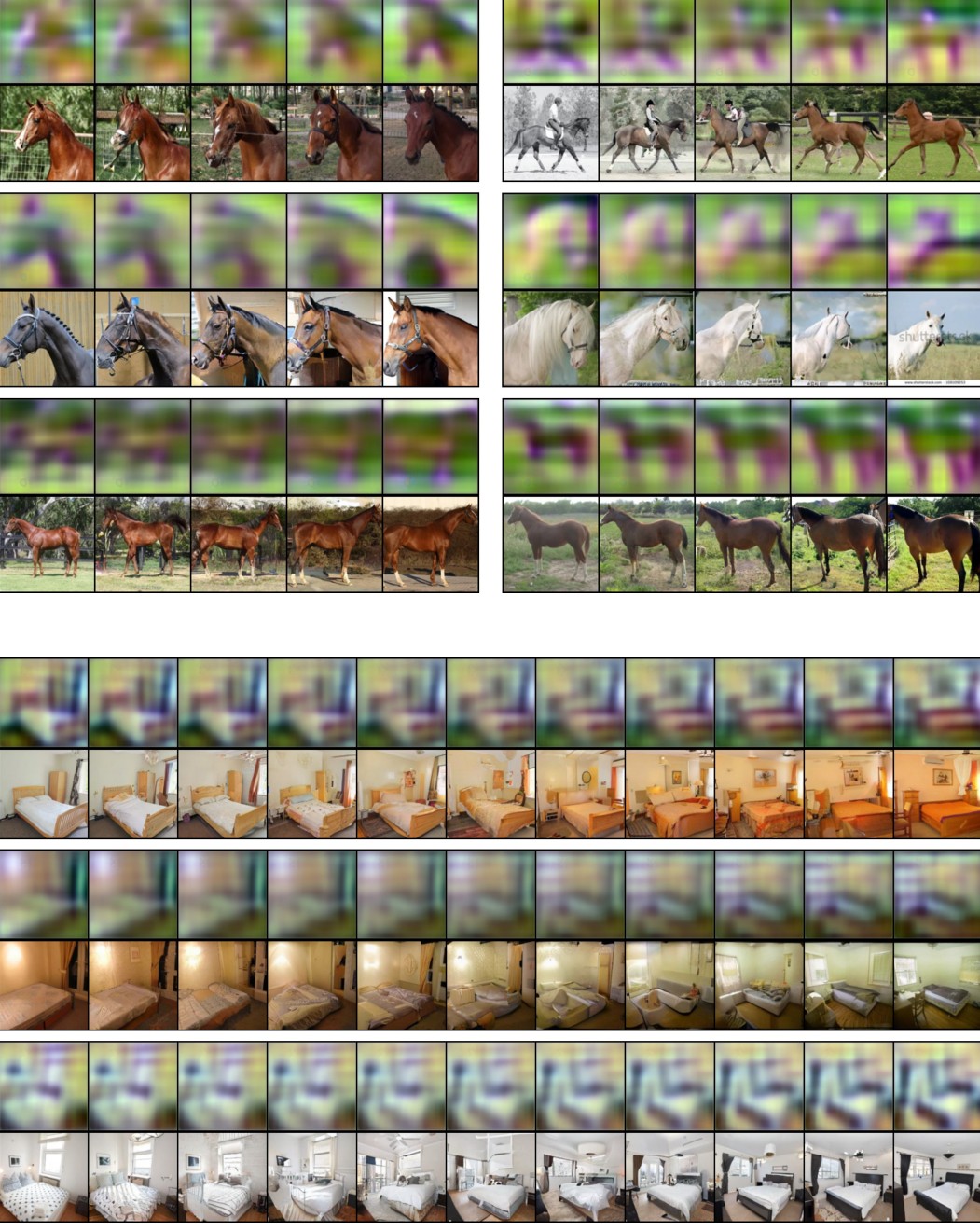

Figure 12: Horse and Bedroom interpolations results with corresponding endpoints $\mathbf{x}_T$. The leftmost and rightmost images are real images.

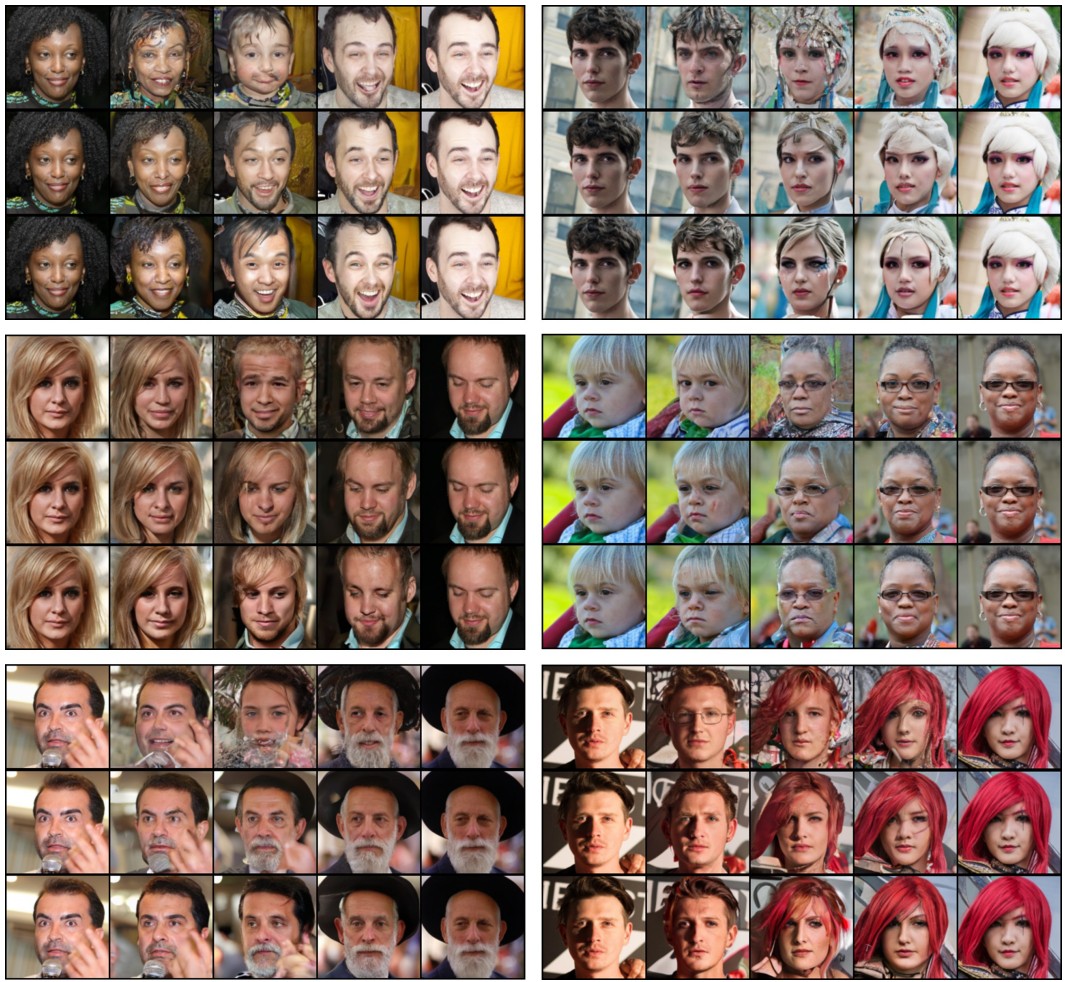

Figure 13: FFHQ interpolation comparison: PDAE (Zhang et al., 2022) (top), DiffAE (Preechakul et al., 2022) (middle) and DBAE (bottom).

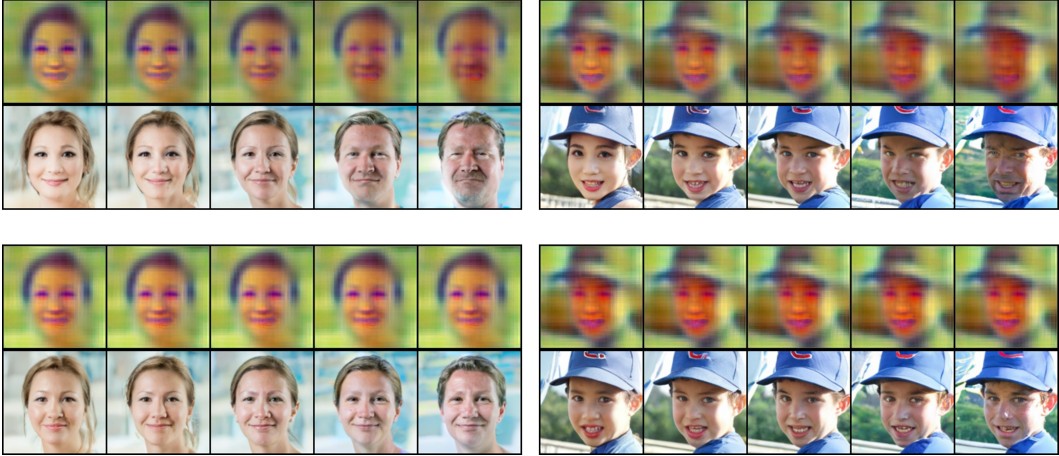

Figure 14: Attribute manipulation on FFHQ using a linear classifier and corresponding endpoints $\mathbf{x}_T$. The top results utilize the manipulated $\mathbf{x}_T$ both as the starting point of the sampling trajectory and as a condition input to the score network. The bottom results use the manipulated $\mathbf{x}_T$ solely as the condition input and maintain the original $\mathbf{x}_T$ as the starting point of the sampling trajectory. All the middle images are the original images.

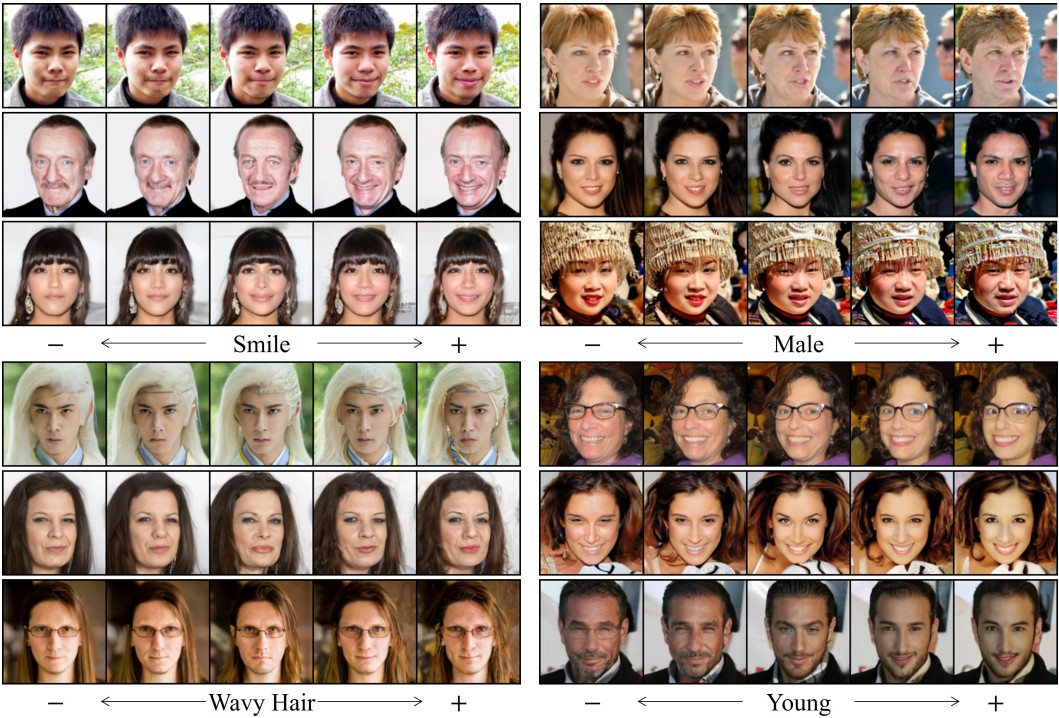

(a) Smooth traversals in the direction of attribute manipulation. All the middle images are the original images.

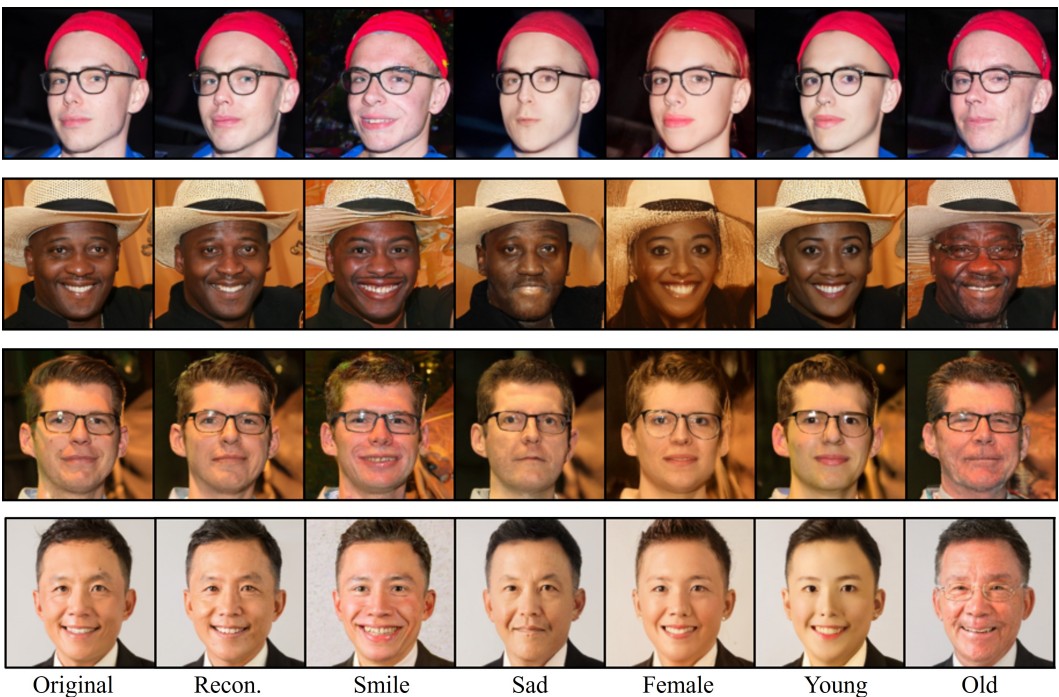

| Original | Recon. | Smile | Sad | Female | Young | Old |

(b) Multiple attribute manipulation on a single image.

Figure 15: Attribute manipulation using a linear classifier on FFHQ and CelebA-HQ.

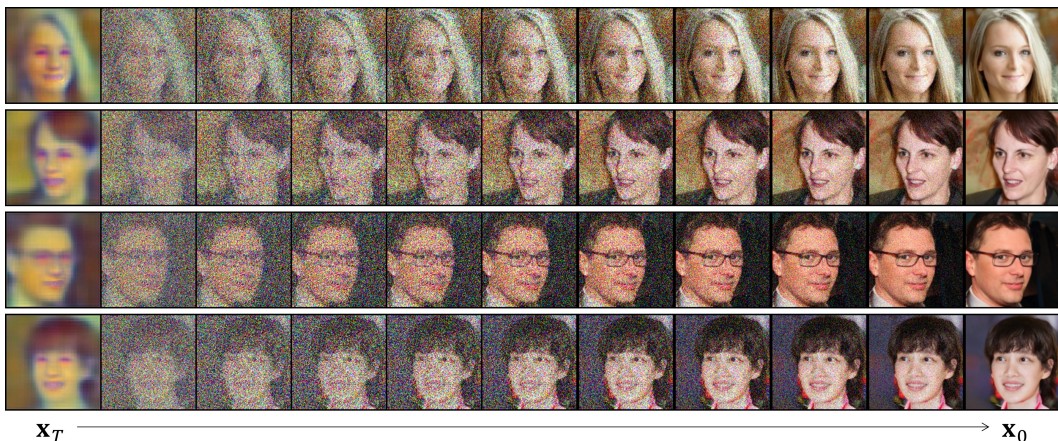

(a) Sampling trajectory of DBAE trained on FFHQ.

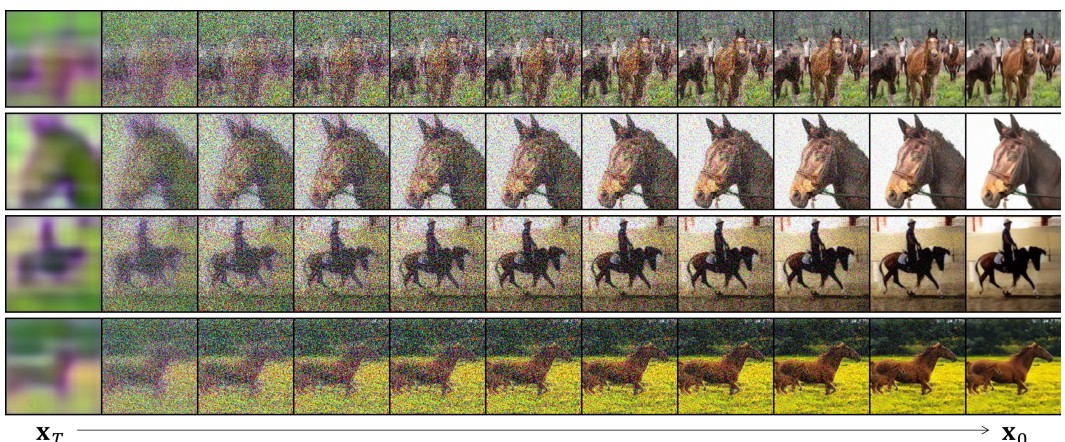

(b) Sampling trajectory of DBAE trained on Horse.

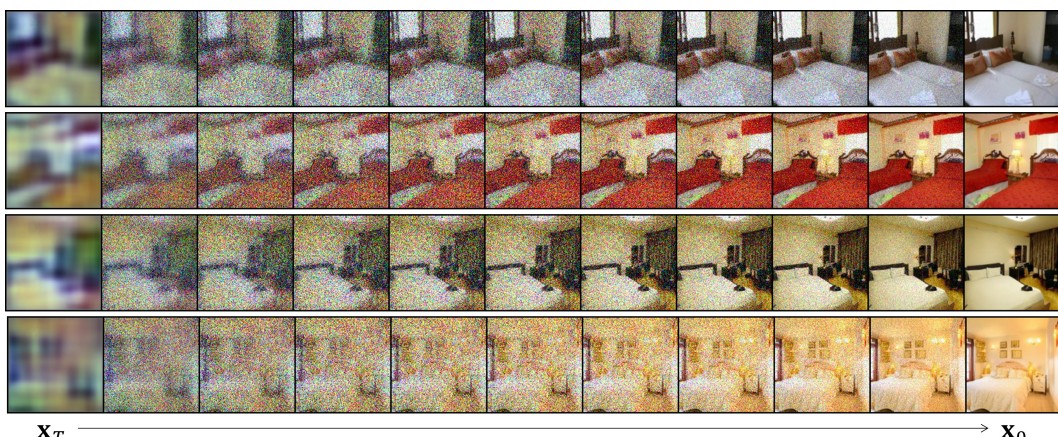

(c) Sampling trajectory of DBAE trained on Bedroom.

Figure 16: Stochastic sampling trajectory of DBAE trained on various datasets.

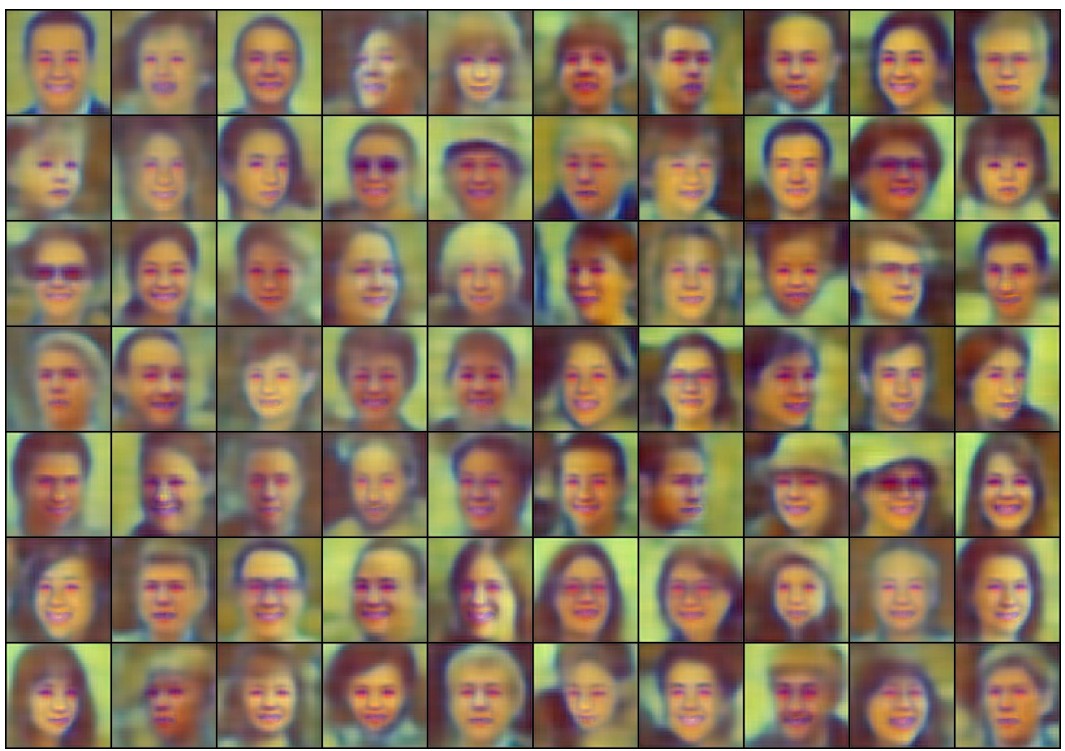

(a) Generated endpoints $\mathbf{x}_T$

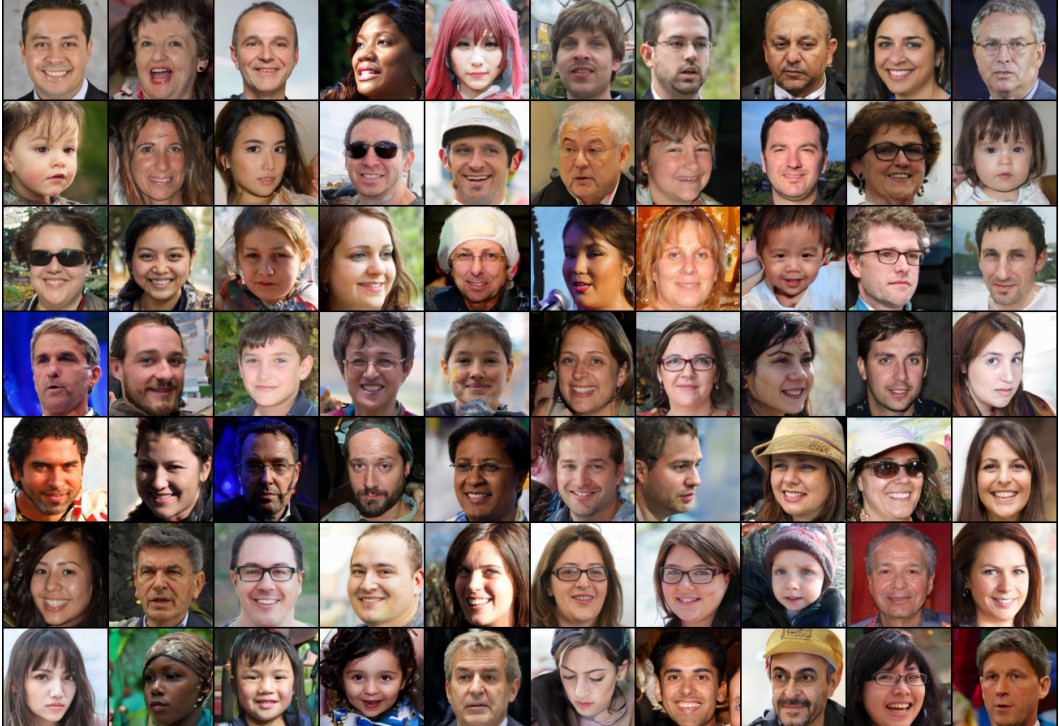

(b) Generated images $\mathbf{x}_0$

Figure 17: Uncurated generated samples with corresponding endpoints from DBAE trained on FFHQ with unconditional generation.

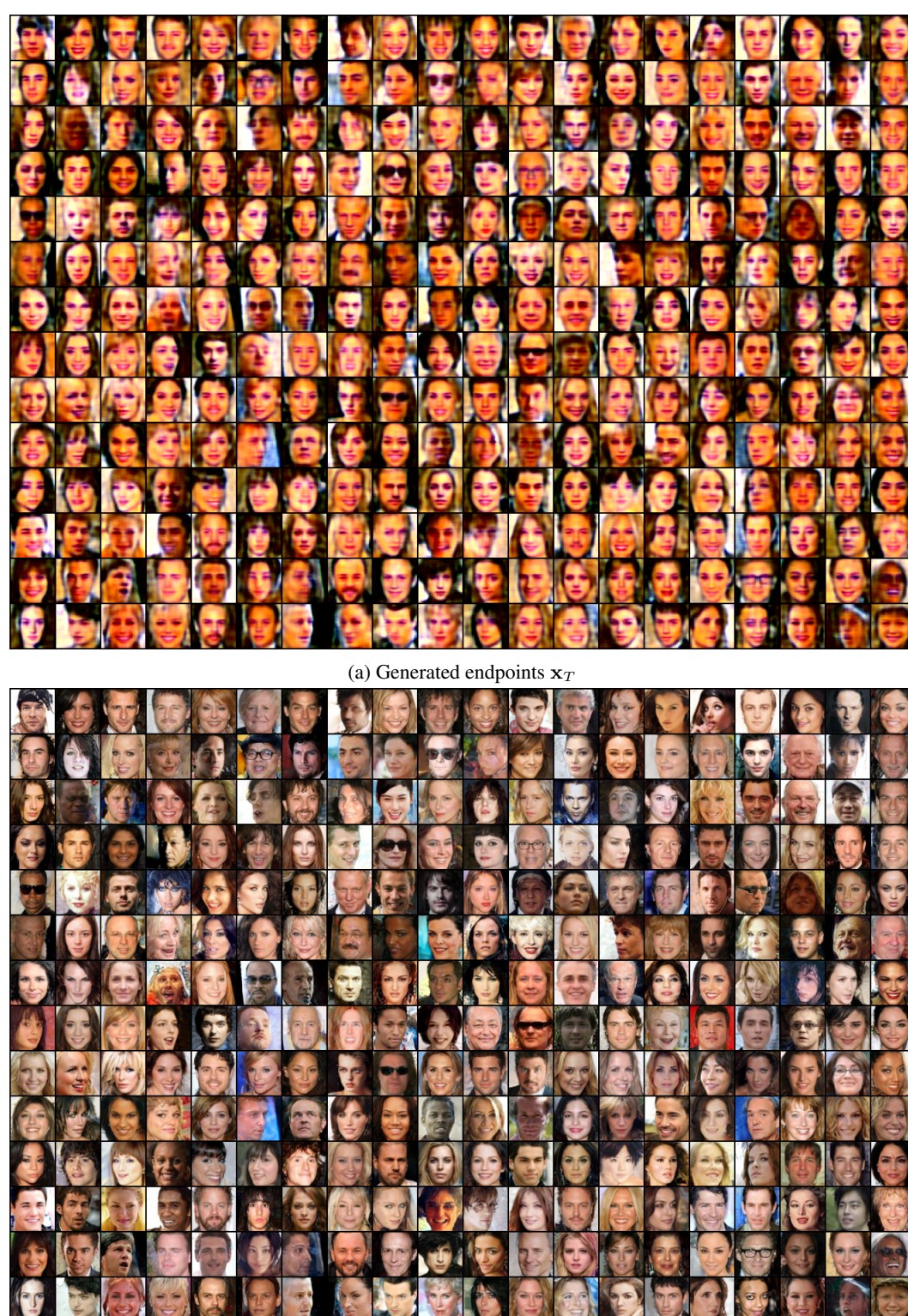

(a) Generated endpoints $\mathbf{x}_T$

(b) Generated images $\mathbf{x}_0$

Figure 18: Uncurated generated samples with corresponding endpoints from DBAE trained on CelebA with unconditional generation.

