# OpenReview forum: "Diffusion Bridge AutoEncoders for Unsupervised Representation Learning"
_ICLR.cc/2025/Conference — ICLR 2025 Spotlight_

### Official Review · Reviewer_rm4h · 2024-11-02

**Soundness:** 2
**Presentation:** 3
**Contribution:** 2
**Rating:** 5
**Confidence:** 3

**Summary:**

This paper introduces the Diffusion Bridge AutoEncoders (DBAE) framework to address the "information split problem" in diffusion-based autoencoding models for unsupervised representation learning. The authors argue that in conventional latent-variable-augmented diffusion models, the information in data $x_0$ is split between two latent variables--$z$, obtained from an auxiliary encoder, and $x_T$, the endpoint of the diffusion process. This split hinders effective encoding, especially for downstream tasks such as reconstruction. DBAE proposes a solution to remedy this by creating a dependency of $x_T$ on $z$, allowing $z$ to serve as a comprehensive information bottleneck and improving representation quality. Specifically, the authors propose the process to encode $x_0$ into $z$, decode $z$ into $x_T$, and then bridge $x_0$ and $x_T$ to define a diffusion process. The authors provide empirical evidence supporting DBAE’s improvements in representation quality, reconstruction, and downstream tasks, suggesting it as a promising model for generative and reconstruction purposes.

**Strengths:**

This paper addresses a key challenge in latent-variable generative models, particularly in diffusion probabilistic models (DPMs) that are augmented with an auxiliary encoded latent variable $z$, which the authors refer to as the "information split problem." The authors clearly illustrate how the information split between $z$ and $x_T$ diminishes the quality of latent representations for high-fidelity tasks. To overcome this issue, they propose a principled framework that integrates autoencoders with diffusion models by (1) encoding $x_0$ to define the latent endpoint $x_T$ of the diffusion process, and (2) connecting these two points utilizing Doob’s $h$-transform. This approach combines the strengths of autoencoders—such as dimensionality reduction and efficient encoding/decoding—with the rich generative capabilities of diffusion models. Experimental results demonstrate that DBAE outperforms existing models across various benchmarks, underscoring its effectiveness in tasks like generation, reconstruction, and attribute manipulation. The paper also discusses potential extensions of DBAE to various downstream applications, highlighting its adaptability and potential for broader impact.

**Weaknesses:**

Despite its strengths, I think the paper’s impact is significantly diminished by its presentation. Below, I outline major concerns with suggestions for improvement.

**Clarifying Motivation and Significance in the Introduction:**
The Introduction lacks a clear articulation of why the “information split problem” is critical to address. This issue is specific to a subset of diffusion probabilistic models (DPMs) that are augmented by an auxiliary encoded latent variable $z$. For readers to appreciate the significance of this issue, it would help to emphasize the relevance of these models, which combine strengths from both VAEs and DPMs. While the first two paragraphs provide an overview of latent-variable generative modeling, the current description introduces these models merely as proposed approaches rather than as leading techniques in the field. Highlighting their role as state-of-the-art or best-performing models would better contextualize the importance of tackling their limitations. Additionally, the Introduction could clarify why the information split problem is particularly relevant for tasks requiring accurate reconstruction of the original data, $x_0$. The authors might adapt statements like those in Lines 285–286—“For downstream inference, attribute manipulation, and interpolation, the model requires reconstruction capability”—to illustrate the utility of generative modeling and explain the advantages of integrating VAE and DPM functionalities. Establishing these goals clearly would help readers understand how the information split issue primarily affects reconstruction and how DBAE addresses this challenge.

**Improving Precision in Terminology and Expression:**
Several terms and expressions in the paper would benefit from clearer definitions and consistent use. For instance, in Line 49, “inference of $x_T$” may be ambiguous; specifying this as “estimating the latent variable $x_T$ conditioned on $x_0$” would clarify the intent. Similarly, the phrase “latent variable inference,” which appears in Section 4.1, could be made more precise by using “estimating $x_T$ conditioned on $z$” or “encoding $x_0$ into $x_T$,” depending on the context. Certain expressions, like "$D_{KL}(q_{\theta}^{\textrm{ODE}}||p_{\textrm{prior}})$ explains the information split problem” (Lines 196–197), are unclear; it would help to specify whether the KL divergence term’s presence illustrates or quantifies this problem. Ensuring precise and consistent terminology throughout would enhance readability and help avoid potential confusion (see “Questions” for further specific comments).

**Providing Precise Description of Methodology/Tasks in Sections 4 & 5:**
- *Methodology in Section 4:* The presentation of the proposed methodology in Section 4 is complex and challenging to follow. A concise pseudocode or boxed algorithm format summarizing DBAE’s key steps, including training and inference procedures, would streamline comprehension. Providing a pseudocode outline for DBAE’s main processes—inferring $z$ from $x_0$, subsequently inferring $x_T$ from $z$, and bridging $x_0$ and $x_T$ using Doob’s $h$-transform—would help organize the workflow and provide readers with a structured overview before diving into the details.
-  *Experimental Tasks in Section 5:* Section 5 could also benefit from a more systematic presentation of the six experimental tasks under investigation. Providing a precise, mathematical description of each task would make it easier for readers to interpret DBAE’s advantages across different contexts. For example, the description of Interpolation problem in Lines 474 - 476 can be a good starting point -- presenting all tasks in a similar manner would help readers understand the desiderata and challenges in each tasks as well as DBAE’s benefits across varied applications in a more structured way.

**Questions:**

Line 46: The phrase “and the endpoint $x_T$” may be unclear. Would it be more precise to revise this to “to the endpoint $x_T$” to clarify the relationship between $z$ and $x_T$?

Line 49: The term “inference of $x_T$” might be ambiguous for some readers. Could this be rephrased or annotated as “estimating $x_T$ conditioned on $x_0$” or “encoding $x_0$ into $x_T$”?

Lines 75–80: The third paragraph of the Introduction could benefit from additional clarity. The challenges described here seem to presuppose the goal of compressing $x_0$ into latent variables ($z$ and $x_T$) for reconstruction, but this goal is not explicitly stated, which could cause confusion. For example, readers focused on new sample generation might not understand why the information split issue affects reconstruction. Would the authors consider clarifying this context and explaining why reconstruction is a primary focus?

Line 99: Consider capitalizing “eq.” to “Eq.” when referencing specific equations. Also, while referencing equations before presenting them can add context, it is uncommon and might disrupt the logical flow. Would it be more effective to present equations first, then reference them?

Lines 160–161: It might be helpful to provide the full names of “VE” and “VP” (at least in a footnote), rather than only using acronyms, to support readers who may be unfamiliar with these terms.

Lines 196–197: For clarity and parallelism, consider specifying that $D_{KL}(q_{\theta}^{\textrm{ODE}}, p_{\textrm{prior}})$ represents the KL divergence between $q_{\theta}^{\textrm{ODE}}$ and $p_{\textrm{prior}}$. The phrase “$D_{KL}$ explains the information split problem” is also somewhat ambiguous. Could the authors clarify whether the KL divergence term’s presence illustrates or quantifies the problem? Additional context could improve reader understanding.

Lines 203–204: Since the references here were cited just two paragraphs above, the authors may want to omit them to save space.

Line 213: The title of Section 4.1, “latent variable inference,” may be unclear—does it refer to $z$, $x_T$, or both? Clarifying whether the authors mean both variables, and if so, how they relate, would improve clarity.

Lines 214–215: The nature of the encoder and decoder is not clearly defined. For instance, is $\text{Enc}{\phi}$ deterministic or stochastic? If it is deterministic, how does it define the conditional probability $q{\phi}$? Clarifying this would assist readers in understanding the model structure.

Section 4: A concise pseudocode format summarizing the main DBAE algorithm, including training and inference procedures, would be helpful. Consider adding boxed pseudocode for core processes, such as (1) inferring latent variables for reconstruction or sample generation, and (2) training the encoder and decoder. This structured overview could improve comprehension before diving into the details.

Section 4.1: It would be helpful if the authors could confirm the proposed procedure: (1) infer $z$ from $x_0$, (2) infer $x_T$ from $z$, and (3) use Doob’s $h$-transform via Eq. (10) to bridge $x_0$ and $x_T$ and determine the distribution of intermediate $x_t$ values. Additionally, it is unclear if the encoder $q_{\phi}$ and decoder $q_{\psi}$ require training. Could the authors clarify this?

Line 285–286: The statement “For downstream inference, attribute manipulation, and interpolation, the model requires reconstruction capability” could be introduced earlier in the Introduction to help illustrate the utility of generative modeling and contextualize the information split problem. Would the authors consider adding this as a motivation for addressing the information split issue?

Line 317 (Theorem 1): The term “linear SDE” appears without prior explanation. Would the authors consider providing a brief definition?

Line 331: Consider removing “the” before “Section 4.4.1” for grammatical clarity.

Line 342: Replacing “optimize” with “minimize” could provide greater clarity regarding the objective.

Lines 354–355: Essential experimental details should ideally appear in the main text, rather than directing readers to the Appendix. This would allow readers to understand the experimental setup, including encoder and model architecture choices, without needing to consult additional sections. Would the authors consider including these details directly?

Section 5: Could the authors provide a concise, mathematically rigorous description of each of the six experimental tasks to clearly communicate the goals, challenges, and benefits of DBAE? For example, describing the interpolation task in mathematically precise terms, as in Lines 474–476, could serve as a model for presenting all tasks consistently.

---

> ### Author Response · Authors · 2024-11-22
> **Thank you for the questions and feedback**
>
> We appreciate for acknowledging the strengths of our paper in terms of motivation, methodology, and experimental results. We truly appreciate your thoughtful feedback on the presentation, along with detailed suggestions for improvement. Your suggestions have been incorporated into the revised manuscript and are highlighted in blue.
>
> **[Q1]. Clarifying Motivation and Significance in the Introduction**
>
> **[Response to Q1]**
>
> Thank you for the suggestion. To highlight the significance of solving the information split problem, we add the following in the introduction.
> - Line 41: We added a statement to highlight that diffusion models with auxiliary encoder framework are dominating generative representation learning studies.
> - Lines 53-54: We added a statement that diffusion models with auxiliary encoder framework combine the strengths of diffusion models and VAEs.
> - Lines 74-79: We added a fact that the primary goal of generative representation learning is reconstruction, and reconstruction capability is important to facilitate downstream inference, attribute manipulation, and interpolation. We state that the information split problem hinders the reconstruction capability.
>
>
>
>
>
>
>
>
>
> **[Q2]. Improving Precision in Terminology and Expression**
>
> **[Response to Q2]**
>
> Thank you for the thoughtful suggestion. We modified the expression you mentioned for clarity. We modified the following.
> - Line 45: We modified it to “to the endpoint $x_T$“ as you suggested.
> - Lines 48-49: We modified it to “encoding $x_0$ into $x_T$” as you suggested.
> - Lines 160–161: We modified “VP" into its full name, variance preserving.
> - Lines 192-201: First, we would like to inform you that Eq.9 has been slightly revised following Reviewer EWF8's feedback Q2. The revised equation remains equivalent to the original. The previous KL term has now been replaced with a Cross-entropy term, but the explanation flow remains the same. We have modified the explanation of the reason why the Cross-entropy term is problematic. We explain: 1) how the discrepancy between $q _{\theta}^{ODE}(x_T|z, x_0)$ and $p _{prior}(x_T)$ affects the value of $CE(q _{\theta}^{ODE}(x_T|z, x_0) || p _{prior}(x_T))$, 2) how the value of $CE(q _{\theta}^{ODE}(x_T|z, x_0) || p _{prior}(x_T))$ affects the mutual information bound, 3) why the discrepancy between $q _{\theta}^{ODE}(x_T|z, x_0)$ and $p _{prior}(x_T)$ is inevitable.
> - Lines 204-205: We removed the redundant citation as you suggested.
> - Line 339: We removed the unnecessary “the” before “Section 4.4.1” for clarity.
> - Line 350: We modified from “optimize” to “minimize” for clarity.
> - Theorem 1: We added the explanation of “linear SDE” as a footnote.
> - Title of Section 4.1: We have modified the title to “Encoding from $x_0$ to $x_T$ conditioned on $z$,” specifying which latent variables are inferred and clarifying their relationships.
> - Equations: We modified every “eq” to “Eq” for readability. We modify the equations to appear before the referencing.

---

> > ### Author Response · Authors · 2024-11-22
> > **continue**
> >
> > **[Q3]. Providing Precise Description of Methodology**
> >
> > **[Response to Q3]**
> >
> > We added the following.
> > - Page 6 (Algorithm 1, 2): We added the training and execution for the reconstruction as Algorithm 1 and Algorithm 2 on page 6 of the modified manuscript. These were originally in the appendix, but we moved them to the main text as suggested, as it seems helpful for understanding the overview. The latent variable inference and encoder-decoder training you mentioned are well explained in the corresponding algorithms
> >
> > **[Q3.1] It would be helpful if the authors could confirm the proposed procedure: (1) infer $z$ from $x_0$, (2) infer $x_T$ from $z$, and (3) use Doob’s h-transform via Eq. (10) to bridge $x_0$ and $x_T$ and determine the distribution of intermediate $x_t$ values. Additionally, it is unclear if the encoder $q_{\phi}$ and decoder $q_{\psi}$ require training. Could the authors clarify this?**
> >
> > **[Response to Q3.1]**
> >
> > Yes, the order you mentioned (1), (2), and (3) is correct. We have already explained these steps sequentially in Section 4.1, but you can refer to Algorithm 1 for a clear overview. Additionally, we have stated that $\mathcal{L} _{AE}$ is simultaneously optimized with respect to $\phi$, $\psi$, and $\theta$, as already described in lines 350–356 of the current manuscript. This also will be clearer if you refer to Algorithm 1.
> >
> > **[Q3.2] Are the encoder and decoders stochastic or deterministic?**
> >
> > **[Response to Q3.2]**
> >
> > Our method supports both encoder and decoder architectures and can be deterministic or stochastic. When the encoder is deterministic, conditional probability follows a Dirac Delta distribution. We clarified in lines 236-238 of our modified manuscript. In the experimental section, we fix the decoder as deterministic and varying encoders for both (deterministic/stochastic) options. We clarified the experimental choice in lines 370-372 of our manuscript.
> >
> >
> >
> >
> >
> > **[Q4]. Providing Precise Description of Experiment**
> >
> > **[Response to Q4]**
> >
> > We added the following.
> >
> > - Section 5.1-5.6: We explained each experimental procedure more mathematically. Please refer to the modified text (marked in blue) in each section.
> > - Lines 360-365:  We have provided detailed experimental settings in Appendix Table 5, but including all of this information in the main text is constrained by space limitations. However, we agree that the architectural choices for the encoder, decoder, and score network, which are important to our model, should be described in the main text. We added it.

---

> > > ### Comment · Reviewer_rm4h · 2024-11-26
> > >
> > > I appreciate the authors' efforts to address my concerns and revise the manuscript. While I remain somewhat unconvinced about the importance of the information split problem, the presentation has been significantly improved. Accordingly, I am willing to raise my score to 5.

---

> > > > ### Author Response · Authors · 2024-11-27
> > > >
> > > > Thank you for taking the time to read our response and the updated manuscript.
> > > >
> > > > The information split into ($z$, $x_T$) poses problems because the latent variable $x_T$ has the following disadvantages, as we mentioned in lines 122–132:
> > > >
> > > > - **(Inflexible in dimension)**: The dimension of $x_T$ must be the same as the data dimension, which is typically large. This makes it difficult to learn a “compact representation”. This hinders its use in downstream inference (Section 5.1), disentanglement (Section 5.3), and attribute manipulation (Section 5.6). For this reason, the baseline only uses $z$ for these applications, and $z$ is less informative because of the information split problem. This results in poor baseline performance, as shown in Table 1 and Table 3.
> > > >
> > > > - **(Computationally expensive to obtain)**: To obtain $x_T$, the baseline framework needs to solve the ODE in Eq. (4) from $x_0$ to $x_T$, which is computationally expensive, as shown in Table 8. This makes it slow to use in reconstruction (Section 5.2), interpolation (Section 5.5), and attribute manipulation (Section 5.6).
> > > >
> > > > The issue is that one part of the information split is $x_T$, which has unfavorable characteristics. If you could specify the points where you remain unconvinced about the information split problem, I would be happy to provide a more detailed explanation.

---

> > > > > ### Comment · Reviewer_rm4h · 2024-12-01
> > > > >
> > > > > I appreciate the authors’ response and would like to clarify that I have acknowledged their key points, as summarized in the “Strengths” section of my original review. I agree that the proposed approach is sensible and that some experimental results are promising. However, I find the overarching logic of the argument to remain somewhat unclear.
> > > > >
> > > > > The “inflexible dimension” issue of latent representations, as noted by the authors, is a broader limitation of diffusion models (and flow-based models in general) and does not appear to be inherently tied to the information split problem. The authors’ argument seems to rely on an implicit premise that having both types of latent representations—a compact representation $z$ and a full-dimensional representation $x_T$ (serving as the endpoint of the diffusion process)—is essential, or at least advantageous.
> > > > >
> > > > > For tasks such as reconstruction (Section 5.2) and interpolation (Section 5.5), I agree that computing $x_T$ from $x_0$ via ODE solving could be computationally challenging if $x_T$ is indeed necessary. However, it is unclear why $z$ alone would not suffice for these tasks. This again points to the need for a clear and convincing justification of the premise that both $z$ and $x_T$ are required or beneficial.
> > > > >
> > > > > While the authors appear to suggest this premise through their literature review in the Introduction and empirical results, it is not explicitly argued in a compelling way. As a result, while I acknowledge the potential of this work, I remain cautious in fully endorsing it and prefer to maintain my original score. Nonetheless, I would not strongly oppose its acceptance.

---

> ### Author Response · Authors · 2024-12-02
>
> We truly appreciate the reviewer taking the time to review our response. We are well aware of the reviewer's positive points in the original review. In our previous response, we addressed the importance of the information split problem in the application cases presented in Section 5.
>
> As you mentioned, our motivation is based on the premise that having latent representations, $z$ and $x_T$, together is beneficial in the baseline. In addition to the literature and experiments, we also aimed to explicitly explain why the use of both ($z$, $x_T$) is beneficial in the baseline in (Section 3) Motivation: Information Split Problem through Eq. 9 and further explanations in Section A.4.1. The followings elaborates the Section 3 with more details to explains **why the use of both inferred ($z$, $x_T$) is beneficial in the baseline?**
>
> ### 1.  Reconstruction with random $x_T$
> To compute the reconstruction quality with inferred $z$, given by $\mathbb{E} _{q _{\textrm{data}}(x_0), q _{\phi} (z|x_0)}[\log{p _{\theta}(x_0|z)}]$, it inherently requires drawing $x_T$. The reconstruction quality depends on how $x_T$ is drawn. Consider $p _{\theta}(x_0|z) = \int{p _{\textrm{prior}}(x_T) p _{\theta}^{ODE}(x_0|z,x_T) }dx_T$, which implies drawing $ x_T \sim  p _{\textrm{prior}}(x_T)$. Then, the following inequality holds (please refer to Eqs.102-109 for the derivation).
>
> (Eq.D1).  $ \mathbb{E} _{q _{\textrm{data}}(x_0), q _{\phi} (z|x_0)}[-CE(q _{\theta}^{ODE}(x_T|z,x_0) || p _{\textrm{prior}}(x_T))] \leq \mathbb{E} _{q _{\textrm{data}}(x_0), q _{\phi} (z|x_0)}[\log{p _{\theta}(x_0|z)}] $
>
> ### 2.  Reconstruction with inferred $x_T$
> Instead of a random draw $x_T \sim p _{\textrm{prior}}(x_T)$, $x_T$ can also be obtained by encoding from $x_0$ to $x_T$ which implies $x_T \sim q _{\theta}^{ODE}(x_T|x_0,z)$. From this drawing, we can get the following inequality.
>
> (Eq.D2). $ \mathbb{E} _{q _{\textrm{data}}(x_0), q _{\phi} (z|x_0)}[-CE(q _{\theta}^{ODE}(x_T|z,x_0) || q _{\theta}^{ODE}(x_T|z,x_0))] \leq \mathbb{E} _{q _{\textrm{data}}(x_0), q _{\phi} (z|x_0)}[\log{p _{\theta}(x_0|z)}] $
>
>
> ### 3. Compare two reconstruction bounds
> The cross-entropy becomes larger as the discrepancy between two distributions becomes larger. This resulted in the LHS of Eq.D2 being larger than the LHS of Eq.D1. This implies that reconstruction with inferred $x_T$ (with ODE solving) provides a tighter bound on reconstruction quality. Table 2 shows the empirical comparison that DiffAE (inferred $x_T$) performs better than DiffAE (random $x_T$), which supports inequality in Eq.D3.
>
> (Eq.D3). $\mathbb{E} _{q _{\textrm{data}}(x_0), q _{\phi} (z|x_0)}[-CE(q _{\theta}^{ODE}(x_T|z,x_0) || p _{\textrm{prior}}(x_T))] \leq \mathbb{E} _{q _{\textrm{data}}(x_0), q _{\phi} (z|x_0)}[-CE(q _{\theta}^{ODE}(x_T|z,x_0) || q _{\theta}^{ODE}(x_T|z,x_0))]$
>
>
> ### 4. Reconstruction quality and representation quality
> The reconstruction quality with inferred $z$ bounds the mutual information between $z$ and $x_0$ (please refer to Eq. 101 in the manuscript for details). Therefore, tighter bounds on reconstruction quality also result in tighter bounds on the mutual information. The higher mutual information indicates better representation quality, which can facilitate downstream inference (Sec 5.1), disentanglement (Sec 5.3), and attribute manipulation (Sec 5.6).
>
> (Eq.D4). $\mathbb{E} _{q _{\textrm{data}}(x_0), q _{\phi} (z|x_0)}[\log{p _{\theta}(x_0|z)}] +\mathcal{H}(q _{\textrm{data}}(x_0)) \leq MI(x_0, z)$

---

### Official Review · Reviewer_Ffum · 2024-11-03

**Soundness:** 3
**Presentation:** 3
**Contribution:** 3
**Rating:** 8
**Confidence:** 3

**Summary:**

This paper proposes a new Diffusion VAE model in order to combine the latent representation power of VAE and the generation power of Diffusion models. In previous Diffusion VAEs, the latent variable $z$ and $x_T$ are encoded separately from the data $x_0$. The latent variable is concatenated during the denoising process. However, the starting point $x_T$ is not dependent on $z$. The authors call this problem an information split, which can cause unfaithful reconstruction from $z$ and a large gap of mutual information. Motivated by this, the authors introduce another decoder from $z$ to $x_T$. Therefore, $x_T$ can obtain information from the original data. This dependence reduces the mutual information gap, and enhances the representation learning performance. The overall model appears to be a concatenation of VAE and the diffusion process. The authors conduct comprehensive experiments to show advantages in latent variable learning and generation.

**Strengths:**

1. Combining VAE with diffusion is of practical interest. The motivation and solution seem reasonable to me. The authors also provide some theoretical justifications of the model and objective functions.

2. There are thorough experiments to demonstrate the usefulness and performance of the proposed model.

**Weaknesses:**

1. In the diffusion bridge Eq. (5), the input is transformed into a fixed target. However, in the proposed model, the target $x_T$ is random due to the randomness of the VAE. Does this affect the theory?

2. In Section 4.4, the authors present objective functions for reconstruction and generation tasks. In experiments, did the authors optimize different objectives and use different models for different tasks? Can we obtain both reconstruction and generation abilities using the same model?

3. I am wondering whether the training is difficult compared to without the AE part. Is there a trade-off between the expressive power of the AE and diffusion model?

**Questions:**

See above

---

> ### Author Response · Authors · 2024-11-22
> **Thank you for the questions and feedback**
>
> We appreciate the reviewer for the constructive and thoughtful feedback. We answer for reviewer’s comments below.
>
> **[Q1]. In the diffusion bridge Eq. (5), the input is transformed into a fixed target. However, in the proposed model, the target $x_T$ is random due to the randomness of the VAE. Does this affect the theory?**
>
> **[Response to Q1]**
>
> Thank you for the helpful feedback. Doob’s $h$-transform is applicable when we wish to modify the original SDE to terminate at a certain random variable (which has a distribution). Please refer to Theorem 7.11 of [C1] for the explanation of Doob’s h-transform. Its derivation does not restrict the endpoint to being a fixed point but allows it to be a random variable that has a probability distribution. Section 7.3. of [C2] also discusses doob’s h-transform with stochastic endpoints.
>
> **[Q2]. In Section 4.4, the authors present objective functions for reconstruction and generation tasks. In experiments, did the authors optimize different objectives and use different models for different tasks? Can we obtain both reconstruction and generation abilities using the same model?**
>
> **[Response to Q2]**
>
> Thank you for the good question. The objective for reconstruction is $\mathcal{L} _{AE}$ in Eq. 17, and the objective for generative modeling is $\mathcal{L} _{AE} + \mathcal{L} _{PR}$, as shown in Theorem 3. As we already explained in lines 350–356, we optimize $\mathcal{L} _{AE}$ with respect to the parameters of the encoder ($\phi$), decoder ($\psi$), and score network ($\theta$), which enables reconstruction. With the fixed parameters ($\phi, \psi, \theta$), we optimize $\mathcal{L} _{PR}$ with respect to the additional parameter $\omega$, which estimating the generative prior distribution $p _{\omega}(z)$. Consequently, the same parameters ($\phi, \psi, \theta$) are used for both experiments, while the generative prior ($\omega$) is additionally used specifically for generative modeling.
>
> If we set the generative prior distribution $p_{prior}(z)$ to be Gaussian, it is possible to optimize $\mathcal{L} _{AE}+\mathcal{L} _{PR}$ simultaneously, performing reconstruction and generative modeling at the same time (similar to VAE). However, this approach suffers from the same issues below faced by VAEs.
>
> - *Representation Learning*: The $L_{PR}$ in Eq.18 forces the matching of the encoding distribution $q_{\phi,\psi}(x_T|x_0):= \int{q_{\psi}(x_T|z)q_{\phi}(z|x_0)dz}  $ and generative prior distribution $p_{\psi}(x_T):=\int p_{\psi}(x_T|z) p_{prior}(z)dz$, which occur over-regularization on the encoding $q_{\phi}(z|x_0)$ when $p_{prior}(z)$ set to be Gaussian (which hinders to learn semantic meaning). Disjoint training enables optimization on encoding parameters ($\phi,\psi$) without being regularized.
>
> - *Generative Modeling*: The mismatch between the prior distribution $p_{prior}(z)$ and the aggregate posterior $q_{\phi}(z)$ is one of the reasons for the low performance of VAE in unconditional generation [C3]. Imitating learned aggregate posterior $q_{\phi}(z)$ with a powerful generative model is an effective way to mitigate the discrepancy.
>
> Therefore, the separate training approach was used in the baselines [C4, C5] and we follow this.
>
>
>
> [C1] Särkkä, S., & Solin, A. (2019). Applied stochastic differential equations (Vol. 10). Cambridge University Press.
>
> [C2][ICLR 2024] Denoising Diffusion Bridge Model.
>
> [C3][NeurIPS 2021] A contrastive learning approach for training variational autoencoder priors
>
> [C4][CVPR 2022] Diffusion Autoencoders: Toward a Meaningful and Decodable Representation
>
> [C5][NeurIPS 2022] Unsupervised Representation Learning from Pre-trained Diffusion Probabilistic Models.

---

> > ### Author Response · Authors · 2024-11-22
> > **continue**
> >
> > **[Q3]. I am wondering whether the training is difficult compared to without the AE part. Is there a trade-off between the expressive power of the AE and diffusion model?**
> >
> > **[Response to Q3]**
> > Thank you for the question. As far as we understand, your question might be whether our encoding scheme is also beneficial for generative training compared to the vanilla diffusion model (encoding is just noise injection). The semantic encoding capability and generative capability are helpful to each other.
> >
> > The below equations describe our objective $\mathcal{L} _{AE}$ as shown in Eq. 17, and the vanilla diffusion model’s objective $\mathcal{L} _{\text{vanilla}}$ in the analogous form. While the vanilla diffusion model predicts the denoised sample based only on the perturbed sample $x_t$, our score network uses the encoded representation $x_T \sim q _{\phi,\psi}(x_T | x_0)$ that has auxiliary information of $x_0$, which makes reconstruction easier. Some papers [C5, C6] propose empirical observations that the auxiliary informative representation can help generative learning.
> >
> > $\mathcal{L} _{AE}=\frac{1}{2}\int_0^T \mathbb{E} _{q^{t} _{\phi,\psi}(x_0,x_t,x _T)}[\lambda (t) || x _{\theta}^0(x_t, t, x_T) - x_0||_2^2],$
> >
> > $\mathcal{L} _{\text{vanilla}}=\frac{1}{2}\int_0^T \mathbb{E} _{q^{t}(x_0,x_t)}[\lambda (t) || x _{\theta}^0(x_t, t) - x_0||_2^2],$
> >
> > Conversely, from the perspective of the encoding distribution $q^{t} _{\phi,\psi}(x_0,x_t,x _T)$, it receives a training signal aimed at producing a good representation that facilitates reconstruction, as already discussed in lines 319-321. Therefore, the encoding capability and the generative capability of diffusion models positively influence each other.
> >
> >
> > [C6][NeurIPS 2024] Return of Unconditional Generation: A Self-supervised Representation Generation Method

---

> > > ### Comment · Reviewer_Ffum · 2024-11-27
> > >
> > > Thanks for the authors' clarification. I think my concerns are all addressed and I will keep my score.

---

### Official Review · Reviewer_GqCP · 2024-11-04

**Soundness:** 3
**Presentation:** 3
**Contribution:** 3
**Rating:** 8
**Confidence:** 3

**Summary:**

The paper proposes Diffusion Bridge AutoEncoders to create latent variable-dependent endpoint embeddings, resolving the information split problem due to the auxiliary encoder and fixed and inflexible dimension problems in diffusion-based models.

With theoretical guarantees, the method outperforms the SOTA methods on various datasets and settings.

**Strengths:**

Quality and clarity: the paper is well-written and the idea is motivated and easy to follow.

Originality: the paper aims to solve the so-called information-split problem in the field. The effectiveness of the proposed method is well-supported by theorems and comprehensive experiments.

The theorems indicate the proposed loss can indeed increase the mutual information between the input images and the latent variable. Moreover, the generated data distribution is guaranteed to be close to the input data distribution by the proposed loss.

The experiments show substantial improvement in the proposed DBAE method in terms of downstream tasks, reconstruction, disentanglement, and generation. Furthermore, the inference speed is also superior obviously.

**Weaknesses:**

1. It is unclear how the loss makes the endpoint dependent on the latent variable. The theorems only show the relationship between the input data and the latent variable instead, which is not aligned with the claim.

2. The intuition is missing in Section 4.1. For example, why the new forward SDE is defined as in Eq. 10? Providing more intuition would help readers to appreciate the forward process.

**Questions:**

1. Could you provide more intuition behind Eq.10?

2. The paper proposes the AutoEncoder structure for the Diffusion-based representation learning methods. Would it also be interesting to appreciate the work from a (Variational)AE perspective? Will this work provide insights to both sides as a bridge in the middle?

3. Lastly, why and when the split of information will be problematic? For example, in the generation step, the information will be combined from both the latent variable and the endpoints. Thus, is it really necessary to have the information stored or learned in a single representation?

---

> ### Author Response · Authors · 2024-11-22
> **Thank you for the questions and feedback**
>
> We appreciate the reviewer's sincere and helpful feedback. We answer for reviewer’s comments below.
>
> **[Q1]. It is unclear how the loss makes the endpoint dependent on the latent variable. The theorems only show the relationship between the input data and the latent variable instead, which is not aligned with the claim.**
>
> **[Response to Q1]**
>
> Thank you for your feedback. The conditional dependencies between all random variables are determined by the proposed graphical model design in Figure 1-(c), DBAE follows an encoding path ($x_0 \rightarrow z \rightarrow x_T$) supporting our claim that $z$ becomes an information bottleneck. This conditional dependency naturally influences the loss term in $\mathcal{L}_{AE}$.
>
> $\mathcal{L}_{AE}=\frac{1}{2}\int_0^T \mathbb{E} _{q^{t} _{\phi,\psi}(x_0,x_t,x _T)}[\lambda (t) || x _{\theta}^0(x_t, t, x_T) - x_0||_2^2],$
>
> where the probability under the expectation $ q^t _{\phi,\psi}(x _0,x _t,x _T)$ is decomposed according to the graphical model structure in Figure 1-(c).
>
> $ q^{t} _{\phi,\psi}(x_0, x_t, x_T) =\int q _{data}(x_0) q _{\phi}(z|x_0) q _{\psi}(x_T|z) q_t(x_t|x_T, x_0) dz,$
>
> where $q _{\phi}(z|x_0)$ is defined by the encoder,  $q _{\psi}(x_T|z)$ is defined by the decoder, and $q _t(x_t|x_T,x_0)$ is derived from the forward process in Eq.10. We show the objective $\mathcal{L} _{AE}$ (which inherently incorporates our conditional dependency design) has a relationship with mutual information between $x_0$ and $z$ in Theorem 2. The original manuscript mentioned the decomposition of $ q^{t} _{\phi,\psi}(x_0, x_t, x_T)$ separately in line 240, and Eq.10. We have included the entire decomposition in Theorem 1 to help readers.
>
> **[Q2]. The intuition behind Eq.10 is missing, why new SDE is required?**
>
> **[Response to Q2]**
>
> Thank you for your feedback. Figure 1-(a, c) show the graphical model of baseline and DBAE. In the baseline approach, the encoding paths $(x_0 \rightarrow x_T)$ and $ (x_0 \rightarrow z)$ existed independently, making it possible to reuse the vanilla diffusion process for $(x_0 \rightarrow x_T)$. However, in our new structure, the encoding path is defined as $(x_0 \rightarrow z \rightarrow x_T) $, which introduces a predefined coupling between $(x_0, x_T)$.
>
> As a result, a new forward process starting at $x_0$ and ending at $x_T$ needed to be defined. To address this, we employed Doob's $h$-transform to define the new forward process, as shown in Eq. (10). As explained in Section 2.3, Doob's $h$-transform is a transformation that can be used to modify an existing SDE to ensure it terminates at a desired endpoint. The original manuscript explained the intuition behind the need for the newly defined forward SDE in the main text, now found in lines 239–242 of the modified manuscript.
>
> **[Q3]. The paper proposes the AutoEncoder structure for the Diffusion-based representation learning methods. Would it also be interesting to appreciate the work from a (Variational)AE perspective? Will this work provide insights to both sides as a bridge in the middle?**
>
> **[Response to Q3]**
>
> Thank you for your interesting suggestion. The encoding paths of AE and DBAE are the same, $(x_0 \rightarrow z)$. However, there is a difference in the process of generating the final sample $x_0$ starting from $z$. AE typically predicts the final sample through a single-step feed-forward architecture $(z \rightarrow x_0)$. In contrast, as explained in Section 4.2, DBAE involves a more complex generation process, which comprises a single-step feed-forward $(z \rightarrow x_T)$ and the reverse process $(x_T \rightarrow x_0)$ in Eq. 12.
>
> If DBAE denoises Eq. 12 in one step and treats the decoder ($\psi$) and score network ($\theta$) as a single neural network, DBAE aligns exactly with AE. However, by iteratively performing Eq. 12 over multiple steps, we can generalize the generative process to be more complex, thereby creating opportunities to improve generation performance compared to AE.

---

> > ### Author Response · Authors · 2024-11-22
> > **Continue**
> >
> > **[Q4]. Lastly, why and when the split of information will be problematic? For example, in the generation step, the information will be combined from both the latent variable and the endpoints. Thus, is it really necessary to have the information stored or learned in a single representation?**
> >
> > **[Response to Q4]**
> >
> > Thank you for your thoughtful feedback. As the title of the paper implies, the focus is primarily on improving the representation learning capability in diffusion models. Learning compact and informative representation is important in this topic. Let me explain some application examples of why representation learning capability is important in diffusion models.
> >
> >  - Image manipulation (sec 5.6): Informative $z$ makes it easier to find directions for manipulating test data, as shown in Figure 6. Even though $x_T$ may contain information, finding a manipulation direction in the space of $(z, x_T)$ is not appropriate in the baseline because $x_T$ is unsuitable for semantic manipulation due to its high dimensionality [B1].
> >
> >  - Downstream inference (sec 5.1): Informative $z$ facilitates the downstream discriminative tasks. However, it is hard to utilize $(z, x_T)$ as a discriminative task, because $x_T$ has high dimensionality, and compact representation is important for this task.
> >
> > Rather than the split itself, the issue lies in splitting into $x_T$. As mentioned in lines 122–132, $x_T$ has inflexible dimensionality and requires solving an ODE to obtain. $x_T$ is not proper to utilize Image manipulation and downstream inference.
> >
> > [B1][NeurIPS 2021] D2C: Diffusion-Decoding Models for Few-Shot Conditional Generation

---

> > > ### Comment · Reviewer_GqCP · 2024-11-30
> > >
> > > Thank you for the rebuttal. My confusions are cleared. I will keep the score.

---

### Official Review · Reviewer_EWF8 · 2024-11-04

**Soundness:** 4
**Presentation:** 3
**Contribution:** 3
**Rating:** 8
**Confidence:** 3

**Summary:**

The paper identifies The paper identifies a problem in diffusion-based representation learning models with auxiliary (lower dimensional) latent variables, which the authors call the "information split problem".
The identified problem is that such models rely on both the auxiliary latents $z$ and the diffusion endpoint $x_T$ when reconstructing (or unconditionally generating) data.
Thus, even a well-trained model (i.e., a model that can reconstruct data well) may not encode all relevant features of a given data point $x_0$ in $z$ and instead rely also on $x_T$ when reconstructing it.
This makes $z$ less useful as a representation of $x_0$ in downstream tasks (e.g., classification, clustering).

The paper proposes to resolve this issue by turning $z$ into an information bottleneck, thus ensuring that $x_T$ is conditionally independent from $x_0$ given $z$ (i.e., $x_T$ cannot contain any additional information about $x_0$ that is not already encoded in $z$).
The paper discusses and empirically evaluates training objectives of their proposed architecture for representation learning, reconstruction, and unconditional generation.

**Strengths:**

The paper addresses an important issue in representation learning with diffusion models.
The proposed solution appears theoretically sound and empirically convincing to me, but I have to admit that I am not an expert on the relevant literature regarding other solutions to the discussed issue (if any), nor on state-of-the-art empirical performance, so I will yield to the judgment of the other reviewers in this regard.

The paper is remarkably well organized.
Despite addressing a rather complicated problem in a domain that requires a lot of prior knowledge, the paper manages to be self-contained and discusses all relevant background in a brief yet (as far as I can tell) complete way.
The addressed problem is introduced in a didactic way (first with examples, then gradually more formal), and its proposed solution is well motivated.

**Weaknesses:**

I am not an expert on diffusion models with auxiliary encoders, so I cannot judge the novelty of the proposed method or the choice of baselines in the empirical evaluation.

I only identified minor weaknesses (although I strongly recommend addressing at least the last point below as it should be relatively easy to fix and would make the paper much more accessible).

I did not understand the significance of the information split problem in unconditional generation (Section 4.4.2).
I appreciate that the discussion of the objective function (Section 4.4) distinguishes between different use cases of diffusion models.
And I can see that the described information split problem can be an issue for representation learning.
But if all we care about are good unconditional samples, then why does it matter whether $x_T$ may contain some information that is not contained in $z$?
We would sample $x_T$ anyway.
Yet, empirically, it seems like the proposed DBAE+AE model outperforms the baselines even for unconditional generation.

Further, as a minor point, Eq. 9 seems suspicious to me.
While I understand how a bound of this form would exist in general, in the specific instance both $p_\theta^\text{ODE}(x_0 | z, x_T)$ and $p_\theta^\text{ODE}(x_T |z_0, z)$ are delta-distributions unless I am mistaken, so the log-density of the former does not exist, and I would expect the KL-divergence term that involves the latter to be infinite.
I presume that the "two infinities cancel" in some sense, but maybe the equation could be rewritten in a way that doesn't rely on "cancellation of infinities".

Apart from this, my only criticism concerns the large number of grammatical errors, which made the paper quite a bit more difficult to read for me than necessary.
This is a shame because it diminishes the otherwise excellent presentation.
I know that being overly pedantic on grammar in scientific papers can impose cultural bias, but I am fairly certain that most of the errors could be caught by automatic tools.
And while many errors are somewhat trivial (singular/plural or missing/superfluous/wrong articles), some sentences were right-out unintelligible to me and I had to pause reading and infer the intended meaning from context.
Some examples include (there are more): lines 132-133, lines 160-162 (two separate sentences that were probably meant as one, but this isn't immediately apparent due to remaining grammar issues even when joining them), and lines 287-288 (missing verb; I actually can't infer the intended meaning here).
I strongly recommend running the entire paper through a grammar checker, or asking an LLM to detect possible grammar issues in each paragraph.
It would make this otherwise great work more accessible to a broad (international) audience.

**Questions:**

What is the significance of the information split problem in unconditional generation (see "Weaknesses")?

---

> ### Author Response · Authors · 2024-11-22
> **Thank you for the questions and feedback**
>
> We appreciate the reviewer for the constructive and thoughtful feedback. We answer for reviewer’s comments below.
>
> **[Q1]. What is the significance of the information split problem in unconditional generation?**
>
> **[Response to Q1]**
>
> Thank you for your valuable feedback. As the title of the paper implies, the primary focus is on improving the representation learning capability based on reconstruction (we further highlighted in the introduction, lines 73–78, in the modified manuscript). The enhanced representation capability facilitates downstream tasks such as inference (Sec. 5.1), reconstruction (Sec. 5.2), disentanglement (Sec. 5.3), interpolation (Sec. 5.5), and attribute manipulation (Sec. 5.6). As you pointed out, the information split problem is not a significant issue in unconditional generation. Please consider the generative modeling capability as an additional feature of our model.
>
> **[Q2]. Formulation of Eq.9. without "cancellation of infinities"**
>
> **[Response to Q2]**
>
> Thank you for your insightful feedback. You are correct that the phenomenon of " cancellation of infinities " occurs in the previous formulation. We have revised Eq. 9 in the modified manuscript to an equivalent formulation that avoids the issue you mentioned. The updated equation still explains the information split problem. Please refer to Eqs. 106–109 in Appendix A.4.1, where we demonstrate the equivalence between the previous and new formulations.
>
> **[Q3]. Grammar errors**
>
> **[Response to Q3]**
>
> We apologize for the lack of completeness in some sentences, which may have made the paper difficult to read. We sincerely express our gratitude for thoroughly reading the paper and providing valuable feedback. We have comprehensively revised the manuscript, including the parts you mentioned. The modified parts related to your feedback are highlighted in blue. You can refer to lines 131–133, 160–162, and 292–296 in the revised manuscript.

---

> > ### Comment · Reviewer_EWF8 · 2024-11-24
> >
> > Thank you for your detailed response. It resolved all my questions and concerns. Reading the other reviews didn't give me the impression that I missed any important issues with the paper, so I am planning to keep my overall rating.
> >
> > > Please consider the generative modeling capability as an additional feature of our model.
> >
> > Seems totally fair to me (my question was more out of genuine curiosity than as a point of criticism).

---

### Author Response · Authors · 2024-11-25
**A kind reminder for further discussion.**

Dear Reviewers,

We sincerely appreciate your valuable feedback, which is crucial for enhancing the quality of our work. We kindly request that you review our responses, as we are eager to address any further concerns or questions and engage in a meaningful discussion. With the discussion deadline approaching, please let us know if there are any additional questions. Thank you once again for your time and thoughtful consideration.

Best regards,

Authors.

---

### Meta-Review · Area_Chair_B5dr · 2024-12-24

**Metareview:**

The paper introduces Diffusion Bridge AutoEncoders (DBAE) to address the "information split problem" in diffusion-based representation learning models. The proposed method creates a dependency between the latent variable and the diffusion endpoint through a feed-forward architecture, ensuring that the latent variable holds the full information of the data. This approach improves downstream inference, reconstruction, and disentanglement, and generates high-fidelity samples. All reviewer agreed that this paper is well-written, and the idea is motivated and easy to follow. The paper addresses an important issue in representation learning with diffusion models and provides a theoretically sound and empirically convincing solution.

**Additional Comments On Reviewer Discussion:**

Reviewers requested more intuition behind certain equations and problem motivation. Some reviewers also noted grammatical errors and sub-optimal phrasing, which made the paper difficult to read. The authors addressed most of the reviewers' concerns, including providing more intuition and revising the manuscript to improve readability.

---

### Decision · Program_Chairs · 2025-01-22

Accept (Spotlight)